# Supervised learning in spiking neural networks with FORCE training

Wilten Nicola[1] & Claudia Clopath[1]

Populations of neurons display an extraordinary diversity in the behaviors they affect and display. Machine learning techniques have recently emerged that allow us to create networks of model neurons that display behaviors of similar complexity. Here we demonstrate the direct applicability of one such technique, the FORCE method, to spiking neural networks. We train these networks to mimic dynamical systems, classify inputs, and store discrete sequences that correspond to the notes of a song. Finally, we use FORCE training to create two biologically motivated model circuits. One is inspired by the zebra finch and successfully reproduces songbird singing. The second network is motivated by the hippocampus and is trained to store and replay a movie scene. FORCE trained networks reproduce behaviors comparable in complexity to their inspired circuits and yield information not easily obtainable with other techniques, such as behavioral responses to pharmacological manipulations and spike timing statistics.

[1] Department of Bioengineering, Imperial College London, Royal School of Mines, London, SW7 2AZ, UK. Correspondence and requests for materials should be addressed to C.C. (email: c.clopath@imperial.ac.uk)

Human beings can naturally learn to perform a wide variety of tasks quickly and efficiently. Examples include learning the complicated sequence of motions in order to take a slap-shot in Hockey or learning to replay the notes of a song after music class. While there are approaches to solving these different problems in fields such as machine learning, control theory, and so on, humans use a very distinct tool to solve these problems: the spiking neural network.

Recently, a broad class of techniques have been derived that allow us to enforce a certain behavior or dynamics onto a neural network[1–10]. These top-down techniques start with an intended task that a recurrent spiking neural network should perform, and

**Fig. 1** The FORCE method explained. **a** In the FORCE method, a spiking neural network contains a backbone of static and strong synaptic weights that scale like $1/\sqrt{N}$ to induce network level chaos (blue). A secondary set of weights are added to the weight matrix with the decoders determined through a time optimization procedure (red). The Recursive Least Squares technique (RLS) is used in all subsequent simulations. FORCE training requires a supervisor $x(t)$ to estimate with $\hat{x}(t)$. **b** Prior to learning, the firing rates for 5 random neurons from a network of 1000 rate equations are in the chaotic regime. The chaos is controlled and converted to steady-state oscillations. **c** This allows the network to represent a 5 Hz sinusoidal input (black). After learning, the network (blue) still displays the 5 Hz sinusoidal oscillation as its macroscopic dynamics and the training is successful. The total training time is 5 s. **d** The decoders for 20 randomly selected neurons in the network, before ($t < 5$), during ($5 \le t < 10$) and after ($t \ge 10$) FORCE training. **e** The eigenvalue spectrum for the effective weight matrix before (red) and after (black) FORCE training. Note the lack of dominant eigenvalues in the weight matrix

determine what the connection strengths between neurons should be to achieve this. The dominant approaches are currently the FORCE method[1,2], spike-based or predictive coding networks[5–7], and the neural engineering framework (NEF)[8,9]. Both the NEF and spike-based coding approaches have yielded substantial insights into network functioning. Examples include how networks can be organized to solve behavioral problems[8] or process information in robust but metabolically efficient ways[5,6].

While the NEF and spike-based coding approaches create functional spiking networks, they are not agnostic toward the underlying network of neurons (although see refs. [6,11,12] for recent advances). Second, in order to apply either approach, the task has to be specified in terms of closed-form differential equations. This is a constraint on the potential problems these networks can solve (although see refs. [7,13] for recent advances). Despite these constraints, both the NEF and spike-based coding techniques have led to a resurgence in the top-down analysis of network function[3].

Fortunately, FORCE training is agnostic toward both the network under consideration, or the tasks that the network has to solve. Originating in the field of reservoir computing, FORCE training takes a high dimensional dynamical system and utilizes this systems complex dynamics to perform computations[1,14–25]. Unlike other techniques, the target behavior does not have to be specified as a closed form differential equation for training. All that is required for training is a supervisor to provide an error signal. The use of any high dimensional dynamical system as a reservoir makes the FORCE method applicable to many more network types while the use of an error signal expands the potential applications for these networks. Unfortunately, FORCE training has only been directly implemented in networks of rate equations that phenomenologically represent how a firing rate varies smoothly in time, with little work done in spiking network implementations in the literature (although see refs. [2,4] for recent advances).

We show that FORCE training can be directly applied to spiking networks and is robust against different implementations, neuron models, and potential supervisors. To demonstrate the potential of these networks, we FORCE train two spiking networks that are biologically motivated. The first of these circuits corresponds to the zebra finch HVC-RA circuit and reproduces the singing behavior while the second circuit is inspired by the hippocampus and encodes and replays a movie scene. Both circuits can be perturbed to determine how robust circuit functioning is post-training.

## Results

**FORCE training weight matrices.** We explored the potential of the FORCE method in training spiking neural networks to perform an arbitrary task. In these networks, the synaptic weight matrix is the sum of a set of static weights and a set of learned weights $\omega = G\omega^0 + Q\eta\phi^T$. The static weights, $G\omega^0$ are set to initialize the network into chaotic spiking[26–28]. The learned component of the weights, $\phi$, are determined online using a supervised learning method called Recursive Least Squares (RLS). The quantity $\phi$ also serves as a linear decoder for the network dynamics. The parameter $\eta$ defines the tuning preferences of the neurons to the learned feedback term. The components of $\eta$ are static and always randomly drawn. The parameters $G$ and $Q$ control the relative balance between the chaos inducing static weight matrix and the learned feedback term, respectively. The parameters are discussed in greater detail in the Methods section. The goal of RLS is to minimize the squared error between the network dynamics ($\hat{x}(t)$) and the target dynamics (i.e., the task, $x(t)$) (Fig. 1a)[29,30]. This method is successful if the network

**Table 1 Model parameters**

| Neuron model | Parameter | Value |
|---|---|---|
| Izhikevich | $C$ | 250 μF |
| | $v_r$ | −60 mV |
| | $v_t$ | −20 mV (−40 mV songbird example) |
| | $b$ | 0 nS |
| | $v_{peak}$ | 30 mV |
| | $v_{reset}$ | −65 mV |
| | $a$ | 0.01 ms$^{-1}$ (0.002 ms$^{-1}$, songbird example) |
| | $d$ | 200 pA (100 pA, songbird example) |
| | $I_{Bias}$ | 1000 pA |
| | $k$ | 2.5 ns/mV |
| | $\tau_R$ | 2 ms |
| | $\tau_D$ | 20 ms |
| Theta model | $I_{Bias}$ | 0 |
| | $\tau_R$ | 2 ms |
| | $\tau_D$ | 20 ms |
| LIF model | $\tau_m$ | 10 ms |
| | $\tau_{ref}$ | 2 ms |
| | $v_{reset}$ | −65 mV |
| | $v_t$ | −40 mV |
| | $I_{Bias}$ | −40 pA (−39 pA, Lorenz example) |
| | $\tau_R$ | 2 ms |
| | $\tau_D$ | 20 ms |

The parameters used for the Izhikevich, theta, and LIF neuron models, unless otherwise stated

dynamics can mimic the target dynamics when RLS is turned off. We considered three types of spiking integrate-and-fire model: the theta model, the leaky integrate-and-fire (LIF), and the Izhikevich model (see Methods section for a more detailed explanation). The parameters for the models can be found in Table 1. All networks considered were constrained to be intermediate in size (1000–5000 neurons) and have low post-training average firing rates (<60 Hz). The synaptic time constants were typically $\tau_R = 2$ ms and $\tau_D = 20$ ms with other values considered in Supplementary Material. The code used for this paper can be found on modelDB ([31]), under accession number 190565.

**FORCE trained rate networks learn using chaos.** To demonstrate the basic principle of this method and to compare with our spiking network simulations, we applied FORCE training to a network of rate equations (Fig. 1a) as in ref. [1]. We trained a network to learn a simple 5 Hz sinusoidal oscillator. The static weight matrix initializes high-dimensional chaotic dynamics onto the network of rate equations (Fig. 1b). These dynamics form a suitable reservoir to allow the network to learn from a target signal quickly. As in the original formulation of the FORCE method, the rates are heterogeneous across the network and varied strongly in time (see Fig. 2a in ref. [1]). RLS is activated after a short initialization period for the network. After learning the appropriate weights $\phi_j$ (Fig. 1d), the network reproduces the oscillation without any guidance from the teacher signal, albeit with a slight frequency and amplitude error (Fig. 1c). To ascertain how these networks can learn to perform the target dynamics, we computed the eigenvalues of the resulting weight matrix before and after learning (Fig. 1e). Unlike other top-down techniques, FORCE trained weight matrices are always high rank[32], and as we will demonstrate, can have dominant eigenvalues for large $Q$.

**FORCE trained spiking networks learn using chaos.** We sought to determine whether the FORCE method could train spiking

neural networks given the impressive capabilities of this technique in training smooth systems of rate equations (Fig. 1) previously demonstrated in ref. [1]. We implemented the FORCE method in different spiking neural networks of integrate-and-fire neurons in order to compare the robustness of the method across neuronal models and potential supervisors.

First, to demonstrate the chaotic nature of these networks, we deleted a single spike in the network[33]. After deletion, the resulting spike trains immediately diverged, indicating chaotic

behavior (Fig. 2a). To determine where the onset of chaos occurred as a function of $G$, we simulated networks of these neurons over a range of $G$ parameters and computed the coefficients of variation and the interspike-interval (ISI) distributions (Supplementary Fig. 1). For the Izhikevich and Theta neurons, there was an immediate onset to chaotic spiking from quiescence ($G \approx 10^3$, $G \approx 0.02$, respectively) as the bias currents for these models were placed at rheobase or threshold value. For the LIF model, we considered networks with both superthreshold

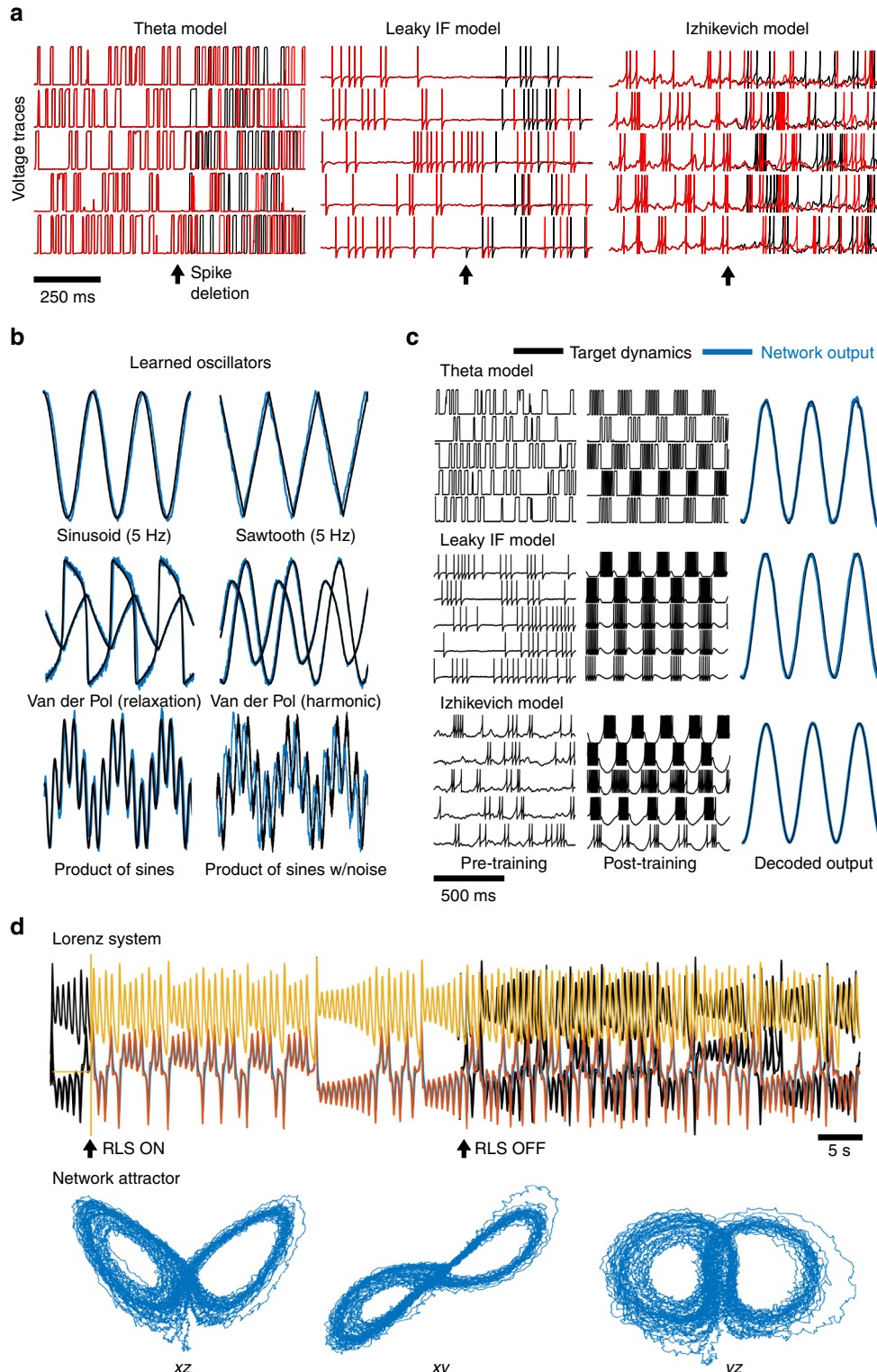

and threshold bias currents (see Table 1 for parameters). In the superthreshold case, the LIF transitioned from tonic spiking to chaotic spiking ($G \approx 0.04$). The subthreshold case was qualitatively similar to the theta neuron model, with a transition to chaotic spiking at a small $G$-value ($0 < G < 0.01$). All neuron models exhibited bimodal interspike-interval distributions indicative of a possible transition to rate chaos for sufficiently high $G$-values[27,28]. Finally, a perturbation analysis revealed that all three networks of neurons contained flux-tubes of stability[34], (Supplementary Fig. 1).

To FORCE train chaotic spiking networks, we used the original work in ref. [1] as a guide to determine the parameter regime for $(G, Q)$. In ref. [1], the contributions of the static and learned synaptic inputs are of the same magnitude. Similarly, we scale $Q$ to ensure the appropriate balance (see Methods secion for a derivation on how Q should be scaled). Finally, FORCE training in ref. [1] works by quickly stabilizing the rates. Subsequent weight changes are devoted to stabilizing the weight matrix. For fast learning, RLS had to be applied on a faster time scale in the spiking networks vs. the rate networks ($< O(1)$ ms vs. $O(10)$ ms, respectively). The learning rate $\lambda$ was taken to be fast enough to stabilize the spiking basis during the first presentation of the supervisor.

With these modifications to the FORCE method, we successfully trained networks of spiking theta neurons to mimic various types of oscillators. Examples include sinusoids at different frequencies, sawtooth oscillations, Van der Pol oscillators, in addition to teaching signals with noise present (Fig. 2b). With a 20 ms decay time constant, the product of sines oscillator presented a challenge to the theta neuron to learn. However, it could be resolved for network with larger decay time constants (Supplementary Fig. 2, Supplementary Table 1). All the oscillators in Fig. 2b were trained successfully for both the Izhikevich and LIF neuron models (Fig. 2c, Supplementary Fig. 2, Supplementary Table 1). Furthermore, FORCE training was robust to the possible types of initial chaotic network states (Supplementary Fig. 3)[27,28]. Finally, a three parameter sweep over the $(G, Q, \tau_D)$ parameter space reveals that parameter regions for convergence are contiguous. This parameter sweep was conducted for sinusoidal supervisors at different frequencies (1, 5, 10, 20, and 40 Hz). Oscillators with higher (lower) frequencies are learned over larger $(G, Q)$ parameter regions in networks with faster (slower) synaptic decay time constants, $\tau_D$ (Supplementary Figs. 4–6). Finally, we compared how the eigenvalues of the trained weight matrices varied (Supplementary Material, Supplementary Fig. 7). For spiking networks, we observe that in some cases, systems without dominant eigenvalues performed better than systems with dominant eigenvalues while in other cases the opposite was true.

For the dynamics under consideration, the Izhikevich model had the greatest accuracy and fastest training times (Supplementary Figs. 4–6, Supplementary Table 2). This is partially due to the fact that the Izhikevich neuron has spike frequency adaptation which operates on a longer time scale (i.e., 100 ms). The long time

scale affords the reservoir a greater capability for memory, allowing it to learn longer signals. Additionally, the dimensionality of the reservoir is increased by having an adaptation variable for each neuron.

We wondered how the convergence rates of these networks would vary as a function of the network size, $N$, for both FORCE trained spiking and rate networks (Supplementary Fig. 8). For a 5 Hz sinusoidal supervisor, the spiking networks had a convergence rate of $\approx N^{-1/2}$ in the $L_2$ error. Rate networks had a higher order convergence rate, of $\approx N^{-1}$ in the $L_2$ error.

As oscillators are simple dynamical systems, we wanted to assess if FORCE can train a spiking neural network to perform more complicated tasks. Thus, we considered two additional tasks: reproducing the dynamics for a low-dimensional chaotic system and statistically classifying inputs applied to a network of neurons. We trained a network of theta neurons using the Lorenz system as a supervisor (Fig. 2d). The network could reproduce the butterfly attractor and Lorenz-like trajectories after learning. As the supervisor was more complex, the training took longer and required more neurons (5000 neurons, 45 s of training) yet was still successful. Furthermore, the Lorenz system could also be trained with a network of 5000 LIF neurons (Supplementary Fig. 9). To quantify the error and compare it with a rate network, we developed an attractor density based metric (Supplementary Materials) for the marginal density functions on the attractor. The spiking network had comparable performance to the rate network (0.27, 0.30, 0.24 for rate and 0.52, 0.38, 0.3 for spiking). Further, the spiking and rate networks were both able to regenerate the stereotypical Lorenz tent map, albeit with superior performance in the rate network (Supplementary Fig. 9). Finally, we showed that populations of neurons can be FORCE trained to statistically classify inputs, similar to a standard feedforward network (Supplementary Note 2, Supplementary Figs. 10–13).

We wanted to determine if FORCE training could be adapted to generate weight matrices that respect Dales law, the constraint that a neuron can only be excitatory or inhibitory, not both. Unlike previous work[2], we opted to enforce Dales law dynamically as the network was trained (Supplementary Note 2, Supplementary Fig. 13). The $\log(L_2)$ error over the test interval for this supervisor was $-1.34$, which is comparable to the values in Supplementary Fig. 6. While Dales law can be implemented, for simplicity, we train all remaining examples with unconstrained weight matrices.

To summarize, for these three different neuron models, we have demonstrated that the FORCE method can be used to train a spiking network using a supervisor. The supervisor can be oscillatory, noisy, chaotic, and the training can occur in a manner that respects Dales law.

**FORCE training spiking networks to produce complex signals.** Neurons can encode complicated temporal sequences such as the mating songs that songbirds learn, store, and replay[35]. We

**Fig. 2** Using spiking neural networks to mimic dynamics with FORCE training. **a** The voltage trace for 5 randomly selected neurons in networks of 2000 integrate-and-fire spiking neurons. The models under consideration are the theta neuron (left), the leaky integrate-and-fire neuron (middle), and the Izhikevich model with spike frequency adaptation (right). For all networks under consideration, a spike was deleted (black arrow) from one of the neurons. This caused the spike train to diverge post-deletion, a clear indication of chaotic behavior. **b** A network of 2000 theta neurons (blue) was initialized in the chaotic regime and trained to mimic different oscillators (black) with FORCE training. The oscillators included the sinusoid, Van der Pol in harmonic and relaxation regimes, a non-smooth sawtooth oscillator, the oscillator formed from taking the product of a pair of sinusoids with 4 Hz and 6 Hz frequencies, and the same product of sinusoids with a Gaussian additive white noise distortion with a standard deviation of 0.05. **c** Three networks of different integrate-and-fire neurons were initialized in the chaotic regime (left), and trained using the FORCE method (center) to mimic a 5 Hz sinusoidal oscillator (right). **d** A network of 5000 theta neurons was trained with the FORCE method to mimic the Lorenz system. RLS was used to learn the decoders using a 45 s long trajectory of the Lorenz system as a supervisor. RLS was turned off after 50 s with the resulting trajectory and chaotic attractor bearing a strong resemblance to the Lorenz system. The network attractor is shown in three different viewing planes for 50 s post training

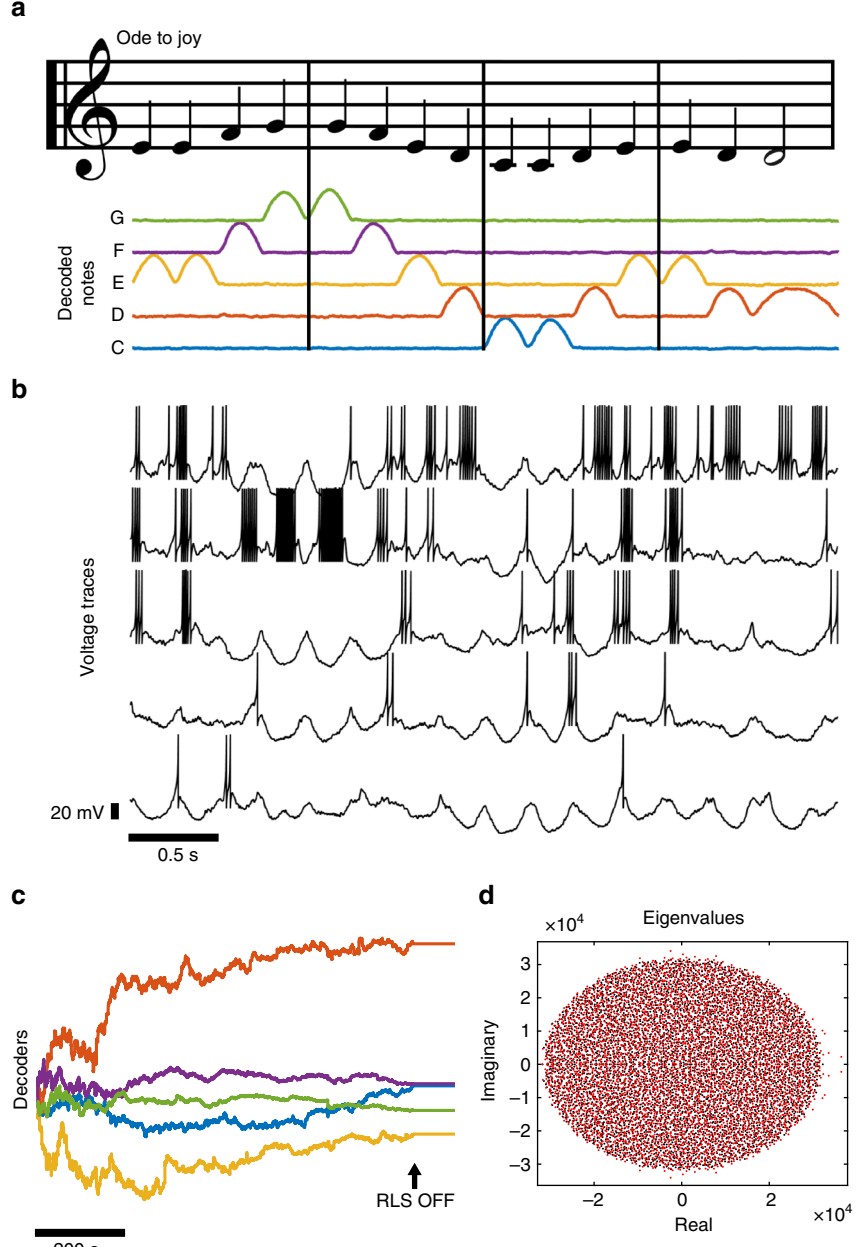

**Fig. 3** Using spiking neural networks for pattern storage and replay with FORCE training. **a** The notes in the song Ode to Joy by Beethoven are converted into a continuous 5-component teaching signal. The sheet music has been modified from Mutopia ([76]) by Wilten Nicola. The presence of a note is designated by an upward pulse in the signal. Quarter notes are the positive portion of a 2 Hz sine wave form while half notes are represented with the positive part of a 1 Hz sine wave. Each component in the teaching signal corresponds to the musical notes **c–g**. The teaching signal is presented to a network of 5000 Izhikevich neurons continuously from start to finish until the network learns to reproduce the sequence through FORCE training. The network output in **a** is after 900 s of FORCE training, or 225 applications of the target signal. For 1000 s of simulation time after training, the song was played correctly 205 times comprising 820 s of the signal (Supplementary Figs. 15, 16). **b** Five randomly selected neurons in the network and their resulting dynamics after FORCE training. The voltage traces are taken at the same time as the approximant in **a**. **c** The decoders before ($1 < t < 900$) and after ($t > 900$) FORCE training. **d** The resulting eigenvalues for the weight matrix $G\omega_{ij}^0 + Q\eta_i\phi_j$ before (black) and after (red) learning. Note that the onset to chaos occurs at $G \approx 10^3$ for the Izhikevich network with the parameters we have considered

wondered whether FORCE could train spiking networks to encode similar types of naturally occurring spatio-temporal sequences. We formulated this as very long oscillations that are repeatedly presented to the network.

The first pattern we considered was a sequence of pulses in a 5-dimensional supervisor. These pulses correspond to the notes in the first bar of Beethoven's Ode to Joy. A network of Izhikevich neurons was successfully FORCE trained to reproduce Ode to Joy (see Fig. 3a, c, d 82% accuracy during testing). The average firing

rate of the network after training was 34 Hz (Fig. 3b), with variability in the neuronal responses from replay to replay, yet forming a stable peri-simulus time histogram (Supplementary Figs. 14–16). Furthermore, Ode to Joy could be learned by all neuron models at larger synaptic decay time constants ($\tau_d = 50$, 100 ms, see Supplementary Figs. 4–6).

While the network displayed some error in replaying the song, the errors were not random but stereotypical. The errors are primarily located after the two non-unique E-note repetitions that

occur in the first bar and the end of the third bar (Supplementary Figs. 14, 15) in addition to the non-unique ED sequences that occur at the end of the second bar and beginning of the fourth bar.

**FORCE trained networks can reproduce songbird singing**. While the Ode to Joy example was non-trivial to train, it pales in complexity to the singing of the zebra finch. To that end, we constructed a circuit that could reproduce a birdsong (in the form of a spectrogram) recorded from an adult zebra finch. The learned singing behavior of these birds is owed to two primary nuclei: the HVC (proper name) and the Robust nucleus of the Arcopallium (RA). The RA projecting neurons in HVC form a chain of spiking activity and each RA projecting neuron fires only once at a specific time in the song[35,36]. This chain of firing is

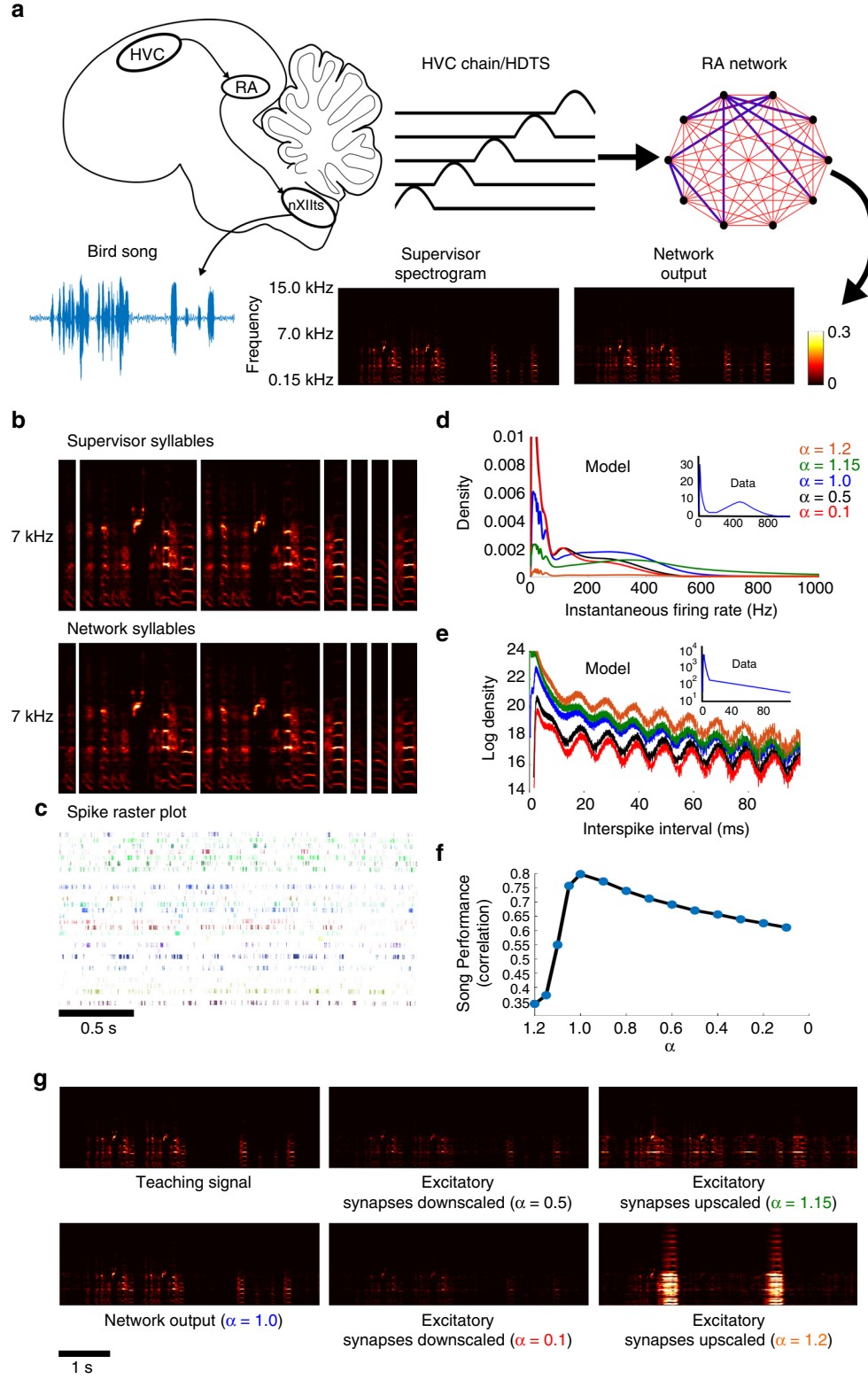

transmitted to and activates the RA circuit. Each RA neuron bursts at multiple precisely defined times during song replay[37]. RA neurons then project to the lower motor nuclei which stimulate vocal muscles to generate the birdsong.

To simplify matters, we will focus on a single network of RA neurons receiving a chain of inputs from area HVC (Fig. 4a). The HVC chain of inputs are modeled directly as a series of successive pulses and are not FORCE trained for simplicity. These pulses feed into a network of Izhikevich neurons that was successfully FORCE trained to reproduce the spectrogram of a recorded song from an adult zebra finch (Fig. 4b, see Supplementary Movie 1). Additionally, by varying the ($G$, $Q$) parameters and some Izhikevich model parameters, the spiking statistics of RA neurons are easily reproduced both qualitatively and quantitatively (Fig. 4d, e inset, see Fig. 2b, c in ref. [37], see Fig. 1 in ref. [37]). The RA neurons burst regularly multiple times during song replay (Fig. 4b, c). Thus, we have trained our model network to match the song generation behavior of RA with spiking behavior that is consistent with experimental data.

We wondered how our network would respond to potential manipulations in the underlying weight matrices. In particular, we considered manipulations to the excitatory synapses which alter the balance between excitation and inhibition in the network (Fig. 4f). We found that the network is robust toward downscaling excitatory synaptic weights. The network could still produce the song, albeit at a lower intensity. This was possible even when the excitatory synapses were reduced by 90% of their amplitude (Fig. 4f, g). However, upscaling excitatory synapses by as little as 15% drastically reduced song performance. The resulting song output had a very different spectral structure than the supervisor as opposed to downscaling excitation. Finally, upscaling the excitatory weights by 20% was sufficient to destroy singing, replacing it with high intensity seizure like activity. Interestingly, a similar result was found experimentally through the injection of bicuculine in RA[38]. Large doses of bicuculine resulted in strong bursting activity in RA accompanied by involuntary vocalizations. Smaller doses resulted in song degradation with increased noisiness, duration, and the appearance of new syllables.

### High-dimensional temporal signals improve FORCE training.
We were surprised at how robust the performance of the songbird network was given the high dimensionality and complicated structure of the output. We hypothesized that the performance of this network was strongly associated to the precise, clock-like inputs provided by HVC and that similar inputs could aid in the encoding and replay of other types of information. To test this hypothesis, we removed the HVC input pattern and found that the replay of the learned song was destroyed (not shown), similar to experimental lesioning of this area in adult canaries[39]. Due to the temporal information that these high-dimensional signals provide, we will subsequently refer to them as high-dimensional temporal signals (HDTS, see Methods section for further details).

To further explore the benefits that an HDTS might provide, we FORCE trained a network of Izhikevich neurons to internally generate its own HDTS, while simultaneously also training it to reproduce the first bar of Ode to Joy. The entire supervisor consisted of the 5 notes used in the first bar of the song, in addition to 16 other components. These correspond to a sequence of 16 pulses that partition time. The network learned both the HDTS and the song simultaneously, with less training time, and greater accuracy than without the HDTS. As the HDTS helped in learning and replaying the first bar of a song, we wondered if these signals could help in encoding longer sequences. To test this hypothesis, we FORCE trained a network to learn the first four bars of Ode to Joy, corresponding to a 16 s song in addition to its own, 64-dimensional HDTS. The network successfully learned the song post-training, without any error in the sequence of notes in spite of the length and complexity of the sequence (see Supplementary Fig. 17, Supplementary Movie 2). Thus, an internally generated HDTS can make FORCE training faster, more accurate, and more robust to learning longer signals. We refer to these networks as having an internally generated HDTS.

### FORCE trained encoding and replay of an episodic memory.
Given the improvements that an HDTS confers over supervisor duration, training time, and accuracy, we wanted to know if these input signals would help populations of neurons to learn natural high dimensional signals. To test this, we trained a network of Izhikevich neurons to learn a 1920 dimensional supervisor that corresponds to the pixels of an 8 s scene from a movie (Fig. 5). Each component of the supervisor corresponds to the time evolution of a single pixel. The HDTS's were either generated by a separate network or were fed directly as an input into an encoding and replay network, as in the songbird example (Fig. 5a, b). We refer to these two cases as the internally generated and the externally generated HDTS, respectively. In the former case, we demonstrate that an HDTS can be easily learned by a network while in the latter case we can freely manipulate the HDTS. As in the long Ode to Joy example, the HDTS could also be learned simultaneously to the movie scene, constituting a 1984 dimensional supervisor (1920 + 64 dimensional HDTS, Supplementary Fig. 18c, d). Note that this can be thought of as spontaneous replay, as opposed to cued recall where the network reconstructs a partially presented stimulus.

We were able to successfully train the network to replay the movie scene in both cases (time averaged correlation coefficient of $r = 0.98$, Fig. 5c). Furthermore, we found that the HDTS inputs (shown in Fig. 5b) were necessary both for training ($r < 0.44$, varies depending on parameters) and replay ($r = 0.25$, Supplementary Fig. 18b). The network could still replay the individual frames from the movie scene without the HDTS; however, the order of scenes was incorrect and appeared to be chaotic. Thus, the HDTS input facilitates both learning and spontaneous replay of high dimensional signals.

---

**Fig. 4** Using spiking neural networks for pattern storage and replay: songbird singing. **a** A series of pulses constructed from the positive portion of a sinusoidal oscillator with a period of 20 ms are used to model the chain of firing output from RA projection neurons in HVC. These neurons connect to neurons in RA and trigger singing behavior. FORCE training was used to train a network of 1000 Izhikevich neurons to reproduce the spectrogram of a 5 s audio recording from an adult zebra finch. **b** The syllables in the song output for the teaching signal (top) and the network (bottom). **c** Spike raster plot for 30 neurons over 5 repetitions of the song. The spike raster plot is aligned with the syllables in **b**. **d** The distribution of instantaneous firing rates for the RA network post training. The $\alpha$ parameter corresponds to the up/down regulation factor of the excitatory synapses and to the different colors in the graphs of **d**, **e**. For $\alpha > 1$, excitation dominates over inhibition while the reverse is true for $\alpha < 1$. The inset of **d**, **e** is reproduced from ref. [37]. Note that the y-axis of the model and data differ, this is due to normalization of the density functions. **e** The log of the interspike interval histogram. **f** The correlation between the network output and the teaching signal as the excitatory synapses are upscaled (negative x-axis) or downscaled (positive x-axis). **g** The spectrogram for the teaching signal, and the network output under different scaling factors for the excitatory synaptic weights. The panels in **g** correspond to the plots in **d**, **e** and the performance measure in **f**

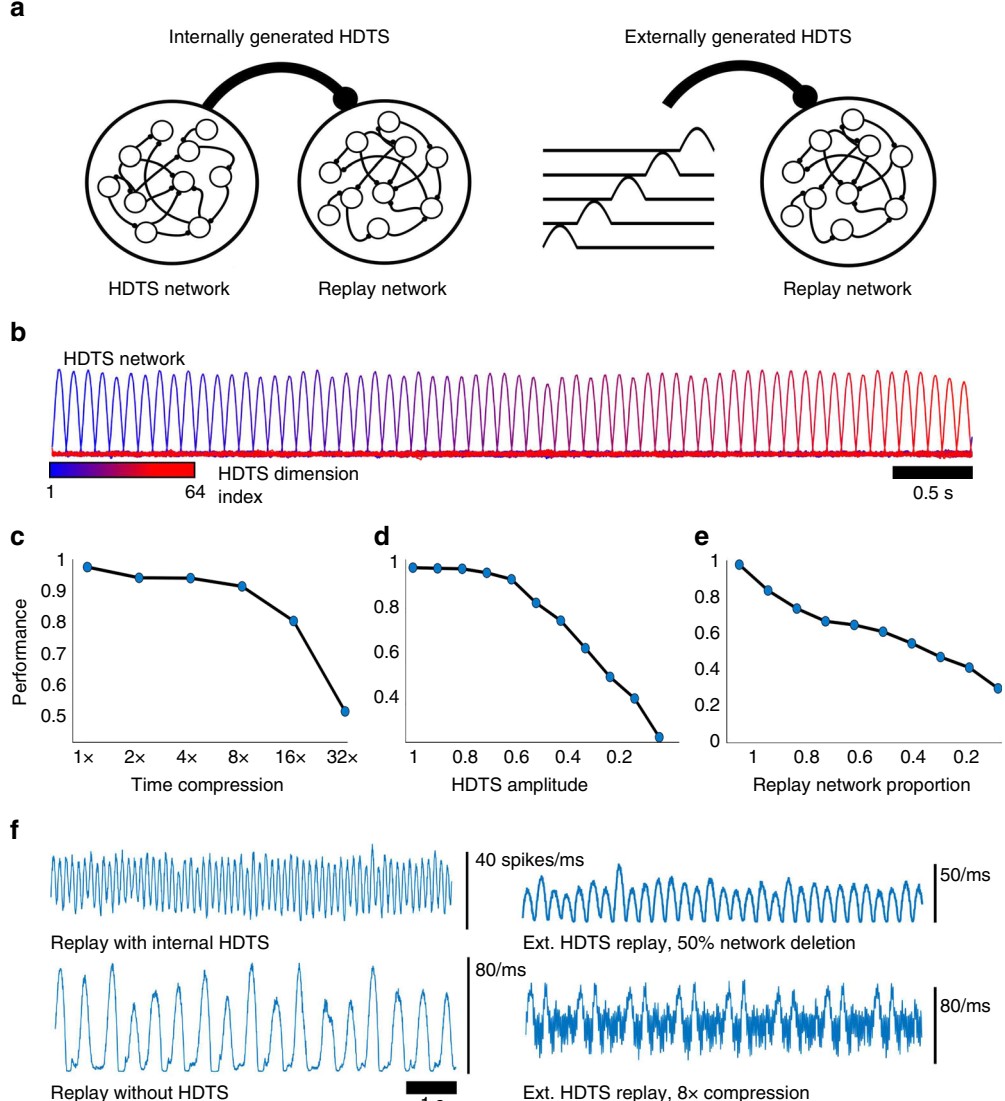

**Fig. 5** Using spiking neural networks for pattern storage and replay: movie replay. **a** Two types of networks were trained to replay an 8 s clip from a movie. In the internally generated HDTS case, both an HDTS and a replay network are simultaneously trained. The HDTS projects onto the replay network similar to how HVC neurons project onto RA neurons in the birdsong circuit. The HDTS confers an 8 Hz oscillation in the mean population activity. In the externally generated HDTS case, the network receives an HDTS supervisor that has not been trained, but is simple to manipulate and confers a 4 Hz oscillation in the mean population activity. **b** The HDTS consists of 64 pulses generated from the positive component of a sinusoidal oscillator with a period of 250 ms. The color of the pulses denotes its order in the temporal chain or equivalently, its dimension in the HDTS. **c** The external HDTS is compressed in time which results in speeding up the replay of the movie clip. The time-averaged correlation coefficient between the teaching signal and network output is used to measure performance. **d** The HDTS amplitude for the network with an internal HDTS was reduced. The network was robust to decreasing the amplitude of the HDTS. **e** Neurons in the replay network were removed and replay performance was measured. The replay performance decreases in an approximately linear fashion with the proportion of the replay network that is removed. **f** The mean population activity for the replay networks under HDTS compression, removal, and replay network lesioning

We were surprised to see that despite the complexity and high dimensionality of the encoding signal, the histogram of spike times across the replay network displayed a strong 4 Hz modulation conferred by the HDTS. This histogram can be interpreted as the mean network activity. Unsurprisingly, reducing the HDTS amplitude yields a steady decrease in replay performance (Fig. 5d). Initially, this is mirrored through a decrease in the amplitude of the 4 Hz oscillations in the mean population activity. Surprisingly however if we remove the HDTS, the mean activity displays a strong slower oscillation (≈2 Hz) (Fig. 5f). The emergence of these slower oscillations corresponds to a sharp decline in replay performance as the scenes are no longer replayed in chronological order (Fig. 5f). The spikes were

also non-uniformly distributed with respect to the mean-activity (Supplementary Fig. 19).

We wanted to determine how removing neurons in the replay network would affect both the mean population activity, and replay performance. We found that the replay performance decreased approximately linearly with the proportion of neurons that were lesioned, with the amplitude of the mean activity also decreasing (Fig. 5e, f). The HDTS network however was much more sensitive to neural lesioning. We found lesioning 10% of a random selection of neurons in the HDTS network was sufficient to stop the HDTS output. Thus, the HDTS network is the critical element in this circuit that can drive accurate replay and is the most sensitive to damaging perturbations such as neural loss.

Finally, we wondered how the frequency and amplitude of the HDTS (and thus the mean activity) would alter the learning accuracy of a network (Supplementary Fig. 18). There was an optimal input frequency located in the 8–16 Hz range for large regions of parameter space. This was robust to different neuronal parameters (Supplementary Fig. 18).

It has been speculated that compressed or reversed replay of an event might be important for memory consolidation[40,41]. Thus we wondered if networks trained with an HDTS could replay the scene in accelerated time by compressing the HDTS post-training. After training, we compressed the external HDTS in time (Fig. 5c). The network was able to replay the movie in compressed time (correlation of $r > 0.8$) up to a compression factor of 16× with accuracy sharply dropping for further compression. The effect of time compression on the mean activity was to introduce higher frequency oscillations (Supplementary Fig. 20, Fig. 5f). The frequency of these oscillations scaled linearly with the amount of compression. However, with increasing compression frequency, large waves of synchronized activity also emerged in the mean population activity (Supplementary Fig. 20, Fig. 5f). Reverse replay was also was successfully achieved by reversing the order in which the HDTS components were presented to the network (accuracy of $r = 0.90$). This loss in accuracy is due to the fact that temporal dynamics of the network is not reversed within a segment of the HDTS. Thus, the network can generalize to robustly compress or reverse replay, despite not being trained to do these tasks.

Compression of a task dependent sequence of firing has been experimentally found in numerous sources[40,42,43]. For example, in ref. [40], the authors found that a recorded sequence of neuronal cross correlations in rats elicited during a spatial sequence task reappeared in compressed time during sleep. The compression ratios between 5.4 and 8.1 and compressed sequences that were originally ≈10 s down to ≈1.5 s. This is similar to the compression ratios we found for our networks without incurring appreciable error in replay (up to a compression factor of 8–16). Time compression and dilation has also been experimentally found in the striatum[42,44]. Here, the authors found that the neuronal firing sequences for striatal neurons were directly compressed in time[42]. Indeed, we also found that accelerated replay of the movie scene compressed the spiking behavior for neurons in the replay network (Supplementary Fig. 20).

To summarize, an HDTS is necessary for encoding and replay of high dimensional natural stimuli. These movie clips can be thought of as proxies for episodic memories. Compression and reversal of the HDTS allows compressed and reversed replay of the memory proxy. At lower compression ratios, the mean population activity mirrors the HDTS while at higher compression ratios (≥8×), large synchronized events in the mean activity emerge that repeat with each movie replay. The optimal HDTS frequency mostly falls in the 8–16 Hz parameter range.

## Discussion
We have shown that FORCE training can take initially chaotic networks of spiking neurons and use them to mimic the natural tasks and functions demonstrated by populations of neurons. For example, these networks were trained to learn low-dimensional dynamical systems, such as oscillators which are at the heart of generating both rhythmic and non rhythmic motion[45]. We found FORCE training to be robust to the spiking model employed, initial network states, and synaptic connection types.

Additionally, we showed that we could train spiking networks to display behaviors beyond low-dimensional dynamics by altering the supervisor used to train the network. For example, we trained a statistical classifier with a network of Izhikevich neurons

that could discriminate its inputs. Extending the notion of an oscillator even further allowed us to store a complicated sequence in the form of the notes of a song, reproduce the singing behavior of songbirds, and encode and replay a movie scene. These tasks are aided by the inclusion of a high-dimensional temporal signal (HDTS) that discretizes time by segregating the neurons into assemblies.

FORCE training is reminiscent of how songbirds learn their stereotypical learned songs[35,46]. Juvenile songbirds are typically presented with a species specific song or repertoire of songs from their parents or other members of their species. These birds internalize the original template song and subsequently use it as an error signal for their own vocalization[35–37,39,46–49]. Our model reproduced the singing behavior of songbirds with FORCE training as the error correction mechanism. Both the spiking statistics of area RA and the song spectrogram were accurately reproduced after FORCE training. Furthermore, we demonstrated that altering the balance between excitation and inhibition post training degrades the singing behavior post-training. A shift to excess excitation alters the spectrogram in a highly non-linear way while a shift to excess inhibition reduces the amplitude of all frequencies.

Inspired by the clock-like input pattern that songbirds use for learning and replay[35,36] we used a similar HDTS to encode a longer and more complex sequence of notes in addition to a scene from a movie. We found that these signals made FORCE training faster and the subsequent replay more accurate. Furthermore, by manipulating the HDTS frequency we found that we could speed up or reverse movie replay in a robust fashion. We found that compressing replay resulted in higher frequency oscillations in the mean population activity. Attenuating the HDTS decreased replay performance while transitioning the mean activity from a 4–8 Hz oscillation to a slower (≈2 Hz) oscillation. Finally, replay of the movie was robust to lesioning neurons in the replay network.

While our episodic memory network was not associated with any particular hippocampal region, it is tempting to conjecture on how our results might be interpreted within the context of the hippocampal literature. In particular, we found that the HDTS conferred a slow oscillation in the mean population activity reminiscent of the slow theta oscillations observed in the hippocampus. The theta oscillation is strongly associated to memory; however, its computational role is not fully understood, with many theories proposed[50–53]. For example, the theta oscillation has been proposed to serve as a clock for memory formation[50,54].

Here, we show a concrete example that natural stimuli that serve as proxies for memories can be bound to an underlying oscillation in a population of neurons. The oscillation forces the neurons to fire in discrete temporal assemblies. The oscillation (via the HDTS) can be sped up, or even reversed resulting in an identical manipulation of the memory. Additionally, we found that reducing the HDTS input severely disrupted replay and the underlying mean population oscillation. This mirrors experimental results that showed that theta power was predictive of correct replay[55]. Furthermore, blocking the HDTS prevents learning and prevents accurate replay with networks trained with an HDTS present. Blocking the hippocampal theta oscillation pharmacologically[56] or optogenetically[57] has also been found to disrupt learning.

The role of the HDTS is reminiscent of the recent discovery of time cells, which also serve to partition themselves across a time interval in episodic memory tasks[58–60]. How time cells are formed is ongoing research however they are dependent on the medial septum, and thus the hippocampal theta oscillation[61]. Time cells have been found in CA1[58], CA3[62] and temporally selective cells occur in the entorhinal cortex[63].

In a broader context, FORCE trained networks could be used in the future to elucidate hippocampal functions. For example, future FORCE trained networks can make use of biological constraints such as Dale's law in an effort to reproduce verified spike distributions for different neuron types with regards to the phase of the theta oscillation[64]. These networks can also be explicitly constructed to represent the different components of the well studied hippocampal circuit.

FORCE training is a powerful tool that allows one to use any sufficiently complicated dynamical system as a basis for universal computation. The primary difficulty in implementing the technique in spiking networks appears to be controlling the orders of magnitude between the chaos inducing weight matrix and the feedback weight matrix. If the chaotic weight matrix is too large in magnitude (via the $G$ parameter), the chaos can no longer be controlled by the feedback weight matrix[1]. However, if the chaos inducing matrix is too weak, the chaotic system no longer functions as a suitable reservoir. To resolve this, we derived a scaling argument for how $Q$ should scale with $G$ for successful training based on network behaviors observed in ref. [1]. Interestingly, the balance between these fluctuations could be related to the fading memory property, a necessary criterion for the convergence of FORCE trained rate networks[65].

Furthermore, while we succeeded in implementing the technique in other neuron types, the Izhikevich model was the most accurate in terms of learning arbitrary tasks or dynamics. This is due to the presence of spike frequency adaptation variables that operate on a much slower time scale than the neuronal equations. There may be other biologically relevant forces that can increase the capacity of the network to act as a reservoir through longer time scale dynamics, such as synaptic depression and NMDA mediated currents for example[66–68].

Furthermore, we found that the inclusion of a high-dimensional temporal signal increased the accuracy and capability of a spiking network to reproduce long signals. In ref. [2], another type of high-dimensional supervisor is used to train initially chaotic spiking networks. Here, the authors use a supervisor consisting of $O(N^2)$ components (see ref. [2] for more details). This is different from our approach involving the construction of an HDTS, which serves to partition the neurons into assemblies and is of lower dimensionality than $O(N^2)$. However, from ref. [2] and our work here, increasing the dimensionality of the supervisor does aid FORCE training accuracy and capability. Finally, it is possible that an HDTS would facilitate faster and more accurate learning in networks of rate equations and more general reservoir methods as well.

Although FORCE trained networks have dynamics that are starting to resemble those of populations of neurons, at present all top-down procedures used to construct any functional spiking neural network need further work to become biologically plausible learning rules[1,5,8]. For example, FORCE trained networks require non-local information in the form of the correlation matrix $\boldsymbol{P}(t)$. However, we should not dismiss the final weight matrices generated by these techniques as biologically implausible simply because the techniques are themselves biologically implausible.

Aside from the original rate formulation in ref. [1], FORCE trained rate equations have been recently applied to analyzing and reproducing experimental data. For example, in ref. [69], the authors used a variant of FORCE training (referred to as Partial In-Network Training, PINning) to train a rate network to reproduce a temporal sequence of activity from mouse calcium imaging data. PINning uses minimal changes from a balanced weight matrix architecture to form neuronal sequences. In ref. [70], the authors combine experimental manipulations with FORCE trained networks to demonstrate that preparatory activity prior to motor behavior is resistant to unilateral perturbations both experimentally, and in their FORCE trained rate models. In ref. [71], the authors demonstrate the dynamics of reservoirs can explain the emergence of mixed selectivity in primate dorsal Anterior Cingulate Cortex (dACC). The authors use a modified version of FORCE training to implement an exploration/exploitation task that was also experimentally performed on primates. The authors found that the FORCE trained neurons had a similar dynamic form of mixed selective as experimentally recorded neurons in the dACC. Finally, in ref. [72], the authors train a network of rate neurons to encode time on the scale of seconds. This network is subsequently used to learn different spatio-temporal tasks, such as a cursive writing task. These FORCE trained networks were able to account for psychophysical results such as Weber's law, where the variance of a response scales like the square of the time since the start of the response. In all cases, FORCE trained rate networks were able to account for and predict experimental findings. Thus, FORCE trained spiking networks can prove to be invaluable for generating novel predictions using voltage traces, spike times, and neuronal parameters.

Top-down network training techniques have different strengths and uses. For example, the Neural Engineering Framework (NEF) and spike-based coding approaches solve for the underlying weight matrices immediately without training[5,6,8,9,11]. The solutions can be analytical as in the spike based coding approach, or numerical, as in the NEF approach. Furthermore, the weight matrix solutions are valid over entire regions of the phase space, where as FORCE training uses individual trajectories as supervisors. Multiple trajectories have to be FORCE trained into a single network to yield a comparable level of global performance over a region. Both sets of solutions yield different insights into the structure, dynamics, and functions of spiking neural networks. For example, brain scale functional models can be constructed with NEF networks[8]. Spike-based coding networks demonstrate how higher order error scaling is possible by utilizing spiking sparsely and efficiently through balanced network solutions. While the NEF and spike based coding approaches provide immediate weight matrix solutions, both techniques are difficult to generalize to other types of networks or other types of tasks. Both the NEF and spike based coding approach require a system of closed form differential equations to determine the static weight matrix that yields the target dynamics.

In summary, we showed that FORCE can be used to train spiking neural networks to reproduce complex spatio-temporal dynamics. This method could be used in the future to mechanically link neural activity to the complex behaviors of animals.

## Methods

**Rate equations**. The network of rate equations is given by the following:

$$\tau_s \dot{s}_i = -s_i + G \sum_{j=1}^{N} \omega_{ij}^0 r_j + Q \eta_i \hat{x}. \tag{1}$$

$$r_j = \begin{pmatrix} F\sqrt{s_j} & s_j \geq 0 \\ 0 & s_j < 0 \end{pmatrix}. \tag{2}$$

$$\hat{x}(t) = \sum_{j=1}^{N} \phi_j r_j(t), \tag{3}$$

where $\omega_{ij}^0$ is the sparse and static weight matrix that induces chaos in the network by having $\omega_{ij}^0$ drawn from a normal distribution with mean 0 and variance $(Np^2)^{-1}$, where $p$ is the degree of sparsity in the network. The variables $s$ can be thought of as neuronal currents with a postsynaptic filtering time constant of $\tau_s$. The encoding variables $\eta_i$ are drawn from a uniform distribution over $[-1, 1]$ and set the neurons encoding preference over the phase space of $x(t)$, the target dynamics. The quantities $G$ and $Q$ are constants that scale the static weight matrix and feedback term, respectively. The firing rates are given by $r_j$ and correspond to the type-I

normal form for firing. The variable $F$ scales the firing rates. The decoders, $\phi_j$ are determined by the recursive least mean squares (RLS) technique, iteratively[29,30]. RLS is described in greater detail in the next section. For Supplementary Figs. 8, 9, we also implemented the $\tanh(x)$ continuous variable equations from[1] for comparison purposes. These equations are described in greater detail in ref. [1].

**Integrate-and-fire networks**. Our networks consist of coupled integrate-and-fire neurons, that are one of the following three forms:

$$\dot{\theta}_i = (1 - \cos(\theta_i)) + \pi^2(1 + \cos(\theta_i))(I) \quad \text{(Theta neuron).} \tag{4}$$

$$\tau_m \dot{v}_i = -v_i + I \quad \text{(LIF neuron).} \tag{5}$$

$$C\dot{v}_i = k(v_i - v_r)(v_i - v_t) - u_i + I \quad \text{(Izhikevich Neuron).} \tag{6}$$

$$\dot{u}_i = a(b(v_i - v_r) - u_i). \tag{7}$$

The currents, $I$ are given by $I = I_{Bias} + s_i$, where $I_{Bias}$ is a constant background current set near or at the rheobase (threshold to spiking) value. The currents are dimensionless in the case of the theta neuron while dimensional for the LIF and Izhikevich models. Note that we have absorbed a unit resistance into the current for the LIF model. The quantities $\theta_i$, or $v_i$ are the voltage variables while $u_i$ is the adaptation current for the Izhikevich model. These voltage variables are instantly set to a reset potential ($v_{reset}$) when a neurons membrane potential reaches a threshold ($v_{thr}$, LIF) or voltage peak ($v_{peak}$, theta model, Izhikevich model). The parameters for the neuron models can be found in Table 1. The parameters from the Izhikevich model are from ref. [73] with a slight modification to the $k$ parameter. The LIF model has a refractory time period, $\tau_{ref}$ where the neuron cannot spike. The adaptation current, $u_i$ for the Izhikevich model increases by a discrete amount $d$ every time neuron $i$ fires a spike and serves to slow down the firing rate. The membrane time constant for the LIF model is given by $\tau_m$. The variables for the Izhikevich model include $C$ (membrane capacitance), $v_r$ (resting membrane potential), $v_t$ (membrane threshold potential), $a$ (reciprocal of the adaptation time constant), $k$ (controls action potential half-width) and $b$ (controls resonance properties of the model).

The spikes are filtered by specific synapse types. For example, the single exponential synaptic filter is given by the following:

$$\dot{r}_j = -\frac{r_j}{\tau_s} + \frac{1}{\tau_s} \sum_{t_{jk} < t} \delta(t - t_{jk}), \tag{8}$$

where $\tau_s$ is the synaptic time constant for the filter and $t_{jk}$ is the $k$th spike fired by the $j$th neuron. The double exponential filter is given by

$$\dot{r}_j = -\frac{r_j}{\tau_d} + h_j. \tag{9}$$

$$\dot{h}_j = -\frac{h_j}{\tau_r} + \frac{1}{\tau_r \tau_d} \sum_{t_{jk} < t} \delta(t - t_{jk}), \tag{10}$$

where $\tau_r$ is the synaptic rise time, and $\tau_d$ is the synaptic decay time. The filters can be of the simple exponential type, double exponential, or alpha synapse type ($\tau_r = \tau_d$). However, we will primarily consider the double exponential filter with a rise time of 2 ms and a decay time of 20 ms. Longer and shorter filters are considered in the Supplementary Materials. Note that the the delta functions are carrying units of pA and thus the $r(t)$ have units of current.

The synaptic currents, $s_i(t)$, are given by the equation:

$$s_i = \sum_{j=1}^{N} \omega_{ij} r_j \tag{11}$$

The matrix $\omega_{ij}$ is the matrix of synaptic connections and controls the magnitude of the postsynaptic currents arriving at neuron $i$ from neuron $j$.

The primary goal of the network is to approximate the dynamics of an $m$-dimensional teaching signal, $x(t)$, with the following approximant:

$$\hat{x}(t) = \sum_{j=1}^{N} \phi_j r_j, \tag{12}$$

where $\phi_j$ is a quantity that is commonly referred to as the linear decoder for the firing rate. The FORCE method decomposes the weight matrix into a static component, and a learned decoder $\phi$:

$$\omega_{ij} = G\omega_{ij}^0 + Q\eta_i \cdot \phi_j^T \tag{13}$$

The sparse matrix $\omega_{ij}^0$ is the static weight matrix that induces chaos. It is drawn from a normal distribution with mean 0, and variance $(Np)^{-1}$, where $p$ is the sparsity of the matrix. Additionally, the sample mean for these weight matrices was explicitly set to 0 for the LIF and theta neuron models to counterbalance the resulting firing rate heterogeneity. This was not necessary with the Izhikevich model. The variable $G$ controls the chaotic behavior and its value varies from neuron model to neuron model. The encoding variables, $\eta_i$ are drawn randomly and uniformly from $[-1, 1]^k$, where $k$ is the dimensionality of the teaching signal. The encoders, $\eta_i$, contribute to the tuning preferences of the neurons in the network. The variable $Q$ is increased to better allow the desired recurrent dynamics to tame the chaos present in the reservoir. The weights are determined dynamically through time by minimizing the squared error between the approximant and intended dynamics, $e(t) = \hat{x}(t) - x(t)$. The RLS technique updates the decoders accordingly:

$$\phi(t) = \phi(t - \Delta t) - e(t)P(t)r(t) \tag{14}$$

$$P(t) = P(t - \Delta t) - \frac{P(t - \Delta t)r(t)r(t)^T P(t - \Delta t)}{1 + r(t)^T P(t - \Delta t)r(t)}, \tag{15}$$

where $P(t)$ is the network estimate for the inverse of the correlation matrix:

$$P(t)^{-1} = \int_0^t r(t)r(t)^T \, dt + \lambda I_N. \tag{16}$$

The parameter $\lambda$ acts both as a regularization parameter[29] and controls the rate of the error reduction while $I_N$ is an $N \times N$ identity matrix. The RLS technique can be seen to minimize the squared error between the target dynamics and the network approximant:

$$C = \int_0^T (\hat{x}(t) - x(t))^2 dt + \lambda \phi^T \phi \tag{17}$$

and uses network generated correlations optimally by using the correlation matrix $P(t)$. The network is initialized with $\phi(0) = 0$, $P(0) = I_N \lambda^{-1}$, where $I_N$ is an $N$-dimensional identity matrix. Note that in the original implementation of ref. [1], Eq. (14) contains the term $1 + r(t)^T P(t - \Delta t)r(t)$ in the denominator. This term adds a slight increase in accuracy and stability, but does not change the overall order of convergence of RLS. For comparison purposes, this modification was applied to Supplementary Fig. 8. All other implementations used Eqs. 14, 15.

**G and Q parameters**. In ref. [1], the learned recurrent component and the static feedback term were both of the same order of magnitude, $O(1)$. Furthermore, if the input stimulus had an amplitude that was too small (ref. [1]), the chaotic dynamics could not be tamed. Operating on the hypothesis that the static term and the learned term require fluctuations of the same order of magnitude, one can derive the following equations using standard arguments about balanced networks:[74]

$$s_i(t) = \sum_{j=1}^{N} G\omega_{ij}^0 r_j(t). \tag{18}$$

$$\langle s_i(t) \rangle \sim 0, N \to \infty. \tag{19}$$

$$\langle s_i(t)^2 \rangle \sim G^2 E\left(\left(\overline{\omega}_{ij}^0\right)^2\right)\langle r_i(t)^2 \rangle, \tag{20}$$

where $\overline{\omega}_{ij}^0 = \sqrt{N}\omega_{ij}^0$ the second term can be derived as follows:

$$s_i(t)^2 = \frac{G^2}{N}\left(\sum_{j=1}^{N}\left(\overline{\omega}_{ij}^0\right)^2 r_j(t)^2 + \sum_{j \neq k} b\overline{\omega}_{ij}^0 \overline{\omega}_{ik}^0 r_j(t)r_k(t)\right) \tag{21}$$

$$\sim G^2 E\left(\left(\overline{\omega}_{ij}^0\right)^2\right)\langle r_j(t)^2 \rangle N \to \infty, \tag{22}$$

where the last step is justified as we can ignore interneuronal correlations $r_i(t)r_j(t)$ in addition to correlations between a weight matrix component and a firing rate, $r_j(t)$ in the large network limit. If the term $\eta \cdot x(t)$ is $O(1)$, this implies that $Q = O\left(G\sigma_\omega \sqrt{\langle r_j(t)^2 \rangle}\right)$, where $\sigma_\omega$ is the standard deviation of the zero mean, static weight matrix distribution.

**High-dimensional temporal signals**. The HDTS serves to stabilize network dynamics by organizing the neurons into assemblies. These assemblies fire at precise intervals of time. To elaborate further, these signals are constructed as a discretization of time, $T$ into $m$ sub-intervals. Each subinterval generates an extra component (dimension) in the supervisor and within the subinterval, a pulse or upward deflection of the supervisor is generated. For example, if we consider the interval from $[0, T]$ with $m$ subintervals of width $I_n = \left[T\left(\frac{n-1}{m}\right), T\left(\frac{n}{m}\right)\right], n = 1, 2, \dots m$. The supervisor is constructed as a series of pulses centered in the intervals $I_n$.

This can be done in multiple ways. For example, Gaussian pulses can be used to construct the HDTS:

$$x_n(t) = \exp\left(-\frac{\left(t - T\left(\frac{2n-1}{2m}\right)\right)^2}{\sigma^2}\right), \quad (23)$$

where $\sigma$ controls the width. Alternatively, one can also use piecewise defined functions such as

$$x_n(t) = \begin{pmatrix} \left|\sin\left(\frac{m\pi t}{T}\right)\right| & t \in I_n \\ 0 & \text{otherwise.} \end{pmatrix} \quad (24)$$

The end effect of these signals is to divide the network into a series of assemblies. A neurons participation in the $n$th assembly is a function of the static weight matrix, in addition to $\eta_{\text{in}}$. Each assembly is activated by the preceding assemblies to propagate a long signal through the network. The assemblies are activated at a specific frequency in the network, dictated by the width of the pulses in the HDTS. Sinusoidal HDTS were used for the all figures with the exception of Supplementary Fig. 18c, where a Gaussian HDTS was used.

**Figure 1 methods.** A network of rate equations is used to learn the dynamics of a teaching signal, $x(t)$, a 5 Hz sinusoidal oscillation. The network consists of 1000 rate neurons with $F = 10$, $\tau_s = 10$ ms, $G = 1$, and $Q = 1.5$, $p = 0.1$. The network is run for 1 s initially, with RLS run for the next 4 s, and shut off at the 5 s mark. The network is subsequently run for another 5 s to ensure learning occurs. The smooth firing rates of the network were less than 30 Hz. The RLS parameters were taken to be $\Delta t = 2$ ms and $\lambda^{-1} = 0.5$ ms.

**Figure 2 methods.** Networks of theta neurons were trained to reproduce different oscillators. The parameters were: $N = 2000$, $G = 10$ (sinusoid, sawtooth), $G = 15$ (Van der Pol), $G = 25$ (product of sinusoids), $G = 15$ (product of sinusoids with noise), $Q = 10^4$, $\Delta t = 0.5$ ms, with an integration time step of 0.01 ms. The resulting average firing rates for the network were 26.1 Hz (sinusoid) 29.0 Hz (sawtooth), 15.0 Hz (Van der Pol relaxation), 18.8 Hz (Van der Pol Harmonic), 23.0 Hz (product of sinusoids), 21.1 Hz (product of sines with noise). The Van der Pol oscillator is given by: $\ddot{x} = \mu(1 - x^2)\dot{x} - x$, for $\mu = 0.3$ (harmonic) and $\mu = 5$ (relaxation) and was rescaled in space to lie within $[-1, 1]^2$ and sped up in time by a factor of 20. The networks were initialized with the static matrix with 10% connectivity. This induces chaotic spiking in the network. We allow the network to settle onto the chaotic attractor for a period of 5 s. After this initial period, RLS is activated to turn on FORCE training for a duration of 5 s. After this period of time, RLS was deactivated for the remainder 5 s of simulation time for all teaching signals except signals that were the product of sinusoids. These signals required 50 s of training time and were tested for a longer duration post training to determine convergence. The $\lambda^{-1}$ parameter used for RLS was taken to be the integration time step, 0.01 ms for all implementations of the theta model in Fig. 2. The integration time step used in all cases of the theta model was 0.01 ms.

The theta neuron parameters were as in Fig. 2b for the 5 Hz sinusoidal oscillator. The LIF network consisted of 2000 neurons with $G = 0.04$, $Q = 10$, with 10% connectivity in the static weight matrix and $\Delta t = 2.5$ ms with an integration time step of 0.5 ms. The average firing rate was 22.9 Hz. The Izhikevich network consisted of 2000 neurons with $G = 5 \times 10^3$, $Q = 5 \times 10^3$, with 10% connectivity in the static weight matrix and $\Delta t = 0.8$ ms and an average firing rate of 36.7 Hz. The $\lambda^{-1}$ parameter used for RLS was taken to be a small fraction of the integration time step, $\lambda^{-1} = 0.0025$ ms for Fig. 2 while the Izhikevich model had $\lambda^{-1} = 1$ ms and an integration time step of 0.04 ms. The training time for all networks considered was 4 s, followed by 5 s of testing.

The Lorenz system with the parameter $\rho = 28$, $\sigma = 10$, and $B = 8/3$ was used to train a network of 5000 theta neurons and is given by the equations:

$$\dot{x} = \sigma(y - x)$$

$$\dot{y} = x(\rho - z) - y$$

$$\dot{z} = xy - Bz.$$

The connectivity parameters were $G = 14$, and $Q = 10^4$. The average firing rate was 22.21 Hz. The network was trained for 45 s, and reproduced Lorenz-like trajectories and the attractor for the remaining 50 s.

**Figure 3 methods.** The teaching signal was constructed using the positive component for a sinusoid with a frequency of 2 Hz for quarter notes and 1 Hz for half notes. Each pulse corresponds to the presence of a note and they can also be used as the amplitude/envelope of the harmonics corresponding to each note to generate an audio-signal from the network (Supplementary Audio 1). The network consisted of 5000 Izhikevich neurons with identical parameters as before, $G = 1 \times 10^4$,

$Q = 4 \times 10^3$, and $\Delta t = 4$ ms and an integration time step of 0.04 ms. The teaching signal was continually fed into the network for 900 s, corresponding to 225 repetitions of the signal. After training, the network was simulated with RLS off for a period of 1000 s. In this interval, correct replay of the song corresponded to 82% of the duration with 205 distinct, correct replays of the song within the signal. Correct replays were automatically classified by constructing a moving average error function:

$$E(t) = \sum_{i=1}^{5} \int_t^{t+T} \left(\hat{x}_i(t') - x_i(t')\right)^2 dt', \quad (25)$$

where $T = 4$ seconds is the length of the song. As we vary $t$, the teaching signal $x(t')$ comes into alignment with correct playbacks in the network output, $\hat{x}(t')$. This reduces the error creating local minima of $E(t)$ that correspond to correct replays at times $t = t^*$. These are automatically classified as correct if $E(t^*)$ is under a critical value. The average firing rate for the network was 34 Hz. $\lambda^{-1} = 2$ ms was used for RLS training.

**Figure 4 methods.** The teaching signal consisted of the spectrogram for a 5 s long recording of the singing behavior of a male zebra finch that was generated with a 22.7 ms window. This data were obtained from the CRCNS data repository[75]. The RA network consisted of 1000 neurons that was FORCE trained for 50 s with $G = 1.3 \times 10^4$, $Q = 1 \times 10^3$, $\lambda^{-1} = 2$ ms, $\Delta t = 4$ ms and an integration time step of 0.04 ms. The average firing rate for the network was 52.93 Hz. The HVC input was modeled as a sequence of 500 pulses that were constructed from the positive component of a sinusoidal oscillation with a period of 20 ms. The pulses fired in a chain starting with the first component. This chain was multiplied by a static, $N \times 500$ feedforward weight matrix, $W^{\text{in}}$ where each weight was drawn uniformly from the $[-8 \times 10^3, 8 \times 10^3]$. The network was tested for another 50 s after RLS was turned off to test if learning was successful. We note that for this songbird example, the network could also learn the output with only the static, chaos inducing weight matrix and the feedforward HVC inputs ($Q = 0$). However when $Q = 0$, the network behavior is similar to the more classic liquid state regime.

To manipulate the weight matrix post training, we introduced a parameter $\alpha$ that allowed us to control the balance between excitation and inhibition:

$$\omega_{ij} = \alpha\left(G\omega_{ij}^0 + Q\eta_i \cdot \phi_j\right)_+ + \left(G\omega_{ij}^0 + Q\eta_i \cdot \phi_j\right)_-, \quad (26)$$

where $(x)_\pm$ denotes:

$$(x)_\pm = \begin{pmatrix} x & x > (<)0 \\ 0 & x \leq (\geq)0 \end{pmatrix}.$$

The values $\alpha = 1$ yield the original weight matrix, while $\alpha > 1$ amplifies the excitatory connections and $\alpha < 1$ diminishes the excitatory connections.

**Figure 5 methods.** The teaching signal consisted of an 8 s movie clip (see [77]). The frames were smoothed and interpolated so that RLS could be applied at any time point, instead of just the fixed times corresponding to the native frames per second of the original movie clip. The movie was downsampled to 1920 pixels ($30 \times 64$) which formed the supervisor. For both implementations (external and internal HDTS), the replay network consisted of 1000 neurons with $G = 5 \times 10^3$, $Q = 4 \times 10^2$, $\lambda^{-1} = 2$ ms, $\Delta t = 4$ ms and an integration time step of 0.04 ms. For the external HDTS network, an HDTS was generated with 32 pulses that were 250 ms in duration. The HDTS was multiplied by a static, $N \times 32$ feedforward weight matrix, $W^{\text{in}}$ where each weight was drawn uniformly from $[-4 \times 10^3, 4 \times 10^3]$ and 74 s of FORCE training was used. The network was simulated for another 370 s after RLS was turned off to test if learning was successful. For the internal HDTS case, a separate HDTS network was trained with 2000 neurons. The supervisor for this network consisted of a 64-dimensional HDTS with pulses that were 125 ms in duration. The HDTS was fed into an identical replay network as in the external HDTS case. The replay network was trained simultaneously to the HDTS network and had $G = 1 \times 10^4$, $Q = 4 \times 10^3$. The HDTS was fed into the replay network in an identical manner to the external HDTS case. RLS was applied to train the HDTS and replay network with identical parameters as in the external HDTS case, only with a longer training time (290 s). We note that for this movie example, the network could also learn the output with only the static, chaos inducing weight matrix and the feedforward HDTS inputs ($Q = 0$). However when $Q = 0$, the network behavior is similar to the more classic liquid state regime.

**Data availability.** The code used for this paper can be found on modelDB[31], under accession number 190565.

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

## Acknowledgements

This work was funded by the Canadian National Sciences and Engineering Research Council (NSERC) Post-doctoral Fellowship, by the Wellcome Trust (200790/Z/16/Z), the Leverhulme Trust (RPG-2015-171), and the BBSRC (BB/N013956/1 and BB/N019008/1). We would like to thank Frances Skinner, Chris Eliasmith, Larry Abbott, Raoul-Martin Memmesheimer, Brian DePasquale, and Dean Buonomano for their comments. Finally, we would like to especially thank the anonymous referees. Their comments and suggestions greatly improved this manuscript.

## Author contributions

W.N. performed wrote software and performed simulations. Investigation and analysis was performed by W.N. and C.C. W.N. and C.C. wrote the manuscript.

## Additional information

**Competing interests:** The authors declare no competing financial interests.

