## [Peer Review File · Nature Communications]

Reviewers' comments:

Reviewer #1 (Remarks to the Author):

Spiking FORCE Review

I commend the authors for their innovations, and also for what appears to be a mountain of work. I found much of the work to be convincing and intriguing. As far as I understand the current literature, the claims made in this paper to train spiking RNNs directly with FORCE are novel, and nontrivial. I know there have been a few attempts in the community, but generally speaking, people have viewed the direct application of FORCE to spiking nets as "difficult" or "impossible." Given the prominence of the FORCE learning rule in the (computational) neuroscience community, and the desire of researchers in that community to study spiking RNNs, the computational findings in this manuscript will receive a wide readership and be cited widely within the community. As the remaining part of the review is pretty negative in one direction, I want to emphasize that I think the computational work around the FORCE learning applied to spiking networks is great, and I endorse the publication of that work in some form.

Having said all of that, this manuscript is really a tale of two stories. This first story is a demonstration that FORCE learning (Sussillo & Abbott, 2009) can be essentially directly applied to biologically inspired spiking neural network models (especially the Izhikevich model). The second story attempts to provide high-level comparisons of trained spiking networks to various brain circuits. The first story is ready (with revision) for publication, while the second is not. Overall, I recommend ditching the superficial comparisons to biology, and exploring further the novel modeling phenomena found in those examples further ("clock" signal and "theta" oscillations), but devoid of the biological links, leaving those for later, more robust studies.

I think an additional contribution would be to release the source code for training these examples in an online repository.

Major concerns involving microbiological miscommunication or misunderstanding:

An example of biological comparison overreach in the intro:

"This demonstrates a well-defined, computational role of theta oscillations in learning and memory". Huh? I do not think these results demonstrate anything of the sort, and is representative of the light-weight approach to biological comparison found throughout the manuscript. At the very best, they provide a hypothesis that could be studied further experimentally and computationally.

Comparison of the models trained to output songbird spectrogram. This entire comparison feels superficial to me. While I accept the idea that perhaps song spectrogram can serve as a proxy for a signal comparable in complexity of the signals that RA emits, I do not think it is a stand-in for anything like the neural signal that RA emits (if I'm wrong, convince me with citations). This renders the entire comparison of the trained network to statistics of RA useless.

Further, whereas physiological computation can be compared between model and neural circuit at an emergent / dynamics level, comparative perturbation and lesion studies rely on the biophysics / connectivity between model and neural circuits to be similar to each other. I just don't see how this can be true between your models and the neural circuits you discuss, so I am deeply suspect of the perturbation studies in this manuscript.

Do neurophysiologists really think of theta as a high-dimensional signal? Are you really arguing that

what is observed in LFP is actually a high-dimensional signal like that found in HVC? Again, convince me I've misunderstood something, but my opinion is the biological connection here is tenuous.

General Remarks on writing -

All in all, this manuscript is pretty confusing, and could stand a pretty significant rewrite. The general thrust of the manuscript is quite clear, but even the mid level details can be quite obscure at times. Adding to this, it appears that the figures have sometimes missing labels, etc.

Up-state vs down-state. If I understand correctly, the entire manuscript is relating to neural network simulations. I would be extremely careful with the use of this terminology (and generally separating biological terminology from modeling terminology). My understanding is that circuit level physiologists have extremely precise meanings for these terms. I suspect that the usage UP/DOWN state in this manuscript will antagonize those readers. I recommend using different terms, but admit that I might be wrong in this example.

Substantive Questions / Minor Criticisms:

It appears that primarily you used Izhikevich neurons. Is this the major reason the simulations work? Please discuss whether or not this is true in the manuscript. For example, could the theta model or the LIF learned the more complex tasks? This is important because it provides a reason to use more complex spiking models. Maass and his colleagues have been using very complex synapses to make their liquid state machines more temporally interesting.

Following from the previous remark, a hallmark of the FORCE method is that it "just works". I would like to see a histogram of the number of successful training attempts for spiking networks, given random initializations in the following form: for all three models train one of the simple sine wave oscillation tasks, and the harder ode-to-joy task. How often does FORCE learning work with spiking networks? Does it "just work"? The reason I ask is because it is well known that a lot of folks have tried FORCE learning directly applied to spiking networks and did not a lot huge success. I am very interested in your explanation for what make FORCE applied to spiking networks so successful in your hands. To me, this should be the key question to answer in a computational paper. Btw, S2 is lovely, but it doesn't get at the question of training robustness for these other model types.

It appears you have worked on the fully connected case. FORCE was also introduced for the sparse case (Sussillo & Abbott 2009), have you tried this? Does it work? (No need for any extra work here, just asking).

I really liked the part about the "clock" driving signal, although I think the name is misleading. I do not think it's fair to call the input to either the birdsong example, or the Ode to Joy example a clock, which I will call a high-dimensional driving signal (HDDS). My guess is that much more important than the exact timing of the driving signal is the dimensionality of it. Namely, that the high dim input drives the reservoir to explore more states, which allows the readout to have more capacity because more weight space is being used. At any rate, calling it a clock signal is incorrect in my opinion and it should be changed (Again please argue if I've missed something).

I do have a substantial criticism here, however. The fact that hand crafted HDDS signal was used also solves a major problem that RNNs suffer from, simply by kicking it down the line. This is the problem of stabilizing long or complex trajectories. If one can train an RNN to generate that HDDS in a stable

manner (as was tried in Fig. 5 before moving to a “hand crafted one, for the rest of the examples), then that is progress, otherwise it is declaring success when there is none. So in my opinion all examples with the hand crafted HDDS should be rerun with the HDDS generated from an actual network (what the authors call the non-autonomous case should be replaced everywhere with what they call the autonomous case). Otherwise, I think it is very misleading.

As a remark on the HDDS, and not a criticism. I want to suggest a thought experiment: I guess that you could actually achieve the same performance if you removed the HDDS as input, and instead encoded the target signal in an analogous way (as it gets fed back to the system). Specifically, taking a particular target signal, chopping it up into T time segments encoded into T different dimensions (say by multiplying by a gaussian bump centered in the middle of the T th section, and then training those segments. So if you had M outputs, now you have $M \times T$ outputs (i.e. the target signal is 0 outside of its assigned segment). The cardinality of the feedback η would also be increased by T , if there were T bumps in the HDDS to begin with. Now because the effective driving signal, η , is different for each T segment, the driving force of the feedback would also explore more of the the full dimensionality of the space, and perhaps work. Worth some thought, but you do not have to do anything here to satisfy me.

Classification Task

I didn't understand what was meant by the upstates correspond to one class while the down states correspond to another. I guess this means that the readout, \hat{x} is correlated?

This entire example was extremely unclear to me, though I understood the gist, which is a spiking network trained with FORCE can classify. It might be helpful to plot the input in 3A, and also label all the axes in the figure.

Ode to Joy

How was the song restarted in “1000 stretch of simulation time after training?”

Birdsong

What is α , is it defined in the methods?

Movie

Same points about clock signal dimensionality above hold for this example as well.

Moving past the dimensionality of the clock signal, I am also concerned about its frequency. Rajan and Abbott (2010?) showed that the removal of chaos in rate networks was dependent on both the amplitude, and the frequency. These frequencies mentioned on this paper are surely a function of the network parameters of the Izhikevich neurons, as are likely the natural periodicities of the chaos inherent in the system. The idea that these particular frequencies have any meaning at all to real neurons is intriguing, yet hardly substantiated here. So using words like θ and δ oscillations is extremely misleading, in my opinion, as they mix terminology from neurobiology and network modeling. If you were to make a detailed analysis, comparing network parameters, and network architectures in depth, sweeping relevant parameters, perhaps this could stand on its own as an additional stand-alone contribution. As it is here, it feels weak, and I recommend removing it.

How does the performance of the clock signal vary as a function of the clock period? Is θ necessary? Why θ ? As some function of the biophysical parameters of the network? I have to

admit, the implied connection here to the hippocampus feels quite tenuous. I think it is interesting of its own right to understand why an abstract network of neurons would benefit from this oscillation. Most critically, one should test the frequency of the clock signal here, as is done for amplitude in Figure 6E.

In the discussion, it's likely worth pointing out why your HDDS is different than the basis set of additional signal in De'Pascale and Abbott. I.e. they also add more signals. If so, it might be meaningful to cite them here.

Having an autonomous recall here really amounts to whether you can train a network to output a HDDS, right? So I don't really think there's much here in terms of highlighting the autonomous recall, in fact the word usage is actively misleading and should be changed, hopefully by removing it, and only using the autonomous case as mentioned earlier.

Line by line comments -

58 - 60 I think there are uses of FORCE that aren't strictly rate based, but are more outside the box. E.g. light reservoirs, as well as biological neuron reservoirs? Not sure, but perhaps should take a look.

170-173 - the spike frequency adaptation has been heavily used by Maass in his spiking liquid state machines. The usage in many of those manuscripts is short-term plasticity ala Tsodyks and Markram. Probably should include this as a citation here. It would be interesting to see FORCE work with that plasticity type as well, though I am not asking you to do it.

RLS vs RLMS - Isn't the technique (eqns 14-16) you are using called recursive least squares, not recursive least mean squares? My understanding is that RLMS is the first-order version that does not involve the inverse correlation matrix (eqn 16), and the learning rate (singular) is chosen heuristically. Btw, it is my strong hunch that with your HDDS, you could actually use the RLMS, and not perhaps need the 2nd order RLS method. Just a guess, no need for you to do anything.

Figure 1D is missing in the caption

Figure 6 Autonomous vs. Nonautonomous

I think these words are misused. My understanding of autonomous is that a dynamical system receives no input whatsoever. That is not the case for either recall network described in figure 6. The left panel of A would be autonomous if the recall network projected back to the clock network. This issue of autonomous or not is relevant because stabilizing these networks is a large part of the challenge.

Reviewer #2 (Remarks to the Author):

In this paper, the authors train spiking neural networks using FORCE on a variety of tasks. Several variants of spiking models are tested, with the Izhikevich slowly adapting neurons performing the best. The main tasks studied are sequence generation, where the authors show the advantage of having an explicit clock signal. Several analogs with experimental results are suggested.

The paper has several interesting results, but I feel that the multitude of results prevents a thorough analysis of any specific one. For instance, what is the main message of the paper? A method for training spiking networks with FORCE, or the importance of clock signals?

Major comments:

1. The paper's title and abstract emphasize the application of FORCE to spiking networks. The introduction also describes the difficulty of using FORCE on spiking networks. Indeed, the authors show that spiking networks can be trained by FORCE, but there is no explanation of why this works in the current study and failed for other groups. Several questions naturally arise that can shed light on this topic:
 - a. How does the size of the network required to achieve a certain performance compare between rate and spiking models?
 - b. Does the slow ($a=0.01$) parameter render this spiking model an effective rate model, for the parameter regimes used in the study?
 - c. The Lorenz attractor was studied in the original Jaeger paper, and in others. How does the performance of the current model compare with previous ones? Specifically, this can be a useful benchmark to assess the effect of spiking vs. rate models.
2. Related to the previous point – what do we gain from using a spiking model? I agree that there is value to being able to train spiking networks. But there can also be added value such as spike timing statistics, spike triggered averages and other measures, that might not be observable in a rate model. These are not explored here.
3. Theta modulation. This is presented as a surprising finding that is consistent with experimental results. But the origin of theta should be determined from the network simulations. Several directions are:
 - a. Is there theta in the input clock signal?
 - b. Does it relate to the single neuron timescales?
 - c. This is an opportunity to take advantage of the spiking models, and analyze the relation between spiking and the phase of theta.
4. The results about compressed and reversed replay seem trivial, but this could be a misunderstanding on my part. The way I understood from the main text and methods, the network essentially acts as a mapping between specific input patterns and short output patterns. There is no relevant internal dynamics for the transition from one note/frame to the next. If this is the case, then the output is completely determined by the clock signal, and there is no need to train the internal dynamics. In fact, this is indeed the case according to the methods (the task also works for $Q=0$). This interpretation is a very natural concern, and should be addressed in the text.
5. There is very little quantitative analysis. How does performance change with network size? With spiking or rate models? With Dale's law?
6. The classification example seems unconnected to the paper. In itself it does not seem very surprising or novel, and I think it only distracts from the other messages of the paper.
7. The procedure for improving the NEF in the supplementary is an interesting result, but I doubt that many readers will notice it.
8. The parameter values vary over very large ranges ($G=1$ vs. $G=13000$). In the rate networks, it is known that $G=1$ is the edge of chaos, and this value has functional importance. For the spiking networks, it would be very helpful to indicate the relation between the chosen parameter values and the onset of chaos.

Minor comments:

1. Line 152 – I think the word "compares" is missing.
2. The size of the networks, their distance from chaos, the dimensionality of the inputs – all of these are in the Methods. Some parameters have important qualitative information, and should appear in the main text.
3. Line 255-257: "... by varying the parameters ... statistics of RA neurons are easily reproduced". I couldn't find in the main text or methods any mention of this procedure. How sensitive are the statistics to the various parameters? Does anything need to be finely tuned?

4. Line 294 “high dimensional” – how many dimensions? The information is in the methods, but I think it is important enough to include here as well.
5. Line 301 “clock inputs were necessary both for training and recall” – this should be quantified.
6. Figure 1, the caption title is “rate”, but the content is spiking.
7. Figure 1D not mentioned in caption
8. Figure 3D caption: “is” appears twice near parantheses.
9. Figure 4C – not clear what is shown here. Are these values of the readout vectors? The lack of Y axis or label does not help.
10. Figure 5C – what are the different colors?
11. Figure 5D caption – could be useful to specify the meaning of $\alpha < 1$ vs > 1 .
12. Figure 5F – the definition of positive/negative is counterintuitive when one looks at the figure and its labels.
13. Figure 5 – is alpha and excitatory gain the same thing? If yes, then using two terms in the same figure is confusing.
14. Figure 6 – the panels are very dense, and it is hard to relate the text to the correct panel (e.g. “Recall network” seems to belong to panel C)
15. Figure S1D – “similarity to FORCE eigenvalues”. In what sense? A reference to a specific figure would be useful.
16. Dale’s law – was it used on the more complex tasks? For instance, how does the error change in the Lorenz attractor when this constraint is introduced?
17. Eigenvalues (like Figure S11D) should have a reference on where is the edge of chaos.
18. The tasks chosen seem to be rather easy compared to previous works using FORCE on rate networks. The fact that several could be solved with $Q=0$ is another indication of this fact.
19. Following the previous comment, I greatly appreciate the mention of the $Q=0$ fact in the methods. Besides saving me a question in the review, I think it is an important service to the readers.
20. It seems that there is no way for the result of Figure 3D to be any other way. If these results tell us something new, then this should be explained better.
21. Many figures lack axes or labels, and the panels are sometimes too close to one another.

Reviewer #3 (Remarks to the Author):

How can we make a general purpose recurrent neural network perform a given computation? As little as ten years ago, it was almost universally believed that recurrent nets are too complicated to train directly in a supervised way. The reservoir computing framework emerged as an alternative in which the recurrent connectivity is fixed and a linear decoder is trained to generate the desired network output. This is possible because a network with complex enough dynamics implements a spatial-temporal equivalent of the kernel trick in machine learning, projecting the inputs in a high dimensional space where problems which were highly nonlinear in the input space become linearly separable. The key here is having ‘complex dynamics’, close to (but not quite) chaotic. Machine learning has built on top of this architecture to the point where we can actually tame chaos and guide it toward desired network behavior. We can now train all connections within these circuits in a supervised way (using algorithms such as FORCE, and related). Moreover, alternative frameworks such as NEF and the distributed spike-based representations from Deneve and collaborators now allow learning with spiking neural networks outside reservoir computing.

The authors present here an alternative way of learning using the original FORCE algorithm but applied to spiking neural networks. They use a vast array of tasks - from toy examples to complex realistic inputs from different domains - to illustrate the power and flexibility of the spiking neural

network as a computational unit. Furthermore, the authors strive to relate the learned networks properties to a wide range of experimental findings (from the birdsong system to the hippocampus).

The numerical results are indeed impressive and I applaud the drive to link the network properties to neural data. Still, I have a range of concerns that have dampened my enthusiasm for the paper:

1) Novelty of the approach: unless I missed something, the training procedure is off-the-shelf FORCE applied to the instantaneous firing rates (low-pass filtered neural outputs). The authors mention in the introduction several past attempts at doing supervised learning in spiking neurons in the FORCE framework - more involved than what is proposed here. I was wondering why do those first and what was the key insight that made learning possible here. More importantly, I was wondering if the spiking nature of the representation actually makes any difference. The authors do mention the expansion of the state space due to having additional slow time constants in the single neuron dynamics (spike frequency adaptation in the Izhikevich model) but the addition of a slow time constant in the rate model would give you the same. None of this is in itself a problem, but the manuscript needs to do a better job at spelling out what exactly is new here - the section dedicated to model comparison in the discussion reads like a history of related approaches (belonging maybe to the introduction) rather than a critical comparison.

Also I find the comparison to NEF and Deneve's work a bit clumsy. All are cast as optimization problems, minimizing L2 reconstruction error for a linear decoder. What differs is the approach taken to find the optimum - either analytically in Deneve and co or numerically for the rest. Each version comes with its advantages and disadvantages. The main benefit of the version considered here is the fact that it allows greater flexibility in terms of the modeling of neurons and synapse dynamics, potentially resulting in more detailed biologically-motivated networks. The distinction between this approach and 'artificial networks' I find misleading - all of them are biologically-inspired artificial nets, some recurrent other not, some rate-based other spike-based, simpler or more complex.

2) Links to biology: while a lot of experimental data is mentioned, I found most of the connections to be very superficial; lots of different observations but very little insight. I was hoping for a better understanding of what is it about the biological substrate that gives rise to the dynamical features seen here. The songbird data is pretty convincing, but the references to hippocampal replay are problematic. One critical issue is emphasizing the emergence of a prominent theta oscillation in the trained network, the amplitude of which correlates with performance. I was wondering why specifically theta - which aspects of the network make this particular power band important (frequency of inputs, of target output, time constants of neurons and synapses etc). Until I dug out the details and found that the period of the clock fed into the network (or trained to reproduce by the secondary clock generating network) is itself theta frequency (4Hz, or 8Hz)... This makes this set of results rather disappointing. There should still be something interesting in figuring out why this particular clock works with this particular input dynamics but since the input statistics are not particularly hippocampal relevant I am not sure what we can actually learn here.

Detailed comments:

1) it is incorrect to say that the spike-based coding technique by Deneve and co "assume specific types of synapses or neurons" (l42-43). In fact the dynamics of synapses and neurons are directly derived from the cost function.

2) what is 'probabilistic/statistical' about the classification task? Also, if the dynamics were trained to

return to rest after each input why is there still a recent history effect on the classification?

3) the increase in variance "when the network fails to classify consistently" seems like a truism to me. The lack of consistency is by definition a fluctuation between up and down states, which of course increases voltage variance.

4) as far as I can tell, what it is investigated in the spatio-temporal pattern learning tasks is spontaneous replay. When people talk about recall -in the context of memory in particular- we tend to think about cued recall, where the network receives the beginning of the sequence and the network completes the rest. it would be important to explicitly clarify the distinction, I think.

5) I found the explanation of the training procedure in I598-601 difficult to follow.

6) Memorizing spatio-temporal patterns: is it the higher dimensionality of the movie task that makes the clock strictly necessary whereas this was not true for the songbird task?

Why is FORCE learning not able to come up with a solution relying on an internal clock directly but rather it always needs a separate subnetwork trained in a supervised way to generate the clock?

7) I may be missing something pretty fundamental here but how can you reverse a clock? I thought this was just an oscillation...

8) the interpretation of the Grossberg data seems dubious to me. At least more explanation is necessary.

9) the idea that hippocampal memory recall operates on top of a 'backbone' of theta oscillations is hardly novel (see Lisman and co). There is something more subtly different about what that means in this particular context, but it needs to be spelled out a lot more clearly.

Typos and co:

l150 then->than (a recurring mistake, check the rest too)

l422 "serve as a backbone" for what?

l620: fo->of

Revisions

Wilten Nicola and Claudia Clopath

June 13, 2017

Referee 1: Major Points

1. I commend the authors for their innovations, and also for what appears to be a mountain of work. I found much of the work to be convincing and intriguing. As far as I understand the current literature, the claims made in this paper to train spiking RNNs directly with FORCE are novel, and nontrivial. I know there have been a few attempts in the community, but generally speaking, people have viewed the direct application of FORCE to spiking nets as “difficult” or “impossible.” Given the prominence of the FORCE learning rule in the (computational) neuroscience community, and the desire of researchers in that community to study spiking RNNs, the computational findings in this manuscript will receive a wide readership and be cited widely within the community. As the remaining part of the review is pretty negative in one direction, I want to emphasize that I think the computational work around the FORCE learning applied to spiking networks is great, and I endorse the publication of that work in some form.

Having said all of that, this manuscript is really a tale of two stories. This first story is a demonstration that FORCE learning (Sussillo & Abbott, 2009) can be essentially directly applied to biologically inspired spiking neural network models (especially the Izhikevich model). The second story attempts to provide high-level comparisons of trained spiking networks to various brain circuits. The first story is ready (with revision) for publication, while the second is not. Overall, I recommend ditching the superficial comparisons to biology, and exploring further the novel modeling phenomena found in those examples further (“clock” signal and “theta” oscillations), but devoid of the biological links, leaving those for later, more robust studies.

We would like to thank referee for their thorough review. With regards to the biological links, we agree that it is difficult to draw direct conclusions to the biology with the networks we have studied. The networks are not sufficiently constrained by the underlying data such that perturbations we have conducted could plausibly yield experimental predictions. However, the editors have requested that we do not remove these examples (songbird, movie). Therefore, we think the optimal approach to satisfy the requests of both the referees and the editors simultaneously would be to highlight these examples not as being necessarily representative of a biological circuit or region (hippocampus, song bird nuclei etc.) but rather demonstrating the potential use of FORCE training to analyze circuits and tasks of similar complexity in future work. This also narrows the main message of the paper to just FORCE training and its potential application.

For example, we have included the following paragraph at the end of the songbird section in the discussion:

“Currently, our FORCE trained spiking networks can replicate singing behavior and allow for perturbative analyses of network function. While this in itself is a significant step forward in the literature, further work remains to be done. For example, the HVC circuit can be modelled as an HDTTS generator network. The predictive power of these models can be increased by constraining these networks both at the neuronal and connectivity level in future work for example by the biophysics of neurons in HVC and RA. The post-trained connectivity matrix could be constrained by using measured connectivity statistics in these regions.”

Further, we make it explicitly clear that our Figure 5 example, the episodic memory task network was not directly related to any hippocampal region, and we merely conjecture on how our results might be interpreted in the context of the hippocampal literature. This is done in the second paragraph of the discussion section “Episodic Memories can be Encoded and Replayed with FORCE Training Using High Dimensional Temporal Signals”

Further, we have implemented the following changes:

- Removed many of the references to the biological system and explicitly stated that examples such as the song-bird circuit should be viewed as demonstrations of potential uses of FORCE training in the future.
- Removed the biological labels of specific frequency ranges such as theta (8-12 Hz), delta (< 2 Hz), etc. and explicitly refer to the numerical values instead.
- Rewritten large parts of the results and discussion to reduce unnecessary references to experimental results for either the song bird circuit or hippocampal functioning.
- Reinforced the results on the HDTs signal, and elaborated how it functions with an added section to the materials and methods “High Dimensional Temporal Signals”.

More specific changes have been implemented and documented in response to specific concerns from the referees (see points below). We hope that these changes will address all their concerns. Finally, we have provided two copies of the revised manuscript; with and without the tracked changes from the previously submitted version.

2. I think an additional contribution would be to release the source code for training these examples in an online repository.

We would like to thank the referee for bringing this up. In the initial submission process, we included a note to the editor with the modelDB accession number and the referee access password. It appears that it was not communicated in the review process and we apologize for that. We agree that disseminating the code through an online repository would be an important contribution to the community. To that end, we have uploaded the code to modelDB (<https://senselab.med.yale.edu/ModelDB/>) under accession number 190565. The code is currently only accessible to referees and editors with the password rv15556 and will be made public after publication of the manuscript. We have added a line in the introductory paragraph of the results section containing the modelDB accession number and a modelDB citation.

3. An example of biological comparison overreach in the intro: “This demonstrates a well-defined, computational role of theta oscillations in learning and memory”. Huh? I do not think these results demonstrate anything of the sort, and is representative of the lightweight approach to biological comparison found throughout the manuscript. At the very best, they provide a hypothesis that could be studied further experimentally and computationally.

We fully agree with the referee. As such, we have removed this line in the introduction as part of our efforts to address the criticisms in the comment [1].

4. Comparison of the models trained to output songbird spectrogram. This entire comparison feels superficial to me. While I accept the idea that perhaps song spectrogram can serve as a proxy for a signal comparable in complexity of the signals that RA emits, I do not think it is a stand-in for anything like the neural signal that RA emits (if I’m wrong, convince me with citations). This renders the entire comparison of the trained network to statistics of RA useless.

We agree that the song spectrogram is not a substitute for the neural signal that RA emits. We have altered the text as to satisfy comment [1.] by reducing the biological comparisons and recasting the songbird training example as a potential application of FORCE training.

For our own interests, we sought to determine if using the songbird spectrogram as a supervisor can be justified formally as a mathematical approximation. The approximation is a linearity assumption that has some experimental support [Leonardo and Fee, 2005, Long and Fee, 2008]. The RA nucleus is the second last nucleus in the songbird circuit prior to generation of the song (spectrogram) while the final nucleus is the hypoglossal motor nucleus which subsequently innervates the syrinx to create the singing behavior. Thus, the chain of activity propagation here reads as follows:

$$v_{HVC}(t) \rightarrow w_{RA}(t) \rightarrow x_{nXII}(t) \rightarrow y_{SX}(t) \rightarrow z(t) \quad (1)$$

where each time varying variable here represents the firing activity of the corresponding subscripted region, while $z(t)$ is the final song spectrogram and $y_{SX}(t)$ can denote the muscle activity of the syrinx.

The RA network has been trained to output the song spectrogram after receiving the HVC synaptic chain of activity, thus the propagation of information through our network is given by:

$$v_{HVC}(t) \rightarrow w_{RA}(t) \rightarrow \hat{z}(t) = \phi^T w_{RA}(t) \quad (2)$$

Thus, the approximation that we could have utilized is that the combined effect of the hypoglossal nucleus and the syrinx could be approximated as a linear operator on the firing rates from RA. As we train the network, this linear operator is learned in terms of the readout weights ϕ . Of course linearity is a substantial assumption to make but not quite as crude as making the spectrogram is serving as a stand in for the RA signal. While linear decoding of the spectrogram from RA spiking has not explicitly been performed in the literature, [Leonardo and Fee, 2005] did show that with as little as 30 neurons from RA one could determine the temporal location of the song with a precision of 10 ms (see Figure 7). This strongly indicates that the song-spectrogram can be linearly decoded from the spike trains of RA. Further experimental support comes from the cooling experiments by [Long and Fee, 2008]. In these experiments, a Peltier device is used to cool HVC, which functions to dilate the spike train of the HVC “clock”. The time dilation is propagated throughout the circuit and dilates the song spectrogram up to approximately 50% the normal song length. This might provide some support for the linear operator approximation, i.e. if the combined effect of the hypoglossal nucleus and the syrinx was substantially nonlinear, we would not expect this degree of time dilation without substantial warping due to the nonlinearity in the composition.

Of course, these were just some internal thoughts we had, and these were not included in the manuscript. We agree with the reviewer, and we have thus altered how this section is presented.

5. **Further, whereas physiological computation can be compared between model and neural circuit at an emergent / dynamics level, comparative perturbation and lesion studies rely on the biophysics / connectivity between model and neural circuits to be similar to each other. I just don’t see how this can be true between your models and the neural circuits you discuss, so I am deeply suspect of the perturbation studies in this manuscript.**

We agree with the referee, and we thank them for pointing this out. Perturbation and lesion studies cannot be used for predictive purposes without the appropriate biological constraints in the underlying model. To address this concern, we have rewritten these sections largely to illustrate what can potentially be shown by disrupting these FORCE trained networks.

On a side note, we hope that future applications of FORCE training could be constrained at the biophysical level and even at the connectivity level. For example, at the biophysical level, one can use networks of neurons with fitted and experimentally verified parameter distributions (see for example [Harrison et al., 2015]). At the connectivity level, connection probabilities obtained experimentally (such as from [Ko et al., 2011]) with high rank FORCE training used to determine connection strengths while still creating functional spiking neuronal networks.

6. **Do neurophysiologists really think of theta as a high-dimensional signal? Are you really arguing that what is observed in LFP is actually a high-dimensional signal like that found in HVC? Again, convince me I’ve misunderstood something, but my opinion is the biological connection here is tenuous.**

We thank the referee for their about the theta oscillation. To address this, we have removed references to the theta oscillation and instead have opted to refer to the numerical frequency value.

7. **All in all, this manuscript is pretty confusing, and could stand a pretty significant rewrite. The general thrust of the manuscript is quite clear, but even the mid level details can be quite obscure at times. Adding to this, it appears that the figures have sometimes missing labels, etc.**

We thank the referee for pointing this out. We have added more details to the results such as the sizes of the networks, the transition point to chaos, and training times/test times and the dimensionality of the supervisors. Additionally, we have added missing labels to the figures where applicable (for example specifically labeling the decoders traces during learning, voltage traces of neurons, decoded network outputs etc.). We hope that these changes, in addition to more specific changes in the subsequent and preceding points and the larger changes with regards to biological references will smooth out reading of the manuscript.

8. **Up-state vs down-state.** If I understand correctly, the entire manuscript is relating to neural network simulations. I would be extremely careful with the use of this terminology (and generally separating biological terminology from modeling terminology). My understanding is that circuit level physiologists have extremely precise meanings for these terms. I suspect that the usage UP/DOWN state in this manuscript will antagonize those readers. I recommend using different terms, but admit that I might be wrong in this example.

We appreciate the referees comment on this and it is very valid as up and down states are normally used terms in cortical circuits. This was careless wording on our part. The up/down states referred to the decoded classification performed by the network. Upward deflection in the decoded signal or supervisor correspond to one class while downward deflections correspond to another class, similar to a simple binary variable classifying two states ($\{0,1\}$). As such, we have modified the text to refer to these as upward pulses or downward pulses.

9. **This entire example was extremely unclear to me, though I understood the gist, which is a spiking network trained with FORCE can classify. It might be helpful to plot the input in 3A, and also label all the axes in the figure.**

We thank the referee for pointing this out. Due to the length of the paper, we have opted to move the classifier example to the supplementary materials. Furthermore, additional labels have been added to the figure (now Supplementary Figure S12) for the sake of clarity.

10. **It appears that primarily you used Izhikevich neurons. Is this the major reason the simulations work? Please discuss whether or not this is true in the manuscript. For example, could the theta model or the LIF learned the more complex tasks? This is important because it provides a reason to use more complex spiking models. Maass and his colleagues have been using very complex synapses to make their liquid state machines more temporally interesting.**

In order to test this, we trained the theta and LIF on more complex tasks. For example we successfully trained both networks to perform the short Ode to Joy (1 Bar, 4 seconds long) without any errors in the note sequence with $\tau_D = 50$ ms in each network. This is further elaborated in the comment directly below. The primary determining factor in whether or not a network has convergent parameter regimes is the synaptic time constants of the network, although the specific neuron model does also matter. Qualitatively (and quantitatively, see point below), we've found that the Izhikevich model requires the least parameter tuning, followed by the LIF neuron and the theta neuron performing the worst. We think that the primary reason that the Izhikevich model outperforms the other models is due to the presence of a mixture of long time constants (in the adaptation variables) and short time constants (in the synapses). Finally, we used the double-exponential model for all three neuron models with some simulations run with the single exponential as well (see comment below). Additionally, we had previously stated the superiority of the Izhikevich model in the text. We have moved this paragraph to earlier in the results section to further highlight its importance. It is placed immediately after the new Supplementary Figures S6-S8, as indeed these demonstrate this result conclusively.

11. **Following from the previous remark, a hallmark of the FORCE method is that it “just works”. I would like to see a histogram of the number of successful training attempts for spiking networks, given random initializations in the following form: for all three models train one of the simple sine wave oscillation tasks, and the harder ode-to-joy task. How often does FORCE learning work with spiking networks? Does it “just work”? The reason I ask is because it is well known that a lot of folks have tried FORCE learning directly applied to spiking networks and did not a lot huge success. I am very interested in your explanation for what make FORCE applied to spiking networks so successful in your hands. To me, this should be the key question to answer in a computational paper. Btw, S2 is lovely, but it doesn't get at the question of training robustness for these other model types.**

We thank the referee for pointing this out, and think this was a very good suggestion. As suggested, we took the three types of networks and trained them on both simple sinusoidal oscillators of frequencies $\nu = 1, 5, 10, 20,$ and 40 Hz, in addition to the Ode to Joy signal (New Supplementary Figures S6-8). These networks had their training parameters (G, Q) vary over a range as in the old Supplementary Figure S2. Furthermore, we varied the network synaptic decay time constants $\tau_D = 10, 20, 50, 100$ ms with a rise time of $\tau_R = 2$ ms to really see where exactly in the parameter space FORCE works. Additionally, we

separately ran identical meshes with $\tau_D = 20$ ms and $\tau_R = 0$ ms to determine if the synaptic rise times were necessary for convergence.

Over these four parameters (Q, G, ν, τ_D) and three model types (Theta, LIF, Izhilkevich) we were able to resolve that yes, FORCE does work. However, this is of course dependent on the parameters for the underlying networks (New Supplementary Figures S6-8). The most important parameter the synaptic decay time constant of the network. If the supervisor has fast dynamics, fast synaptic time constants are necessary while if the supervisor has slow dynamics, slow synaptic constants are necessary. This is explicitly stated in the results section now after the new Supplementary Figures S6-S8 are referenced. The Izhikevich model is the most robust when it comes to this particular trend with working parameter regimes for the majority of sinusoids tested. We have explicitly stated that the Izhikevich model performs the best in the paragraph immediately after these new Supplementary Figures are referenced. All neuron models had convergent parameter regimes for the Ode to Joy example, with the Theta and LIF models requiring a 50 ms decay time constant, while the Izhikevich model only required a 20 ms decay time constant. The convergent regions in the (Q, G) parameter space were contiguous, and not patchy.

We conjecture that the primary reason we were successful in applying FORCE to spiking networks was through a pair of heuristics we used after observing the qualitative behaviors of the rate networks from David Sussillo’s implementation from [Sussillo and Abbott, 2009].

In particular, it was noted in the original FORCE paper that if the amplitude of the target function is too low (see Figure 2K from [Sussillo and Abbott, 2009]), the learning does not converge to a successful solution. Furthermore, Figure 2K shows that the rate basis generated by the FORCE training did not stabilize, as it had done for Figure 2B from [Sussillo and Abbott, 2009]. We found a similar result in the spiking network, where if the feedback was too low, the resulting spiking basis would not stabilize. Our solution to this was to add the Q parameter to the feedback weight matrix:

$$\Omega = G\omega^0 + Q\eta\phi^T$$

The parameter Q has to scale with G in some fashion such that learned feedback can stabilize the dynamics. Rather than attempting to non-dimensionalize the spiking network, we applied the heuristic that Q should be large enough to have an observable effect on the spiking during training. To make this heuristic more formal, we can examine the original FORCE paper in more detail. In this paper, tuning curve used was $r = \tanh(x) \in (-1, 1)$ which is bounded. Furthermore, $g = O(1)$ with the feedback term $E\hat{x}(t)$, which is also $O(1)$ provided that $\hat{x}(t)$ was $O(1)$, and that was sufficient to stabilize the rate dynamics. Here, we hypothesized that we needed a similar balance between the order of magnitude for fluctuations in the term $G\omega^0$ and the feedback term $Q\eta\phi^T$. Operating on this hypothesis, we can justify the resulting orders of magnitude for Q for the the Theta and LIF models and the Izhikevich model (after rescaling into identical units of s^{-1} for the filtered spike train). This is done by analyzing the balanced network equations pre-training. Defining $s_i(t)$ as the following:

$$s_i(t) = \sum_{j=1}^N G\omega_{ij}^0 r_j(t) \quad (3)$$

then the standard balanced network argument shows the following large network asymptotics:

$$\langle s_i(t) \rangle \sim 0, \quad N \rightarrow \infty \quad (4)$$

$$\langle s_i(t)^2 \rangle \sim O(G^2 \langle r_i(t)^2 \rangle), \quad N \rightarrow \infty \quad (5)$$

Thus, in order to stabilize the fluctuations generated by the balanced network we need a feedback term that is of $O(G\sqrt{\langle r_i(t)^2 \rangle})$. Thus, if $G = 10^{-1}$ for the LIF neuron, while the instantaneous firing rate has a $\langle r_i(t)^2 \rangle = O(10^2)$ then we should require an $Q = O(10)$, which is roughly what we observe. Note that this is based on the assumption that the fluctuations in $\eta\phi^T \mathbf{r}(t) = \eta\hat{x}(t)$ are $O(1)$. Higher dimensional supervisors require another rescaling to reduce effect of the feedback term.

In the original FORCE paper, the time constant for the network was $\tau = 10$ ms, with $P = \alpha^{-1}I$ and $\alpha = O(1)$ with RLS applied every 20 ms (in David Sussillo’s uploaded code, [Sussillo and Abbott, 2009]). While in the previous case we could easily conduct a dimensional analysis to determine how to scale Q, G , it is more difficult here. We again employed a heuristic: the spiking basis used for an oscillator should stabilize rapidly (within the first period of oscillator presentation). This ensures that you operate similar

to FORCE in the rate equations where the basis is stabilized rapidly, and the remaining weight changes are used to stabilize the weight matrix. To that end, after some trial and error we realized a smaller Δ_t was required and our values vary between $O(10^{-1})$ to $O(1)$ ms. The effective rate parameter γ was taken to be as large as possible to ensure stability, yet maintain fast learning.

These two heuristics, stabilizing the spiking basis in the first period of supervisor presentation, and balancing the fluctuations in the static and learned recurrences to be of the same order have been added to the main text of the manuscript. The derivation of equation (4) and (5) has been added to the materials and methods (Section “ G and Q Parameters”). We hope that these additions both address the degree of parameter tuning required for direct implementation of FORCE training, in addition to explaining why this technique works with spiking neurons.

12. **It appears you have worked on the fully connected case. FORCE was also introduced for the sparse case (Sussillo & Abbott 2009), have you tried this? Does it work? (No need for any extra work here, just asking).**

We have not implemented the fully connected FORCE case. The primary reason we did not attempt this is due to the fact that it requires an $O(N^2)$ computation at each application of RLS which slows down simulations. We think however that this would be an improvement over the low rank feedback case in terms of accuracy as there are more free parameters.

13. **I really liked the part about the “clock” driving signal, although I think the name is misleading. I do not think it’s fair to call the input to either the birdsong example, or the Ode to Joy example a clock, which I will call a high-dimensional driving signal (HDDS). My guess is that much more important than the exact timing of the driving signal is the dimensionality of it. Namely, that the high dim input drives the reservoir to explore more states, which allows the readout to have more capacity because more weight space is being used. At any rate, calling it a clock signal is incorrect in my opinion and it should be changed (Again please argue if I’ve missed something).**

We think that renaming the “clock” is an excellent suggestion. We opted for the term High Dimensional Temporal Signal (HDTs) as both the dimensionality and the temporal content of these signals seems to matter. In particular, we have found that stretching or dilating the HDTs has a corresponding effect on the downstream networks.

14. **I do have a substantial criticism here, however. The fact that hand crafted HDDS signal was used also solves a major problem that RNNs suffer from, simply by kicking it down the line. This is the problem of stabilizing long or complex trajectories. If one can train an RNN to generate that HDDS in a stable manner (as was tried in Fig. 5 before moving to a “hand crafted one, for the rest of the examples), then that is progress, otherwise it is declaring success when there is none. So in my opinion all examples with the hand crafted HDDS should be rerun with the HDDS generated from an actual network (what the authors call the non-autonomous case should be replaced everywhere with what they call the autonomous case). Otherwise, I think it is very misleading.**

We appreciate the criticism from the referee here. We agree that it is true that if a network could not generate an HDTs on its own (as in the songbird task), no progress can be claimed. However, from our experience these high dimensional driving signals are simple to train a network to reproduce, see for example Supplementary Figure S18 with the long form of Ode to Joy trained on an Izhikevich network. Here, the supervisor consists of 69 components. The first 5 components contain the supervisor while the last 64 components contain the HDTs. There are no inputs into the network and the network has to generate its own internal HDTs, simultaneously to the production of the song. This is demonstrated in Figure 1(b) attached to the bottom of this document as well for the shorter Ode to Joy example.

To further demonstrate this, we have also trained a network to simultaneously generate its own internal HDTs, in addition to replaying the movie. The supervisor consists of 1984 dimensions (1920 pixels, 64 dimensional HDTs). This figure has been added to the supplementary materials. Note that this is entirely different from the movie example we had considered before, as now there is only a single network, and all neurons simultaneously encode the HDTs and the movie scene, as requested by the reviewer. A figure demonstrating this result has been added to the supplementary materials (Supplementary Figure S19).

15. **As a remark on the HDDS, and not a criticism. I want to suggest a thought experiment: I guess that you could actually achieve the same performance if you removed the HDDS as input, and instead encoded the target signal in an analogous way (as it gets fed back to the system). Specifically, taking a particular target signal, chopping it up into T time segments encoded into T different dimensions (say by multiplying by a gaussian bump centered in the middle of the T th section, and then training those segments. So if you had M outputs, now you have $M \times T$ outputs (i.e. the target signal is 0 outside of it's assigned segment). The cardinality of the feedback η would also be increased by T , if there were T bumps in the HDDS to begin with. Now because the effective driving signal, η , is different for each T segment, the driving force of the feedback would also explore more of the the full dimensionality of the space, and perhaps work. Worth some thought, but you do not have to do anything here to satisfy me**

This is an interesting thought experiment. This may solve a problem we were having. We could only ever elicit spontaneous replay (continuous replay of a song or stored movie) with a HDTS. However we did discover a mechanism for replay that could reconstruct a stored signal with partial signal input. This operates in a similar fashion to an attractor network. For example, for the long Ode to Joy song we could activate any 8 sequential notes, in addition to their associated HDTS components and the network would replay the rest of the song starting from the last note activated. However, this attractor network like reconstruction of the song required knowledge of the notes absolute position in the entire song (via the HDTS), thus we regarded it as biologically implausible. However, with the implementation suggested by Referee 1, a partial component of the stimulus is dissected into discrete components and an absolute sense of time or order via the HDTS is no longer required. Only a relative sense of order in the subset of inputs used for cued replay is required. We greatly appreciate this thought experiment.

16. **Classification Task I didn't understand what was meant by the upstates correspond to one class while the down states correspond to another. I guess this means that the readout, \hat{x} is correlated?**

The readout, $\hat{x}(t)$ is in one of two states after presentation of an input at time $t = 0$:

$$\hat{x}(t) = \sin(2\pi t) \quad t \in [0, 1] \quad (6)$$

which corresponds to the upward deflection of the sine wave, or

$$\hat{x}(t) = \sin(2\pi(t + 1)) \quad t \in [0, 1] \quad (7)$$

which corresponds to the downward deflection. Downward deflections correspond to one statistical class while upward deflections correspond to the other class in a binary classification problem. In any case, due to the length of the paper, we have decided to put this example in the supplementary material. These figures are now Supplementary Figures S12-S14.

17. **Ode to Joy How was the song restarted in "1000 stretch of simulation time after training?"**

The network was initialized with random initial conditions. The initial voltages were drawn from a uniform distribution over $[v_{reset}, v_{peak}]$, and all other variables (adaptation, synaptic currents etc.) were set to 0. The weight matrix for the network was set to

$$\Omega = G\omega^0 + Q\eta\phi^T$$

where ω^0 , η and ϕ were from the previous iteration of FORCE training (with ϕ having been learned, and the former two being static). The network has an initial transient (on the order of seconds) before it settles back into the Ode to Joy Oscillator.

We have included a figure that demonstrates this behavior at the end of this document for both an Ode to Joy Oscillator trained without a HDTS (Figure 1A, 5 component supervisor) and one with a HDTS (Figure 1B, 21 component supervisor). In both cases, the networks vary randomly for a short period of time due to the initial transient until they settle onto the learned attractor for both networks. Both networks settle on their oscillators after a transient that is on the order of 10 seconds.

18. **Birdsong What is alpha, is it defined in the methods?**

α is an extra term we used to control the amplitude of the positive weights in the final learned weight matrix:

$$\omega_{ij} = \alpha (G\omega_{ij}^0 + Q\eta_i \cdot \phi_j)_+ + (G\omega_{ij}^0 + Q\eta_i \cdot \phi_j)_- \quad (8)$$

where $(x)_\pm$ denotes:

$$(x)_\pm = \begin{cases} x & x > (<)0 \\ 0 & x \leq (\geq)0 \end{cases}$$

The values $\alpha = 1$ yield the original weight matrix, while $\alpha > 1$ amplifies the excitatory connections and $\alpha < 1$ diminishes the excitatory connections. We have added an identical explanation to the materials and methods to aid readers (section ‘‘Using Spiking Neural Networks for Pattern Storage and Replay’’ (Figure 4)).

19. Same points about clock signal dimensionality above hold for this example as well.

We agree with the referee here, the dimensionality of the clock does help hence the relabeling of the signal as the high-dimensional temporal signal.

20. Moving past the dimensionality of the clock signal, I am also concerned about its frequency. Rajan and Abbott (2010?) showed that the removal of chaos in rate networks was dependent on both the amplitude, and the frequency. These frequencies mentioned on this paper are surely a function of the network parameters of the Izhikevich neurons, as are likely the natural periodicities of the chaos inherent in the system. The idea that these particular frequencies have any meaning at all to real neurons is intriguing, yet hardly substantiated here. So using words like theta and delta oscillations is extremely misleading, in my opinion, as they mix terminology from neurobiology and network modeling. If you were to make a detailed analysis, comparing network parameters, and network architectures in depth, sweeping relevant parameters, perhaps this could stand on it’s own as an additional stand-alone contribution. As it is here, it feels weak, and I recommend removing it.

To understand how the amplitude and frequency of the HDTS interact with the learning, we have run a two parameter mesh over both the frequencies (at discrete values of 1 Hz, 2 Hz, 4 Hz, 8 Hz, 16 Hz, 32 Hz, 64 Hz, and 128 Hz) in addition to a parameter that controls the HDTS amplitude (from 0, 0.1, . . . 1). While this 66 point mesh is somewhat low in resolution, it has revealed that for a given HDTS amplitude, there is an optimum frequency for the HDTS input to produce the most accurate replay, as measured a pixel by pixel correlation coefficient. This maximum is usually in the lower frequency range in between 8-16 Hz. Surprisingly, for larger fixed HDTS amplitudes, the optimum is usually either in the 8 Hz, or 16 Hz HDTS frequency. We hope this also helps to address point 20. below, ‘‘How does the performance of the clock signal vary as a function of the clock period’’. The frequency mesh has been added as a supplementary figure (Supplementary Figure S19).

To address the point on the frequency ranges used in this paper, we have removed usages of the physiological frequency ranges (theta, delta, etc.) used in the literature and have instead directly stated the numerical values for the frequencies.

21. How does the performance of the clock signal vary as a function of the clock period? Is theta necessary? Why theta? As some function of the biophysical parameters of the network? I have to admit, the implied connection here to the hippocampus feels quite tenuous. I think it is interesting of its own right to understand why an abstract network of neurons would benefit from this oscillation. Most critically, one should test the frequency of the clock signal here, as is done for amplitude in Figure 6E.

This is partially addressed in point 20, above. The numerics indicate that in our network, the 8-16 Hz range is optimal for storing the movie stimulus (Supplementary Figure S19). This may be due to an interplay between the temporal dynamics of the stimulus and the ability of the HDTS frequency to help stabilize the chaotic network dynamics. Furthermore, we have reduced references to the hippocampal literature (see Point 1., above).

22. In the discussion, it’s likely worth pointing out why your HDDS is different than the basis set of additional signal in De’Pascale and Abbott. I.e. they also add more signals. If so, it might be meaningful to cite them here.

We thank the referee for mentioning this point of comparison between our work and De’Pasquale and Abbott. We have made the comparison between these two high-dimensional signal types as its own paragraph in the discussion, section “FORCE training and other top down approaches”.

23. **Having an autonomous recall here really amounts to whether you can train a network to output a HDDS, right? So I don’t really think there’s much here in terms of highlighting the autonomous recall, in fact the word usage is actively misleading and should be changed, hopefully by removing it, and only using the autonomous case as mentioned earlier.**

The naming, “autonomous” has been changed. This is addressed below in greater detail in comment 28 and comment 14 above.

24. **58 - 60 I think there are uses of FORCE that aren’t strictly rate based, but are more outside the box. E.g. light reservoirs, as well as biological neuron reservoirs? Not sure, but perhaps should take a look.**

We thank the referee for bringing this to our attention. We discussed any literature we did find on this topic in the discussion session, “FORCE training and other top down approaches”. However, we could not find any additional references on “light reservoirs” and biological neuron reservoirs specifically. We are happy however to include any additional references that the referee recommends.

25. **the spike frequency adaptation has been heavily used by Maass in his spiking liquid state machines. The usage in many of those manuscripts is short-term plasticity ala Tsodyks and Markram. Probably should include this as a citation here. It would be interesting to see FORCE work with that plasticity type as well, though I am not asking you to do it.**

We thank the referee for bringin up Wolfgang Maas’s work with adaption and the Tsyodyks-Markram synapse to our attention. We have discussed the literature we found on this topic in the discussion session, “FORCE training and other top down approaches” in the first paragraph of “FORCE training and other top-down approaches”.

26. **RLS vs RLMS - Isn’t the technique (eqns 14-16) you are using called recursive least squares, not recursive least mean squares? My understanding is that RLMS is the first-order version that does not involve the inverse correlation matrix (eqn 16), and the learning rate (singular) is chosen heuristically. Btw, it is my strong hunch that with your HDDS, you could actually use the RLMS, and not perhaps need the 2nd order RLS method. Just a guess, no need for you to do anything.**

We thank the referee for pointing this out. The technique is actually called RLS as opposed to the first order version RLMS. We have made sure that this has been corrected in the paper.

With regards first order learning, we have done some unpublished work primarily working with the rate equations on this very topic. The first order method (RLMS, or pure gradient descent) takes a long time to converge and we have not had much luck with it beyond learning very simple oscillators. However, we have had some promising initial results with a pair of more biologically plausible learning rules. The techniques we tried were RMSprop (Root Mean Square Propagation) and Rprop. RMSprop is a first order method that stores the magnitude of the gradients and uses them to alter the learning rates for each component of the decoder. Rprop is another first order method based on the signs of the gradient and both techniques did at least have convergence with some more complicated oscillator examples (sum of sines etc). We thank the referee for pointing out the potential use with an HDTS and a first order technique. We intend to pursue it in future work.

27. **Figure 1D is missing in the caption**

A caption for Figure 1D has been added. We thank the referee for pointing this out.

28. **Figure 6 Autonomous vs. Nonautonomous I think these words are misused. My understanding of autonomous is that a dynamical system receives no input whatsoever. That is not the case for either recall network described in figure 6. The left panel of A would be autonomous if the recall network projected back to the clock network. This issue of autonomous or not is relevant because stabilizing these networks is a large part of the challenge.**

We appreciate the referees comments on this because it can lead to confusion. As such, we have altered the names of these two cases as “Internally Generated HDTS” and ”Externally Generated HDTS” where in the former, a network generates an Internal HDTS (post FORCE training) where as in the latter, the HDTS is supplied and manipulated by us.

For our own interest, we did look further into the definition of autonomous and non-autonomous with regards dynamical systems theory. A system being autonomous or non-autonomous does not necessarily mean one with inputs or without. It is explicitly the time dependence in the differential equation. For example, the following system is autonomous, despite being able to explicitly solve for y with $y(t) = \exp(-t)$

$$\dot{x} = F(x, y) \tag{9}$$

$$\dot{y} = -y \tag{10}$$

which is exactly equivalent to the following non-autonomous system.

$$\dot{x} = F(x, y_0 \exp(-t)) \tag{11}$$

This approach does not always yield an autonomous/non-autonomous representation that is useful for example the system $\dot{x} = tx$ can be written as $\dot{x} = yx, \dot{y} = 1$, but has no stable equilibria to analyze and is better analyzed as a non-autonomous system (with exponential dichotomies for example).

In any case, we agree that changing the label is an improvement and thus it is now “internally/externally generated HDTS”

Referee 2: Major Points

- In this paper, the authors train spiking neural networks using FORCE on a variety of tasks. Several variants of spiking models are tested, with the Izhikevich slowly adapting neurons performing the best. The main tasks studied are sequence generation, where the authors show the advantage of having an explicit clock signal. Several analogs with experimental results are suggested. The paper has several interesting results, but I feel that the multitude of results prevents a thorough analysis of any specific one. For instance, what is the main message of the paper? A method for training spiking networks with FORCE, or the importance of clock signals?**

We thank the referee for pointing this out. To address this, we have rewritten sections of this paper to make the primary focus about FORCE training and its use in training spiking neural networks, with the secondary messages being that FORCE training improves with the inclusion of clocks or as we now refer to them, High-Dimensional Temporal Signals (HDTS, see Point 12 to referee 1) and that FORCE trained networks can be potentially to construct functional, biologically constrained spiking models of neuronal circuits.

The manuscript has been streamlined by shortening parts of the introduction, and discussion in addition to rewriting the abstract. Finally, we have provided two copies of the revised manuscript; with and without the tracked changes from the previously submitted version.

- The paper’s title and abstract emphasize the application of FORCE to spiking networks. The introduction also describes the difficulty of using FORCE on spiking networks. Indeed, the authors show that spiking networks can be trained by FORCE, but there is no explanation of why this works in the current study and failed for other groups. Several questions naturally arise that can shed light on this topic:**

We achieved success where other groups did not in FORCE training spiking networks by using a pair heuristics we employed to determine where the convergent parameter regimes were. These were derived after observing the qualitative behaviors of the rate networks from David Sussillo’s implementation from [Sussillo and Abbott, 2009].

In particular, it was noted in the original FORCE paper that if the amplitude of the target function is too low (see Figure 2K from [Sussillo and Abbott, 2009]), the learning does not converge to a successful solution. Furthermore, Figure 2K shows that the rate basis generated by the FORCE training did not stabilize, as it had done for Figure 2B from [Sussillo and Abbott, 2009]. We found a similar result in the

spiking network, where if the feedback was too low, the resulting spiking basis would not stabilize. Our solution to this was to add the Q parameter to the feedback weight matrix:

$$\Omega = G\omega^0 + Q\eta\phi^T$$

The parameter Q has to scale with G in some fashion such that learned feedback can stabilize the dynamics. Rather than attempting to non-dimensionalize the spiking network, we applied the heuristic that Q should be large enough to have an observable effect on the spiking during training. To make this heuristic more formal, we can examine the original FORCE paper in more detail. In this paper, tuning curve used was $r = \tanh(x) \in (-1, 1)$ which is bounded. Furthermore, $g = O(1)$ with the feedback term $E\hat{x}(t)$, which is also $O(1)$ provided that $\hat{x}(t)$ was $O(1)$, and that was sufficient to stabilize the rate dynamics. Here, we can hypothesize that we need a similar balance between the order of magnitude for fluctuations in the term $G\omega^0$ and the feedback term $Q\eta\phi^T$. Operating on this hypothesis, we can justify the resulting orders of magnitude for Q for the the Theta and LIF models and for the Izhikevich model (after rescaling $r_i(t)$ to units of s^{-1}). Considering the balanced network equations and define $s_i(t)$ as the following:

$$s_i(t) = \sum_{j=1}^N G\omega_{ij}^0 r_j(t) \quad (12)$$

then a standard balanced network argument shows the following large network asymptotics:

$$\langle s_i(t) \rangle \sim 0, \quad N \rightarrow \infty \quad (13)$$

$$\langle s_i(t)^2 \rangle \sim O(G^2 \langle r_i(t)^2 \rangle), \quad N \rightarrow \infty \quad (14)$$

Thus, in order to stabilize the fluctuations generated by the balanced network we need a feedback term that's of $O(G\sqrt{\langle r_i(t)^2 \rangle})$. Thus, if $G = 10^{-1}$ for the LIF neuron, while the instantaneous firing rate has a $\langle r_i(t)^2 \rangle = O(10^2)$ then we should require an $Q = O(10)$, which is roughly what we observe. Note that this is based on the assumption that the fluctuations in $\eta\phi^T \mathbf{r}(t) = \eta\hat{x}(t)$ are $O(1)$. Higher dimensional inputs require another rescaling to reduce effect of the feedback term.

In the original FORCE paper, the time constant for the network was $\tau = 10$ ms, with $P = \alpha^{-1}I$ and $\alpha = O(1)$ with RLS applied every 20 ms (in David Sussillo's uploaded code, [Sussillo and Abbott, 2009]). While in the previous case we could easily conduct a dimensional analysis to determine how to scale Q, G , it is more difficult here. We again employed a heuristic: the spiking basis used for an oscillator should stabilize rapidly (within the first period of oscillator presentation). This ensures that you operate similar to FORCE in the rate equations where the basis is stabilized rapidly, and the remaining weight changes are used to stabilize the weight matrix. To that end, after some trial and error we realized a smaller Δ_t was required and our values vary between $O(10^{-1})$ to $O(1)$ ms. The effective rate parameter γ was taken to be as large as possible to ensure stability, yet maintain fast learning.

These two heuristics, stabilizing the spiking basis in the first period of supervisor presentation, and balancing the fluctuations in the static and learned recurrences to be of the same order have been added to the main text of the manuscript. The derivation of equation (4) and (5) has been added to the materials and methods (Section " G and Q Parameters"). We hope that these additions better illustrate why we had success with this technique where others did not.

3. How does the size of the network required to achieve a certain performance compare between rate and spiking models?

This is an excellent question. To address this, we have run a parameter sweep with the network size varying from $N = O(1)$ to $N = O(10^4)$ for networks of LIF neurons learning a 5 Hz sinusoidal oscillator in addition to networks of rate neurons with the standard type-I firing rate $r(x) = \sqrt{x}$ at similar network sizes, and the $\tanh(x)$ firing rate using the code uploaded from [Sussillo and Abbott, 2009] (New Supplementary Figure S10). The LIF network scaled like $\approx N^{-1/2}$ in the L_2 error while both rate networks were closer to $O(N^{-1})$. The exact values of the slopes are included in Supplementary Figure S10 with a more thorough explanation. These results have been added to the main text in the section "FORCE trained spiking networks can learn dynamics rapidly with chaotic spiking initialization" in addition to the new Supplementary Figure S10. We hope that the comparison between spiking and two different rate-models will also address point 11 below.

4. **Does the slow ($a=0.01$) parameter render this spiking model an effective rate model, for the parameter regimes used in the study?**

This is an interesting question. We do not think that the slow parameter turns the Izhikevich model into an effective rate model. To the best of our knowledge, there are two manipulations that can turn a spiking network closer to a rate network: long time constants of the post-synaptic filters, or faster firing rates. For our networks, the long-time constant is in the adaptation variable which does not take part in the decoding and is an auxiliary variable for the neuronal equation that serves to slow down firing rates. However, our spiking network is still likely employing a rate code. This is demonstrated by the error scaling (point [2.] above) of \sqrt{N}^{-1} in the L_2 error as opposed to the more optimal N^{-1} scaling in spike-based codes [Boerlin et al., 2013].

5. **The Lorenz attractor was studied in the original Jaeger paper, and in others. How does the performance of the current model compare with previous ones? Specifically, this can be a useful benchmark to assess the effect of spiking vs. rate models.**

We thank the referee for bringing this to our attention. We have opted to compare our Lorenz attractor with one generated in convergent parameter regions of the original FORCE rate implementation in [Sussillo and Abbott, 2009] using the code uploaded by David Sussillo. Some minor adjustments were necessary for convergence (increasing network size, adding a $Q = 3$ parameter to the feedback term) to obtain convergence as in the original FORCE implementation, only the x variable was used as a supervisor. The metric we used to compare attractors was to generate a probability density over the steady dynamics and compare density functions for rate and spiking networks. By randomly sampling long realizations on the simulated attractor, steady state density functions for the three marginal densities (for the (x, y, z) variables) can be estimated. The same procedure can also be applied to the trained spiking network attractor. With these two density functions, a standard L_2 comparison yielded errors of $(0.27, 0.30, 0.24)$ for the rate network while $(0.52, 0.38, 0.30)$ for the spiking network. This indicates similar levels of performance on the longer time scale. On the short time scale, we computed the stereotypical Lorenz tent map that plots the successive peaks in the z_n variable as a function of the previous peak z_{n-1} . Here, the rate network performed better than the spiking network. This entire procedure was explained in greater detail the new Supplementary Figure S9.

6. **Related to the previous point – what do we gain from using a spiking model? I agree that there is value to being able to train spiking networks. But there can also be added value such as spike timing statistics, spike triggered averages and other measures, that might not be observable in a rate model. These are not explored here.**

We agree with the referee that there is added value in computing spike timing statistics with a spiking network model that cannot be gleaned from a rate model. For example, for the song bird task, we did compute interspike-interval distributions and compare these with the experimentally obtained data. Additionally, we perturbed the underlying networks (through the learned weights) to ascertain how both the performance of the network in the task, and the underlying spiking statistics of the network would change (Figure 5 E). The coefficient of variation was also computed both before and after learning in Supplementary Figure S5 and S11 for the networks learning a sinusoidal oscillator. Finally, the spike density as a function of the phase of theta oscillation was also added as a supplementary figure to address point 10 (New Supplementary Figure S21).

7. **Theta modulation. This is presented as a surprising finding that is consistent with experimental results. But the origin of theta should be determined from the network simulations. Several directions are:**

8. **Is there theta in the input clock signal?**

The input clock or HDTS signal drives the network into a theta oscillation by activating a different assembly of neurons every 125 ms. To elaborate further, these signals are constructed as a discretization of time, T into m sub-intervals. Each subinterval generates an extra component (dimension) in the supervisor and within the subinterval, a pulse or upward deflection of the supervisor is generated. For example, if we consider the interval from $[0, T]$ with m subintervals of width $I_n = [T(\frac{n-1}{m}), T(\frac{n}{m})]$, $n = 1, 2, \dots, m$. The supervisor is constructed as a series of pulses centered in the intervals I_n . This can

be done in multiple ways such as by using Gaussian pulses

$$x_n(t) = \frac{1}{\gamma} \exp\left(-\frac{(t - T \left(\frac{n-1}{2m}\right))^2}{\sigma^2}\right) \quad (15)$$

where σ controls the width and γ is used to scale the maximum of the Gaussian to 1 (unlike the typical normalization of the Gaussian where its used as a density function). Alternatively, one can also use piecewise defined functions such as

$$x_n(t) = \begin{cases} |\sin(m\pi t)| & t \in I_n \\ 0 & \text{otherwise} \end{cases} \quad (16)$$

The end effect of these signals is to divide the network into a series of assemblies. Each assembly is activated by the preceding assemblies to propagate a long signal through the network. The assemblies are activated at a specific frequency in the network, dictated by the width of the pulses in the signal. For the networks we considered, the activation rate was a theta frequency. However, we have considered other frequency ranges (see new Supplementary Figure S19). An identical explanation of the construction of the clock and its effects on network functioning is now included in the Materials and Methods section ‘‘High Dimensional Temporal Signals’’.

9. Does it relate to the single neuron timescales?

We think that the activity of our LFP proxy is partially related to the single-neuron time scales, but the theta oscillation is from clock input. If we remove the theta oscillation, the network does have some irregular oscillatory component that is on a slower time scale (1-2 Hz). This is a combination of the balanced network state, the adaptation parameters of the Izhikevich model, and the FORCE trained component that tries to stabilize (but fails) the replay. We did implement different single neuronal time scales ($C = 250$ vs. $C = 150$). We found the results largely identical (see Supplementary Figure S19) at either capacitance value for the Izhikevich model.

10. This is an opportunity to take advantage of the spiking models, and analyze the relation between spiking and the phase of theta.

We appreciate this comment from the referee and think this is a good idea. This has been implemented as a new supplementary figure (Supplementary Figure S21). The result is a non-uniform distribution of the spiking across the theta phase in the input. The non-uniform distribution is similar to the asymmetric, unimodal distributions observed experimentally (see [Mizuseki et al., 2009]). However, depending on the specific neuron type, the mode is either before or after the peak of the reference theta oscillation (Figure 3, [Mizuseki et al., 2009]). This varies depending on neuron type (pyramidal vs. interneuron) and region (CA1, CA3, EC etc.).

11. The results about compressed and reversed replay seem trivial, but this could be a misunderstanding on my part. The way I understood from the main text and methods, the network essentially acts as a mapping between specific input patterns and short output patterns. There is no relevant internal dynamics for the transition from one note/frame to the next. If this is the case, then the output is completely determined by the clock signal, and there is no need to train the internal dynamics. In fact, this is indeed the case according to the methods (the task also works for $Q=0$). This interpretation is a very natural concern, and should be addressed in the text.

We thank the referee for their concern about this. Certainly, it would be true that if given a particular input, $u(t)$ to a static function, F which generates the temporal basis $r(t) = F(u(t))$ then it would be entirely trivial that the L_2 optimizing approximant

$$\hat{x}(t) = \phi^T r(t) = \phi^T F(u(t)) \quad (17)$$

can be compressed/dilated or reversed in time by compressing, dilating or reversing the input signal $u(t)$.

This is proven below by considering the optimal decoders, ϕ^*

$$\phi^* = \left(\int_0^T x(t)F(u(t))^T dt \right) \left(\int_0^T F(u(t))F(u(t))^T dt \right)^{-1} \quad (18)$$

$$= \left(c \int_0^{T/c} x(\tau)F(u(\tau))^T d\tau \right) \left(c \int_0^{T/c} F(u(\tau))F(u(\tau))^T d\tau \right)^{-1} \quad (\text{sub } t = c\tau) \quad (19)$$

$$= \left(\int_0^{T/c} x(\tau)F(u(\tau))^T d\tau \right) \left(\int_0^{T/c} F(u(\tau))F(u(\tau))^T d\tau \right)^{-1} \quad (20)$$

$$= \phi_c^* \quad (21)$$

which proves that the decoders for the function $x(ct)$ on $[0, T/c]$ using the basis $F(u(ct))$ are invariant to c , the compression factor. However, for two very important reasons it is a non-trivial result for any dynamical system and indeed, and more importantly any recurrent spiking neural network.

As a mathematical counter example, we can consider the forced Van der Pol oscillator that receives an oscillatory input $u(t)$. The response of this system can vary from stable oscillations to chaotic dynamics with different frequencies of $u(t)$. Thus, compressing or dilating $u(t)$ will alter its frequency, and the results can vary dramatically in the response of the Van der Pol oscillator.

For a spiking neural network, the basis is generated as a post-synaptically filtered spike train from a recurrent neural network. Thus, if we compress our input clock or controller $u(t)$ there is no reason at all to expect our output basis $r(t)$ to compress. In particular, such a compression is not possible as it would require both the sequence of spikes across the network to be compressed/dilated in time by a factor of c , and the post-synaptic filters by the same amount. The post-synaptic filters cannot be altered by a network. This explanation is mentioned (albeit much more succinctly) in the results section detailing the compression of the HDTS.

Finally, even in the $Q = 0$ case, the network still has recurrent dynamics through the G parameter and thus the basis is generated dynamically by the feedforward weights and clock input, and the recurrence. Thus, compression or dilation of the HDTS again does not necessarily imply compression or dilation of the resulting temporal basis and thus the movie replay.

12. There is very little quantitative analysis. How does performance change with network size? With spiking or rate models? With Dale’s law?

We thank the referee for pointing out this weakness in the manuscript. This has been partially addressed in point [2.] above for the performance with network size, and spiking versus rate models. This is not easily addressed with Dale’s Law. The specific implementation we have for enforcing Dale’s law requires an $O(N^2)$ manipulation in the weight matrix at each time step of RLS training. The offending computation is enforcing the sign bounds on the summed weight matrix, $G\Omega^0 + Q\eta\Phi^T$. This computation forces the simulations to take much longer than not-enforcing Dale’s law, and makes the type of quantitative analysis required harder. However, we did look at our simulation in Figure S11 and computed the $\log(L_2)$ error as -1.34 . This comparable to the values in Supplementary Figure S8. The computed error was added to the results section in “FORCE trained spiking networks learn dynamics rapidly with chaotic spiking initialization”.

13. The classification example seems unconnected to the paper. In itself it does not seem very surprising or novel, and I think it only distracts from the other messages of the paper.

We thank the referee for this comment and we also think that it distracts from the main message of the paper. Therefore, we have relocated the classifier example to the supplementary materials and minimize the references to it throughout the text (New Supplementary Figure S12).

14. The procedure for improving the NEF in the supplementary is an interesting result, but I doubt that many readers will notice it.

We thank the referee for their interest in the high-rank NEF result. To draw the readers attention to this result, we have rewritten a section of the results and added a paragraph that outlines the steps involved high-rank NEF derivation in the supplementary material.

15. **The parameter values vary over very large ranges (G=1 vs. G=13000). In the rate networks, it is known that G=1 is the edge of chaos, and this value has functional importance. For the spiking networks, it would be very helpful to indicate the relation between the chosen parameter values and the onset of chaos.**

We appreciate this suggestion from the referee and concur. Thus, we have simulated networks of LIF, Theta, and Izhikevich model types of for increasing G values from $G = 0$ and computed the coefficient of variation and interspike interval distribution to ascertain the onset of chaos. For the Theta and Izhikevich models, there is an immediate transition from quiescent spiking to chaotic spiking behaviors at $G \approx 0.02$ and $G \approx 10^3$, respectively. This is due to the fact that the applied current to these networks is set to the threshold or rheobase value. We considered the LIF model for two parameters, $I_{bias} = -40(\text{threshold})$ and $I_{bias} = -39$ (superthreshold). In the threshold current regime, the behavior is qualitatively similar to the theta model with a small G value determining the onset to chaos. The harsh nonlinearity in the transfer function for the LIF model makes determining its exact value difficult however it is small $G < 0.01$. In the superthreshold regime, the LIF model transitions from tonic spiking with a constant rate to chaotic spiking at $G = 0.04$. In addition to this, we conducted a perturbation analysis similar to that in [Monteforte and Wolf, 2012]. We simulated the network for 5-10 seconds depending on the network type. Then, we initialized two trajectories repeatedly, a reference trajectory and a perturbed one with a perturbation magnitude of ϵ for varying values of ϵ . This was iterated 200 times per ϵ . The results mimicked the “flux-tube” result of [Monteforte and Wolf, 2012]. The perturbations either decayed exponentially because they were inside the radius or they caused exponential divergence. The perturbations have a probability of causing a divergence from the reference trajectory that varies like $P(\epsilon) = 1 - \exp(-\epsilon/\epsilon_{FT})$ where ϵ_{FT} is the flux tube radius, as in [Monteforte and Wolf, 2012]. The results for the theta, LIF (superthreshold) and the Izhikevich model have been included as the new supplementary Figure S2, while the subthreshold response for the LIF has been stated verbally in the results section “FORCE trained Spiking Networks Learn Dynamics Rapidly with Chaotic Spiking Initialization”

Referee 2: Minor Points

16. **Line 152 – I think the word “compares” is missing.**

The wording for this line has been altered to “In order to compare the trained networks of spiking neurons with the networks of rate equations, we investigated how eigenvalues...”. We thank the referee for pointing this out.

17. **The size of the networks, their distance from chaos, the dimensionality of the inputs – all of these are in the Methods. Some parameters have important qualitative information, and should appear in the main text.**

We thank the referee for pointing this out. We have included further levels of detail in the results section. For example we have now stated the network size in the results section and the chaotic transition points in the section describing the chaotic activity of the spiking network.

18. **Line 255-257: “. . . by varying the parameters . . . statistics of RA neurons are easily reproduced”. I couldn’t find in the main text or methods any mention of this procedure. How sensitive are the statistics to the various parameters? Does anything need to be finely tuned?**

From our experience, the shape of the firing rate/ISI distributions in figures 5 D,E, correspond qualitatively with the data and are approximately the same order of magnitude. For example, the instantaneous firing rate distribution is always bi-modal due to the bursting behavior that is almost always present in these networks. The second mode is also usually >200 Hz. Some light fine-tuning was necessary for a closer quantitative match to the experimental data, but nothing exhaustive such as a large scale parameter sweep, or explicitly using something like a genetic algorithm to force an exact dependence on the experimentally measured distribution.

19. **“high dimensional” – how many dimensions? The information is in the methods, but I think it is important enough to include here as well.**

The dimensionality of the movie clip supervisor (1920) has been added.

20. **Line 301 “clock inputs were necessary both for training and recall” – this should be quantified.**

We now quantify this with r values for both training ($r < 0.44$) and recall ($r = 0.25$).

21. **Figure 1, the caption title is “rate”, but the content is spiking.**

We thank the referee for pointing this out, we used the same schematic for both rate and spiking networks as there is not adjustment to the FORCE algorithm between the two, aside from parameters. The caption title has been altered to “The Force Method Explained”

22. **Figure 1D not mentioned in caption**

This has been corrected.

23. **Figure 3D caption: “is” appears twice near parantheses.**

This has been corrected.

24. **Figure 4C – not clear what is shown here. Are these values of the readout vectors? The lack of Y axis or label does not help.**

These are indeed the readout vectors, ϕ^* , during and after the learning procedure. The y-axis has been updated to include this.

25. **Figure 5C – what are the different colors?**

The different colours correspond to the scaling coefficient, α that alters the balance between inhibition and excitation in the weight matrix after learning. Additionally, we have updated the materials and methods to describe this scaling factor in greater detail.

26. **Figure 5D caption – could be useful to specify the meaning of alpha < 1 vs > 1.**

We have updated caption in Figure 5D to describe the effect of α on the balance between excitation and inhibition post-training.

27. **Figure 5F – the definition of positive/negative is counterintuitive when one looks at the figure and its labels.**

We agree with the referee. This has been altered to be expressed as a ratio between excitation and inhibition where $\alpha > 1$ shifts the balance towards excitation and $\alpha < 1$ shifts the balance towards inhibition.

28. **Figure 5 – is alpha and excitatory gain the same thing? If yes, then using two terms in the same figure is confusing.**

We agree with the referee that this is confusing. The excitatory gain and α are indeed the same quantity. We have altered the x-axis label for 5F to only refer to α , and elsewhere in the figure caption and text.

29. **Figure 6 – the panels are very dense, and it is hard to relate the text to the correct panel (e.g. “Recall network” seems to belong to panel C)**

The space between the panels has been increased.

30. **Figure S1D – “similarity to FORCE eigenvalues”. In what sense? A reference to a specific figure would be useful.**

Both FORCE and the high-rank NEF derived weight matrices have the majority of eigenvalues distributed within a central circle, and if the FORCE Q parameter is sufficiently large, there are also a small number of dominant eigenvalues outside the central circle. The number of eigenvalues is roughly indicative of the dimensionality of the dynamics. We have added a reference to supplementary Figure S3.

31. **Dale’s law – was it used on the more complex tasks? For instance, how does the error change in the Lorenz attractor when this constraint is introduced?**

We only enforced Dale’s law on a very simple sinusoidal oscillator task. The primary reason for this is that enforcing the sign constraint on the weight is an $O(N^2)$ computation at each time step of RLS. This makes the training take much longer than necessary, and as such, we primarily stayed with simple examples such as the sinusoidal oscillator that were fast to train.

32. **Eigenvalues (like Figure S11D) should have a reference on where is the edge of chaos.**

We thank the referee for this good suggestion. We have updated the captions in all these figures to include where the transition in the G parameter is to chaos.

33. **The tasks chosen seem to be rather easy compared to previous works using FORCE on rate networks. The fact that several could be solved with $Q=0$ is another indication of this fact.**

While it is true that the two final tasks can be learned with $Q = 0$, this is because a strong enough, long and high dimensional external signal (the HDTS or clock signals) stabilizes the spiking dynamics for replay. We have demonstrated that these signals can be trained by a separate recurrent neural network (the HDTS or clock network), or simultaneously encoded into a network that generates its own HDTS in addition to the movie scene. These signals are not trivial to learn since the network has to replay a signal on a behavioral time scale with the complexity of natural stimuli.

34. **Following the previous comment, I greatly appreciate the mention of the $Q=0$ fact in the methods. Besides saving me a question in the review, I think it is an important service to the readers.**

We thank the referee for this comment.

35. **It seems that there is no way for the result of Figure 3D to be any other way. If these results tell us something new, then this should be explained better.**

We thank the referee for pointing this out, and agree that the result in Figure 3D is not surprising. The entirety of figure 3 has been moved to the supplementary materials.

36. **Many figures lack axes or labels, and the panels are sometimes too close to one another.**

We thank the referee for pointing this out. Labels have been added where missing, and the panels have been space further apart from each other (Figure 5 for example).

Referee 3: Major Points

How can we make a general purpose recurrent neural network perform a given computation? As little as ten years ago, it was almost universally believed that recurrent nets are too complicated to train directly in a supervised way. The reservoir computing framework emerged as an alternative in which the recurrent connectivity is fixed and a linear decoder is trained to generate the desired network output. This is possible because a network with complex enough dynamics implements a spatial-temporal equivalent of the kernel trick in machine learning, projecting the inputs in a high dimensional space where problems which were highly nonlinear in the input space become linearly separable. The key here is having ‘complex dynamics’, close to (but not quite) chaotic. Machine learning has built on top of this architecture to the point where we can actually tame chaos and guide it toward desired network behavior. We can now train all connections within these circuits in a supervised way (using algorithms such as FORCE, and related). Moreover, alternative frameworks such as NEF and the distributed spike-based representations from Deneve and collaborators now allow learning with spiking neural networks outside reservoir computing.

The authors present here an alternative way of learning using the original FORCE algorithm but applied to spiking neural networks. They use a vast array of tasks - from toy examples to complex realistic inputs from different domains - to illustrate the power and flexibility of the spiking neural network as a computational unit. Furthermore, the authors strive to relate the learned networks properties to a wide range of experimental findings (from the birdsong system to the hippocampus)

The numerical results are indeed impressive and I applaud the drive to link the network properties to neural data. Still, I have a range of concerns that have dampened my enthusiasm for the paper:

1. **Novelty of the approach:** unless I missed something, the training procedure is off-the-shelf FORCE applied to the instantaneous firing rates (low-pass filtered neural outputs). The authors mention in the introduction several past attempts at doing supervised learning in

spiking neurons in the FORCE framework -more involved than what is proposed here. I was wondering why do those first and what was the key insight that made learning possible here. More importantly, I was wondering if the spiking nature of the representation actually makes any difference. The authors do mention the expansion of the state space due to having additional slow time constants in the single neuron dynamics (spike frequency adaptation in the Izhikevich model) but the addition of a slow time constant in the rate model would give you the same. None of this is in itself a problem, but the manuscript needs to do a better job at spelling out what exactly is new here - the section dedicated to model comparison in the discussion reads like a history of related approaches (belonging maybe to the introduction) rather than a critical comparison.

We appreciate the referees comment on addressing the novelty of the manuscript. The referee is correct, the training procedure is off the shelf FORCE to spiking networks. However there are a few critical modifications and changes to the parameter regimes that make this possible. These can be summarized with a pair heuristics we employed to determine where the convergent parameter regimes were. These were derived after observing the qualitative behaviors of the rate networks from David Sussillo’s implementation from [Sussillo and Abbott, 2009].

In particular, it was noted in the original FORCE paper that if the amplitude of the target function is too low (see Figure 2K from [Sussillo and Abbott, 2009]), the learning does not converge to a successful solution. Furthermore, Figure 2K shows that the rate basis generated by the FORCE training did not stabilize, as it had done for Figure 2B from [Sussillo and Abbott, 2009]. We found a similar result in the spiking network, where if the feedback was too low, the resulting spiking basis would not stabilize. Our solution to this was to add the Q parameter to the feedback weight matrix:

$$\Omega = G\omega^0 + Q\eta\phi^T$$

The parameter Q has to scale with G in some fashion such that learned feedback can stabilize the dynamics. Rather than attempting to non-dimensionalize the spiking network, we applied the heuristic that Q should be large enough to have an observable effect on the spiking during training. To make this heuristic more formal, we can examine the original FORCE paper in more detail. In this paper, tuning curve used was $r = \tanh(x) \in (-1, 1)$ which is bounded. Furthermore, $g = O(1)$ with the feedback term $E\hat{x}(t)$, which is also $O(1)$ provided that $\hat{x}(t)$ was $O(1)$, and that was sufficient to stabilize the rate dynamics. Here, we can hypothesize that we need a similar balance between the order of magnitude for fluctuations in the term $G\omega^0$ and the feedback term $Q\eta\phi^T$. Operating on this hypothesis, we can justify the resulting orders of magnitude for Q for the the Theta and LIF models and for the Izhikevich model (after rescaling $r_i(t)$ to units of s^{-1}). Considering the balanced network equations and define $s_i(t)$ as the following:

$$s_i(t) = \sum_{j=1}^N G\omega_{ij}^0 r_j(t) \quad (22)$$

then a standard balanced network argument shows the following large network asymptotics:

$$\langle s_i(t) \rangle \sim 0, \quad N \rightarrow \infty \quad (23)$$

$$\langle s_i(t)^2 \rangle \sim O(G^2 \langle r_i(t)^2 \rangle), \quad N \rightarrow \infty \quad (24)$$

Thus, in order to stabilize the fluctuations generated by the balanced network we need a feedback term of $O(G\sqrt{\langle r_i(t)^2 \rangle})$. If $G = 10^{-1}$ for the LIF neuron, while the instantaneous firing rate has a $\langle r_i(t)^2 \rangle = O(10^2)$ then we require $Q = O(10)$, which is roughly what we observe. Note that this is based on the assumption that the fluctuations in $\eta\phi^T \mathbf{r}(t) = \eta\hat{x}(t)$ are $O(1)$. Higher dimensional inputs require another rescaling to reduce effect of the feedback term.

In the original FORCE paper, the time constant for the network was $\tau = 10$ ms, with $P = \alpha^{-1}I$ and $\alpha = O(1)$ with RLS applied every 20 ms (in David Sussillo’s uploaded code, [Sussillo and Abbott, 2009]). While in the previous case we could easily conduct a dimensional analysis to determine how to scale Q, G , it is more difficult here. We again employed a heuristic: the spiking basis used for an oscillator should stabilize rapidly (within the first period of oscillator presentation). This ensures that you operate similar to FORCE in the rate equations where the basis is stabilized rapidly, and the remaining weight changes are used to stabilize the weight matrix. To that end, after some trial and error we realized a smaller Δ_t

was required and our values vary between $O(10^{-1})$ to $O(1)$ ms. The effective rate parameter γ was taken to be as large as possible to ensure stability, yet maintain fast learning.

These two heuristics, stabilizing the spiking basis in the first period of supervisor presentation, and balancing the fluctuations in the static and learned recurrences to be of the same order have been added to the main text of the manuscript. The derivation of equation (4) and (5) has been added to the materials and methods (Section “ G and Q Parameters”). We hope that these additions better illustrate the novelty of our approach, and demonstrate how FORCE is so generally applicable.

We do not know whether the spiking nature of the networks makes any difference, merely that FORCE is robust to training these networks. Reservoir techniques make minimal assumptions about the underlying dynamics of the reservoir, only that the dynamics be complex and high dimensional as the referee has stated in their introduction. Here, the spiking basis yields complex initial dynamics, as a rate basis would. Interestingly however, our spiking networks could be trained prior to the transition to “rate chaos”, where the initial spike trains were more Poissonian. If we had an equivalent identical rate network with identical parameters as the spiking network, then its dynamics would be a steady equilibrium in the Poissonian spiking regime, which may not be conducive to learning. However, we have not explored this issue in greater detail and merely suggest its possibility. It is true that unlike the spike-based coding approach, we do not use spikes optimally. However, we have demonstrated that by treating the network as a reservoir we obtain the same versatility as in previously analyzed rate equations.

2. **Also I find the comparison to NEF and Deneve’s work a bit clumsy. All are cast as optimization problems, minimizing L2 reconstruction error for a linear decoder. What differs is the approach taken to find the optimum - either analytically in Deneve and co or numerically for the rest. Each version comes with it’s advantages and disadvantages. The main benefit of the version considered here is the fact that it allows greater flexibility in terms of the modeling of neurons and synapse dynamics, potentially resulting in more detailed biologically-motivated networks. The distinction between this approach and ‘artificial networks’ I find misleading - all of them are biologically-inspired artificial nets, some recurrent other not, some rate-based other spike-based, simpler or more complex.**

We thank the referee for their input on this. We concur that these techniques each come with their own advantages and yield different insights into network function. For example, FORCE training is versatile for learning individual trajectories. The NEF and spike-based coding approaches can learn a dynamical system over entire regions of phase space instead of individual trajectories. The first alteration we have made is to parts of the discussion and introduction to better and further elucidate the differences between techniques. The second alteration we have made is to remove sections which recast these techniques as being different from artificial neural networks. Indeed, the entire reservoir approach to training explicitly implies that any complicated dynamical enough dynamical system can be treated as a neural network. Finally, we have provided two copies of the revised manuscript; with and without the tracked changes from the previously submitted version.

3. **Links to biology: while a lot of experimental data is mentioned, I found most of the connections to be very superficial; lots of different observations but very little insight. I was hoping for a better understanding of what is it about the biological substrate that gives rise to the dynamical features seen here. The songbird data is pretty convincing, but the references to hippocampal replay are problematic. One critical issue is emphasizing the emergence of a prominent theta oscillation in the trained network, the amplitude of which correlates with performance. I was wondering why specifically theta - which aspects of the network make this particular power band important (frequency of inputs, of target output, time constants of neurons and synapses etc). Until I dug out the details and found that the period of the clock fed into the network (or trained to reproduce by the secondary clock generating network) is itself theta frequency (4Hz, or 8Hz)... This makes this set of results rather disappointing. There should still be something interesting in figuring out why this particular clock works with this particular input dynamics but since the input statistics are not particularly hippocampal relevant I am not sure what we can actually learn here.**

We agree with the referee that on first glance, it is disappointing that the clock confers the theta oscillation, rather than the oscillation being an emergent property of the underlying input. Thus, we have expanded this section in several intriguing directions.

First, we sought to determine if a single network could simultaneously encode its own internal clock and replay the movie simultaneously. To demonstrate this, we have trained a network to simultaneously generate its own clock, in addition to replaying the movie. The supervisor consists of 1984 dimensions (1920 pixels, 64 dimensional clock, Supplementary Figure S19) . Note that this is entirely different from the movie example we had considered before, as now there is only a single network, and all neurons simultaneously encode the clock and the movie scene, as requested by the reviewer. A figure demonstrating this result has been added to the supplementary materials (Supplementary Figure S19)

Another intriguing question that emerged is how network performance varies as a function of the clock frequency. We simulated a network of Izhikevich neurons receiving these clock signals with both varying input frequencies and varying amplitudes. Here, we found that the optimal performance occurred with higher clock amplitudes, and frequencies that were in the 8-16 Hz range (Supplementary Figure S19). Furthermore, we varied the parameters of the Izhikevich model and the synaptic connectivity and considered three cases ($C = 250, \tau_D = 20$ ms, $C = 150, \tau_D = 20$ ms, $C = 250, \tau_D = 50$ ms) to determine if the optimal frequency range was a function of the single neuron or synaptic dynamics. We found that under different parameter regimes, the optimum was still in the 8-16 Hz range with some excursions to 32 Hz for $\tau_D = 50$ ms. Thus, 8-16 Hz range is optimal and this optimum is robust to single neuron parameter values.

Finally, we have reduced the references to the hippocampus in the results section.

Minor Comments

4. **it is incorrect to say that the spike-based coding technique by Deneve and co “assume specific types of synapses or neurons” (142-43). In fact the dynamics of synapses and neurons are directly derived from the cost function.**

We agree with the referee and thank them for clarifying this. This has been reformulated, as ”.... synapses and neurons were derived from a cost function”.

5. **what is ‘probabilistic/statistical’ about the classification task? Also, if the dynamics were trained to return to rest after each input why is there still a recent history effect on the classification?**

Due to the recent history effect on the classification, the neural network classifier does not have a well defined boundary between classes. The boundaries generated by these networks are fuzzy, in the sense that an input will not be consistently classified as “Class 1” or “Class 2” every time. The recent history effect occurs because the decoded state of the network is reset to $\hat{x} = 0$. In these regimes, the current that goes to the network is given by

$$\mathbf{s}(t) = G\Omega^0 \mathbf{r}(t) + Q\mathbf{E}\phi^T \mathbf{r}(t) \quad (25)$$

$$= G\Omega^0 \mathbf{r}(t) + Q\mathbf{E}\hat{x}(t) \quad (26)$$

$$\approx G\Omega^0 \mathbf{r}(t) \quad (\text{Between inputs}) \quad (27)$$

and thus the network returns to chaotic dynamics, with the previous classification $\hat{x}(t)$ determining where in the high-dimensional state space the currents $\mathbf{s}(t)$ initialize the subsequent chaotic dynamics in.

6. **the increase in variance “when the network fails to classify consistently” seems like a truism to me. The lack of consistency is by definition a fluctuation between up and down states, which of course increases voltage variance.**

We thank the referee for pointing this out, and we agree. Thus, we have moved this result (in addition to the entire classifier example) to the supplementary material. Additionally, we have rewritten how these results are presented as indeed, an increase in variance would be expected given inconsistent classification.

7. **as far as I can tell, what it is investigated in the spatio-temporal pattern learning tasks is spontaneous replay. When people talk about recall -in the context of memory in particular- we tend to think about cued recall, where the network receives the beginning of the sequence and the network completes the rest. it would be important to explicitly clarify the distinction, I think.**

We thank the referee for pointing this out, and it is the case that we our networks are displaying spontaneous replay. The term has been altered to “spontaneous replay” or “replay” in the manuscript from recall and the difference between replay and cued recall has been made explicit in the text. This is done in the movie replay section “High-Dimensional Temporal Signals Help FORCE Trained Networks to Encode and Retrieve Proxies of Episodic Memories”

8. I found the explanation of the training procedure in 598-601 difficult to follow.

This training procedure on lines 598-601 has been rewritten to include more detail:

“The networks were initialized with the static matrix with 10% connectivity. This induces chaotic spiking in the network. We allow the network to settle onto the chaotic attractor for a period of 5 seconds. After this initial period, RLS is activated to turn on FORCE training for a duration of 5 seconds. After this period of time, RLS was deactivated for the remainder 5 seconds of simulation time for all teaching signals except signals that were the product of sinusoids...”

9. Memorizing spatio-temporal patterns: is it the higher dimensionality of the movie task that makes the clock strictly necessary whereas this was not true for the songbird task? Why is FORCE learning not able to come up with a solution relying on an internal clock directly but rather it always needs a separate subnetwork trained in a supervised way to generate the clock?

We thank the referee for the comment. To address this, we have trained a network to simultaneously generate its own clock, in addition to replaying the movie. The supervisor consists of 1984 dimensions (1920 pixels, 64 dimensional clock). This figure has been added to the supplementary materials (Supplementary Figure S19). Note that this is entirely different from the movie example we had considered before, as now there is only a single network, and all neurons simultaneously encode the clock and the movie scene, as requested by the reviewer. In both the songbird example and the movie example, the clock can be trained internally in the network. However, the songbird circuit has distinct nucleus (the high vocal center, HVC) that generates this signal and thus we opted for a similar modeling paradigm.

10. I may be missing something pretty fundamental here but how can you reverse a clock? I thought this was just an oscillation...

The input clock signal drives by activating a different assembly of neurons at a specific frequency that varies depending on the task. To elaborate further, these signals are constructed as a discretization of time, T into m sub-intervals. Each subinterval generates an extra component (dimension) in the supervisor and within the subinterval, a pulse or upward deflection of the supervisor is generated. For example, if we consider the interval from $[0, T]$ with m subintervals of width $I_n = [T(\frac{n-1}{m}), T(\frac{n}{m})]$, $n = 1, 2, \dots, m$. The supervisor is constructed as a series of pulses centered in the intervals I_n . For example, Gaussian pulses can be used

$$x_n(t) = \frac{1}{\gamma} \exp\left(-\frac{(t - T(\frac{n-1}{m}))^2}{\sigma^2}\right) \quad (28)$$

where σ controls the width and γ is used to scale the maximum of the Gaussian to 1 (unlike the typical normalization of the Gaussian where its used as a density function). Alternatively, one can also use piecewise defined functions such as

$$x_n(t) = \begin{cases} |\sin(m\pi t)| & t \in I_n \\ 0 & \text{otherwise} \end{cases} \quad (29)$$

To reverse the clock, one can flip the dimensions of the clock with dimension k becoming dimension $\text{mod}(k - m, m) + 1$. For these clock inputs, this is also mathematically equivalent to the time reversion substitution in dynamical systems theory, $\hat{t} = -t$. An identical explanation of how these clocks function has been added to the discussion section “High-Dimensional Temporal Signals”.

11. the interpretation of the Grossberg data seems dubious to me. At least more explanation is necessary.

We thank the referee for pointing this out. This paragraph has been removed from the discussion in an effort to satisfy the criticisms from Referee 1, point 1.

12. **The idea that hippocampal memory recall operates on top of a ‘backbone’ of theta oscillations is hardly novel (see Lisman and co). There is something more subtly different about what that means in this particular context, but it needs to be spelled out a lot more clearly**

We thank the referee for their comment. We have added extra detail into the discussion and results section. In particular, we now state

“Here, we show a concrete example that natural stimuli that serve as proxies for memories can be bound to an underlying oscillation in a population of neurons. The oscillation forces the neurons to fire in discrete temporal assemblies. The individual assemblies act as a partition of time that are used to learn segments of the movie scene.” with other alterations illuminating how the clock or HDTS signals function.

Typos and co.

13. **150 then-;than (a recurring mistake, check the rest too)**

This has been corrected at line 150, and throughout the manuscript.

14. **422 “serve as a backbone” for what?**

This has been corrected to ”backbone for learning”/

15. **620: fo-;of**

This has been corrected.

Additional Changes

1. The I_{bias} parameter value for the LIF model was misreported as $I_{bias} = -39$ pA instead of $I_{bias} = -40$ pA. This has been corrected. On a further note, we did however simulate the Lorenz system with this parameter value, in addition to determining where the transition to chaos was. We kept these results, but distinguished between earlier implementations in the table of parameter values
2. Supplementary Figure S4 (Now Supplementary Figure S5). This figure had poor resolution, with CV and firing rate panel identical for the theta network. We opted to regenerate this figure with identical parameter values to Figure 2C, for comparison purposes. Furthermore, we replaced the Gaussian filtered firing rates in the bottom panel with the voltage traces. We determined it was too arbitrary to filter the spike train to demonstrate the transition to heterogeneous rate dynamics. Thus, we opted instead to use the voltage traces as these are more informative and less arbitrary.
3. Supplementary Figure S5 (Now Supplementary Figure S11) The CV and firing rate distribution labels were flipped with each other. This has been corrected
4. Supplementary Figure S9 (Now Supplementary Figure S16) The time 980-1000 S was a duplicate of 960-980 seconds. This has been replaced with the correct time interval.
5. Numerous additional references have been added. These include references to novel top-down training techniques [Gilra and Gerstner, 2017, Alemi et al., 2017, Brendel et al., 2017], references to hippocampal time cells [Pastalkova et al., 2008, MacDonald et al., 2011, Eichenbaum, 2014], balanced network derivations [van Vreeswijk and Sompolinsky, 1998], a reference to the Izhikevich model parameter source [Dur-e Ahmad et al., 2012], and an additional reference to the songbird literature[Vicario and Raksin, 2000].
6. A slight stability and accuracy improvement can be made to the RLS technique we used. This is described in greater detail in the Materials and Methods section and was only used for Supplementary Figure S9 ($\tanh(x)$ rate equations from [Sussillo and Abbott, 2009] and S10. We thank Larry Abbott for bringing this to our attention.

References

- [Alemi et al., 2017] Alemi, A., Machens, C., Denève, S., and Slotine, J.-J. (2017). Learning arbitrary dynamics in efficient, balanced spiking networks using local plasticity rules. *arXiv preprint arXiv:1705.08026*.
- [Boerlin et al., 2013] Boerlin, M., Machens, C. K., and Denève, S. (2013). Predictive coding of dynamical variables in balanced spiking networks. *PLoS Comput Biol*, 9(11):e1003258.
- [Brendel et al., 2017] Brendel, W., Bourdoukan, R., Vertechi, P., Machens, C. K., and Denève, S. (2017). Learning to represent signals spike by spike. *arXiv preprint arXiv:1703.03777*.
- [Dur-e Ahmad et al., 2012] Dur-e Ahmad, M., Nicola, W., Campbell, S. A., and Skinner, F. K. (2012). Network bursting using experimentally constrained single compartment ca3 hippocampal neuron models with adaptation. *Journal of computational neuroscience*, 33(1):21–40.
- [Eichenbaum, 2014] Eichenbaum, H. (2014). Time cells in the hippocampus: a new dimension for mapping memories. *Nature Reviews Neuroscience*, 15(11):732–744.
- [Gilra and Gerstner, 2017] Gilra, A. and Gerstner, W. (2017). Predicting non-linear dynamics: a stable local learning scheme for recurrent spiking neural networks. *arXiv preprint arXiv:1702.06463*.
- [Harrison et al., 2015] Harrison, P. M., Badel, L., Wall, M. J., and Richardson, M. J. (2015). Experimentally verified parameter sets for modelling heterogeneous neocortical pyramidal-cell populations. *PLoS Comput Biol*, 11(8):e1004165.
- [Ko et al., 2011] Ko, H., Hofer, S. B., Pichler, B., Buchanan, K. A., Sjöström, P. J., and Mrsic-Flogel, T. D. (2011). Functional specificity of local synaptic connections in neocortical networks. *Nature*, 473(7345):87–91.
- [Leonardo and Fee, 2005] Leonardo, A. and Fee, M. S. (2005). Ensemble coding of vocal control in birdsong. *The Journal of Neuroscience*, 25(3):652–661.
- [Long and Fee, 2008] Long, M. A. and Fee, M. S. (2008). Using temperature to analyse temporal dynamics in the songbird motor pathway. *Nature*, 456(7219):189–194.
- [MacDonald et al., 2011] MacDonald, C. J., Lepage, K. Q., Eden, U. T., and Eichenbaum, H. (2011). Hippocampal “time cells” bridge the gap in memory for discontinuous events. *Neuron*, 71(4):737–749.
- [Mizuseki et al., 2009] Mizuseki, K., Sirota, A., Pastalkova, E., and Buzsáki, G. (2009). Theta oscillations provide temporal windows for local circuit computation in the entorhinal-hippocampal loop. *Neuron*, 64(2):267–280.
- [Monteforte and Wolf, 2012] Monteforte, M. and Wolf, F. (2012). Dynamic flux tubes form reservoirs of stability in neuronal circuits. *Physical Review X*, 2(4):041007.
- [Pastalkova et al., 2008] Pastalkova, E., Itskov, V., Amarasingham, A., and Buzsáki, G. (2008). Internally generated cell assembly sequences in the rat hippocampus. *Science*, 321(5894):1322–1327.
- [Sussillo and Abbott, 2009] Sussillo, D. and Abbott, L. F. (2009). Generating coherent patterns of activity from chaotic neural networks. *Neuron*, 63(4):544–557.
- [van Vreeswijk and Sompolinsky, 1998] van Vreeswijk, C. and Sompolinsky, H. (1998). Chaotic balanced state in a model of cortical circuits. *Neural computation*, 10(6):1321–1371.
- [Vicario and Raksin, 2000] Vicario, D. S. and Raksin, J. N. (2000). Possible roles for gabaergic inhibition in the vocal control system of the zebra finch. *Neuroreport*, 11(16):3631–3635.

Figures

Figure 1: The initial transient for the short Ode to Joy example with and without a HDTS.

Reviewers' comments:

Reviewer #1 (Remarks to the Author):

The authors have done a great job of satisfying most of my concerns. I have one remaining large concern and a minor point.

6. I am not satisfied with their answer to 6 (with a quote from the revised manuscript below).
"We were surprised to see that despite the complexity and high dimensionality of the encoding signal, the histogram of spike times across the replay network displayed a strong 4 Hz modulation conferred by the HDTS. This histogram can be thought of as a proxy for the local field potential (LFP) of a network. - This cannot stand. 6 did not answer my question, which is about the dimensionality of LFP, not it's speed.

Every example of LFP that I personally have ever looked at is a low-dimensional signal. Literally, 90% of a local cortical region are entrained by the LFP in the same way, oscillation after oscillation. I.e. it is a *low dimensional* signal. So I don't see how one can use the LFP as an example of how a brain could implement a the HDTS. The HVC comparison makes much more sense to me.

Minor point:

In the discussion, this statement is factually incorrect.

"Additionally, in [Sussillo et al., 2015], the authors used FORCE training on a rate network to reproduce the firing rates and population level responses of monkey grasping [Churchland et al., 2012]."

This is incorrect, the authors of that study used back-propagation only.

Reviewer #2 (Remarks to the Author):

Rate based recurrent neural networks have been trained in various settings: FORCE [1], reward based [2,3], using more powerful algorithms [4], while respecting Dale's law [5] and more.

Spiking based works are less abundant, and this is the primary result of the current paper. As such, I believe the paper should illustrate in the most direct manner possible:

1. FORCE training works for spiking networks
 - a. Why this works?
 - b. Why is it non-trivial? Is it more than a noisy rate network?
 - c. How well does it work? Compare to rate networks – performance, is there a one-to-one correspondence between neurons? How many spiking neurons do you need per rate neuron?
2. What additional insights can we gain from having spiking networks?

I think this is an important contribution, and the current revision does a much better job at presenting it, but in my opinion still requires more focus.

Major comments:

1. The scaling of $1/\sqrt{N}$ is very different from the $1/N$ of rate networks. Is this different from taking a trained rate network and using a large number of spiking neurons to replace every rate neuron?
2. The HDTS. First, as far as I understand, this result is not related to the spiking nature of the network, but relates to reservoir computing at large.

3. Second, my concern from the previous review (regarding time reversal) is still not answered. My understanding of HDTS is the following. Networks can be trained to produce a time varying output signal in the absence of input. Networks can also be trained to produce different such signals in response to different static inputs. HDTS is a succession of (almost) static inputs, leading to a succession of pre-trained outputs. Reversing the order of HDTS pulses should reverse the order of output segments. I could not understand from the text or movie S6 whether the output reversal was also within segments, or just the order of the segments.
4. Internal HDTS – this is a nice result, that is in line with recent work [6]. In a sense, this result shows a possible limitation of the learning algorithm. By explicitly requiring a certain internal representation, the final output of the network is better. In principle, the learning algorithm could have found this out on its own (ignoring the higher rank of the perturbation to the connectivity) – but it didn't.
5. I'm not sure my comment on $Q=0$ was understood. $Q=0$ means that the dynamics are not altered by the training process. The network is externally driven, and has its own internal dynamics, and there's readout. In this case, there is no point in using FORCE, as opposed to batch training.
6. The heuristic of "the spiking basis used for an oscillator should stabilize rapidly (within the first period of oscillator presentation)" – what do you do for non-oscillatory cases [10]?
7. The requirement to suppress chaos and stabilize the network has been studied before (e.g. [7–9], and the condition depends on the derivative of the nonlinearity. Specifically, in rate networks it is $g'^2 < r'^2 >$ smaller than one. Could it be that the condition of matching the fluctuations described in the paper is a prerequisite to this?

Minor

1. I know that I complained about NEF being tucked away in the supplementary, but I'm undecided about whether it's new place in results helps the papers or detracts from the main message.
2. Figure S2 legend "for A networks..."
3. Line 184 "anyway" -> "any way"
4. Line 186 "networks" -> "network's"
5. Line 192 the higher (lower) sentence is not very clear. Perhaps it's better to remove the "(lower)" part
6. Line 110 "For biological plausibility (1000-5000)" – not clear what this means
7. Line 82 : " The HDTS ... discretize into assemblies." Are these non-overlapping groups of neurons? Static/dynamic modes of the network?
8. Line 102-103: " η defines tuning of neurons in the network" – tuning to the output/feedback. Not tuning in general.
9. Line 170 - "In the rate based FORCE implementation, the static and learned rate equations have fluctuations of the same magnitude". Perhaps change to "The contribution of the static and learned synaptic inputs..."
10. Line 174 "RLS had to be applied on a faster time scale in the spiking...", but the numbers in parentheses are opposite.
11. Line 236: "error ... was -1.34, which is comparable to ... S8". I could't understand the relationship between the errors here. What is the scaling? Where on Figure S8 do we see $\log(L2) \approx -1.3$?

References

1. Sussillo D, Abbott LF: Generating coherent patterns of activity from chaotic neural networks. *Neuron* 2009, 63:544–557.
2. Hoerzer GM, Legenstein R, Maass W: Emergence of complex computational structures from chaotic neural networks through reward-modulated Hebbian learning [Internet]. *Cereb. Cortex* 2012, [no volume].
3. Song HF, Yang GR, Wang X-J: Reward-based training of recurrent neural networks for cognitive and

value-based tasks. eLife 2017, 6:e21492.

4. Martens J, Sutskever I: Learning recurrent neural networks with Hessian-free optimization [Internet]. In Proc. 28th Int. Conf. on Machine Learning. . 2011.
5. Song HF, Yang GR, Wang X-J: Training Excitatory-Inhibitory Recurrent Neural Networks for Cognitive Tasks: A Simple and Flexible Framework. PLoS Comput Biol 2016, 12:e1004792.
6. Enel P, Procyk E, Quilodran R, Dominey PF: Reservoir Computing Properties of Neural Dynamics in Prefrontal Cortex. PLOS Comput Biol 2016, 12:e1004967.
7. Massar M, Massar S: Mean-field theory of echo state networks. Phys. Rev. E 2013, 87:042809.
8. Rivkind A, Barak O: Local Dynamics in Trained Recurrent Neural Networks. Phys. Rev. Lett. 2017, 118:258101.
9. Rajan K, Abbott LF, Sompolinsky H: Stimulus-dependent suppression of chaos in recurrent neural networks. Phys. Rev. E 2010, 82:011903.
10. Mante V, Sussillo D, Shenoy KV, Newsome WT: Context-dependent computation by recurrent dynamics in prefrontal cortex. Nature 2013, 503:78–84.

Reviewer #3 (Remarks to the Author):

Overall I find the presentation of the materials much improved - the structuring of the material is much more straightforward. And the additional results are helpful in clarifying the properties of force trained spiking networks.

Nonetheless, the presentation of the results remains very descriptive and I am still missing some intuition of how and why things work. While the amount of simulations is huge and should not increase in any way I wonder whether the text could be further improved in this respect.

One aspect that I think should be further clarified is the nature of the 'internal' HDTS - if I haven't missed anything this is a separate network explicitly trained in a supervised way to generate a target sequence-like output which is in its turn fed into a second network trained to do the actual task of interest. As a neural network technique there's nothing wrong with doing things this way however I do wonder about the neurobiological implications -e.g where the supervision signal may originate either in the birdsong system or within the hippocampus.

Speeding up or reversing replay: is this simply because of learning element-in-HDTS to output associations such that the HDTS input actually drives the output? Or there something more complex going on?

Minor typos and co.

Abstract: "illuminating information not easily obtainable in rate networks" not sure what this means exactly

line 51-52 I don't see how the interest in alternative models leads to the conclusion ***Hence***, the necessity for a more widely applicable approach.

line 92 "confers an oscillation in a proxy of the local field potential (LFP)." needs rephrasing

line 248 remove space before .

line 317 "on its own" -> autonomously?

Revisions

Wilten Nicola and Claudia Clopath

July 31, 2017

Referee 1: Major Points

The authors have done a great job of satisfying most of my concerns. I have one remaining large concern and a minor point

1. I am not satisfied with their answer to 6 (with a quote from the revised manuscript below).
“We were surprised to see that despite the complexity and high dimensionality of the encoding signal, the histogram of spike times across the replay network displayed a strong 4 Hz modulation conferred by the HDTS. This histogram can be thought of as a proxy for the local field potential (LFP) of a network. - This cannot stand. 6 did not answer my question, which is about the dimensionality of LFP, not it’s speed.

Every example of LFP that I personally have ever looked at is a low-dimensional signal. Literally, 90% of a local cortical region are entrained by the LFP in the same way, oscillation after oscillation. I.e. it is a low dimensional signal. So I don’t see how one can use the LFP as an example of how a brain could implement a the HDTS. The HVC comparison makes much more sense to me.

We thank the referee for pointing this out. Indeed, for many cortical regions it is true that the LFP entrains the firing statistics of local populations of neurons. To address this concern, we have stopped referring to our spike-time histogram as a proxy of the local field potential and have removed references to it. We now refer to it as the “mean-activity” or “mean population activity” of the network. We also note that we previously reduced the incidences of direct comparison of the replay example and the hippocampus.

Referee 1 Minor Points

2. In the discussion, this statement is factually incorrect. ”Additionally, in [Sussillo et al., 2015], the authors used FORCE training on a rate network to reproduce the firing rates and population level responses of monkey grasping [Churchland et al., 2012].”

This is incorrect, the authors of that study used back-propagation only.

We thank the referee for clarifying this. After reviewing the supplementary materials in Sussillo et. al., 2015 we also observed that standard backpropagation was used. Thus this reference has been corrected.

Referee 2: Major Points

Rate based recurrent neural networks have been trained in various settings: FORCE [1], reward based [2,3], using more powerful algorithms [4], while respecting Dale’s law [5] and more. Spiking based works are less abundant, and this is the primary result of the current paper. As such, I believe the paper should illustrate in the most direct manner possible: FORCE training works for spiking networks. Why this works? Why is it non-trivial? Is it more than a noisy rate network? How well does it work? Compare to rate networks – performance, is there a one-to-one correspondence between neurons? How many spiking neurons do you need per rate neuron? What additional insights can we gain from having spiking networks? I think this is an important contribution, and the current revision does a much better job at presenting it, but in my opinion still requires more focus.

To address the referees concerns, we have added 5 additional figures/schematics (see detailed points below):

- Spiking vs. Rate Networks, Major Point 1.

- HDTS for general reservoirs, Major Point 2.
- Time reversal of single pixels within HDTS segments Major Point 3.
- Extension of heuristic 2 to non-oscillatory signals, Major Point 6.
- Numerical validation of the fading memory property, Major Point 7.

We remark that we are happy to include these additional figures in the supplementary materials. However, due to the length of the manuscript and the request by Referee 3 to not include more simulations, we have only included them in this response at present.

1. The scaling of $1/\sqrt{N}$ is very different from the $1/N$ of rate networks. Is this different from taking a trained rate network and using a large number of spiking neurons to replace every rate neuron?

We thank this referee for this comment, and indeed there is a substantial difference. We have implemented the case the referee refers to where we train a large number of spiking neurons to replace an individual rate neuron. Then, one can FORCE train a rate network and replace each node with a representative spiking network, as the referee has suggested. Let N be the number of neurons in each node population, and K be the number of nodes. The system of differential equations describing the population of nodes is given by:

$$\tau \dot{x}_i = -x_i + \sum_{j=1}^K \omega_{ij} \tanh(x_j) \quad (1)$$

where $\omega_{ij} = \omega_{ij}^0 + \eta_i \phi_j^T$ is obtained from FORCE training. We considered a 5 Hz sinusoidal oscillator as a supervisor. This is a test case to see if this approach would work.

In order to implement equation (1), two populations of N neurons are required per node. One population computes the non-linearity $r = \tanh(x)$ after receiving x as an input. The other population receives r_i for $i = 1, 2, \dots, K$ nodes and subsequently computes the dynamics:

$$\tau \dot{\hat{x}}_i = -\hat{x}_i + \sum_{j=1}^K \omega_{ij} \hat{r}_j$$

Thus, the secondary population acts as a leaky integrator for the input $h = \sum_{j=1}^K \omega_{ij} r_j$. Using the Neural Engineering Framework, a pair of conditions can be derived for decoders in each population. In total, the network consists of $2NK$ neurons. When N increases, we simulate the dynamics of a single node with greater accuracy while when K increases, we increase the dimensionality of the original reservoir. Unfortunately, this rapidly increases the size of the network. We considered $N = 200$ with $K = 500$ which results in a network of 200,000 neurons. Interestingly, the network can reproduce the stability to chaos transition of the classical $\tanh(x)$ at $g \approx 1$ (see Figure 1) with a random ω_{ij}^0 where $E(\omega_{ij}^0) = 0$ and $E((\omega_{ij}^0)^2) = 1/K$. Note that at this scale (in K), we are far from the thermodynamic limit from which the transition is exactly at $g = 1$. However, when using a FORCE trained ω_{ij} , the network fails to reproduce the intended dynamics (Figure 1). We used the code from [Sussillo and Abbott, 2009] with $g = 1.6$ to FORCE train the network. This is likely due to the fact that the dominant component of the weights ω_{ij} scales like \sqrt{N}^{-1} through ω_{ij}^0 . This serves to induce instability in the spiking network that can only be tamed with online training. Thus, a FORCE trained rate network cannot easily be substituted with a spiking network.

We note for clarity that there is a subtle difference between this approach, and the approach advocated by [DePasquale et al., 2016]. These authors use a rate network or a network of continuous variable equations to train every synapse in a spiking network in an online fashion by acting as a supervisor. This is different from training a spiking network to replace every rate equation, as the referee has suggested.

2. The HDTS. First, as far as I understand, this result is not related to the spiking nature of the network, but relates to reservoir computing at large

The referee brings up a very important point. Indeed, it is independent of the nature of the reservoir. It is also independent of the nature of the HDTS as well with orthogonality of the HDTS being the primary

requirement. To demonstrate this, consider an m -dimensional orthogonal basis $\mathbf{u}(t)$ that serves as our HDTS. The network is trained to approximate this basis. The previously used sinusoidal pulse HDTS basis satisfied this criterion. Indeed, the HDTS can conveniently be thought of as a orthogonal wavelet basis. After training the network, we have the following:

$$\phi^T \mathbf{r} = \mathbf{T} \mathbf{r} = \mathbf{u} \quad (2)$$

where \mathbf{T} serves as a linear operator mapping \mathbf{r} to \mathbf{u} , or more formally $\mathbf{T} : \mathbf{R} \rightarrow \mathbf{U}$ where $\mathbf{R} = \text{span}\{\mathbf{r}_1(t), \mathbf{r}_2(t), \dots, \mathbf{r}_N(t)\}$ and $\mathbf{U} = \text{span}\{\mathbf{u}_1(t), \mathbf{u}_2(t) \dots \mathbf{u}_m(t)\}$. We have not made any assumptions on the nature of $\mathbf{r}(t)$ and it can be generated by either a spiking or rate network. It is also worth mentioning that equation (2) is only approximately valid as the network approximation to the basis is not error free. However, the error is negligible for the purposes of the following argument. Applying the rank-nullity theorem yields the following:

$$\begin{aligned} \dim(\text{im}(\mathbf{T})) + \dim(\text{ker}(\mathbf{T})) &= \dim(\mathbf{R}) \\ \Rightarrow \dim(\mathbf{R}) &\geq \dim(\text{im}(\mathbf{T})) = m \end{aligned}$$

Thus, by successfully training an HDTS into a recurrent neural network (either spiking or rate), we impose a lower bound on the dimensionality of the spiking basis $\mathbf{r}(t)$. In practice however, this lower bound is very conservative as the reservoir $\mathbf{r}(t)$ receives $\mathbf{u}(t)$ (or more accurately $\hat{\mathbf{u}}(t)$) as a feedback term, and increases its dimensionality significantly greater than m .

As with most series expansions, the higher the dimensionality of the series the greater the accuracy. Unfortunately, aside from the classical Fourier series, we are not aware of equivalent results to Jacksons theorem which determines the convergence rate. Thus, little more can be said analytically.

One can verify this argument with the singular value decomposition (SVD). However, it is important to note that the singular values produced by the SVD are all non-zero, which seemingly indicates a full-rank matrix and N dimensional \mathbf{R} . This is not the case however and is an artifact of noise and finite numerical precision in computing singular values. One can compute the effective rank or dimensionality of $\mathbf{r}(t)$ by setting a threshold in the singular values generated, λ_s . This procedure is well documented in [Konstantinides and Yao, 1988]. The effective rank or dimension of $\mathbf{r}(t)$ is determined by the number of singular values such that $\lambda_i > \lambda_s$ where λ_s can be set in numerous ways [Konstantinides and Yao, 1988]. Applying an HDTS (with a suitably high dimension) in addition to the intended supervisor increases the magnitude of all singular values except for the largest ones. This implies a greater basis dimensionality, independent of the threshold λ_s , and thus a greater accuracy in the approximation of an arbitrary supervisor, $x(t)$. This is demonstrated numerically in Figure 2 with the $\tanh(x)$ rate network considered by [Sussillo and Abbott, 2009]. We have added a sentence in ‘‘High-Dimensional Temporal Signals Improve Replay and Encoding of the Ode to Joy Song’’ stating the potential benefits of an HDTS to rate networks as well.

3. **Second, my concern from the previous review (regarding time reversal) is still not answered. My understanding of HDTS is the following. Networks can be trained to produce a time varying output signal in the absence of input. Networks can also be trained to produce different such signals in response to different static inputs. HDTS is a succession of (almost) static inputs, leading to a succession of pre-trained outputs. Reversing the order of HDTS pulses should reverse the order of output segments. I could not understand from the text or movie S6 whether the output reversal was also within segments, or just the order of the segments.**

We apologize for the previous response. The referee brings up a good point with regards to intra-segment reversal. Indeed, we agree with the referee that reversing the order of the HDTS would reverse the order of the network output segments. To investigate whether or not the time reversal occurs within a segment, we performed the time reversal while only plotting the temporal evolution of a few pixels for reference, $\hat{x}_i(t)$ (Figure 3). Additionally, we have plotted the time reversed supervisor, for comparison. Figure (3) illustrates that intra-segment reversal is not occurring. For the forward replay, the approximation is more accurate as the decoder accounts for the dynamics of the reservoir in addition to the HDTS output. Both sources are used optimally to decode the pixels. However, it appears that in reverse replay, the HDTS inputs are likely driving the decoding, and thus there is a corresponding loss of accuracy ($r = 0.98$ vs. $r = 0.9$). As the referee states, this is due to the intra-segment dynamics of the network not being

time-reversed. A statement addressing this has been added to the section “FORCE Training and Other Top Down Approaches”

4. **Internal HDTS – this is a nice result, that is in line with recent work [6]. In a sense, this result shows a possible limitation of the learning algorithm. By explicitly requiring a certain internal representation, the final output of the network is better. In principle, the learning algorithm could have found this out on its own (ignoring the higher rank of the perturbation to the connectivity) – but it didn’t.**

We thank the referee for referring [Enel et al., 2016]. After reviewing reference [6.], we agree that our work and [Enel et al., 2016] both show that specific internal representations of a supervisor are superior. This reference has been added to the section “FORCE Training and Other Top Down Approaches”. We also agree with the referee that the learning algorithm could have found this representation on its own. By imposing a dynamically richer representation, FORCE training performed better.

5. **I’m not sure my comment on $Q=0$ was understood. $Q=0$ means that the dynamics are not altered by the training process. The network is externally driven, and has its own internal dynamics, and there’s readout. In this case, there is no point in using FORCE, as opposed to batch training.**

We agree with the referee that in the $Q = 0$ case, the network can indeed be batch trained (or trained in a non-online fashion) and thus FORCE training is not necessary. Indeed, it is the classical liquid state machine implementation.

6. **The heuristic of “the spiking basis used for an oscillator should stabilize rapidly (within the first period of oscillator presentation)” - what do you do for non-oscillatory cases [10]?**

We thank the referee for this question as it brings up an interesting point: the encoding properties of neurons on complicated attractors. For non-oscillatory cases, a reasonable assumption to make is that the spiking basis is stabilized with respect to regions of phase space. In particular, consider the phase space X of the target dynamics $x(t)$ in the small neighbourhood Ω_X . The heuristic can now be rephrased as subsequent incursions into Ω_X should yield identical (or similar) network spiking dynamics as the initial incursion into Ω_X during training. For example, we can consider the Lorenz system with a small region on the Lorenz attractor (Figure 4). When the spiking trajectory re-enters this region, the spike train generated for the initial visit and subsequent visit should be similar. This rephrases the heuristic for the spiking basis in terms of the phase space of the dynamical system, as opposed to time. The heuristic is identical for oscillatory systems due to the relationship between time and the phase of an oscillator.

7. **The requirement to suppress chaos and stabilize the network has been studied before (e.g. [7–9]), and the condition depends on the derivative of the nonlinearity. Specifically, in rate networks it is $g^2 < r^2 >$ smaller than one. Could it be that the condition of matching the fluctuations described in the paper is a prerequisite to this?**

This is an excellent suggestion. In particular, [Rivkind and Barak, 2017] provides a strong argument as to how convergence is achieved. The authors analyze two rate systems:

$$\dot{x} = -x + W\phi(x) + w_{FB}f + w_{in}u \quad (\text{open loop or echo-state}) \quad (3)$$

$$\dot{x} = -x + W\phi(x) + w_{FB}(w_{OUT}^T\phi(x)) + w_{in}u \quad (\text{closed loop}) \quad (4)$$

and consider static f with either multiple possible steady states or a single one. The important derivation is that in the closed loop system, exactly M of the eigenvalues can fall either inside or outside the central cluster of eigenvalues. The authors also prove in a supplementary that the fading memory property is a necessary condition for FORCE training rate networks. We think that the referee is correct, that matching the fluctuations in magnitude is likely a prerequisite to at least achieving the fading memory property. Then, a more thorough parameter search can ensure convergence with either dominant eigenvalues or non-dominant eigenvalues depending on the relative strengths of (G, Q) . Thus, we have added this reference in the discussion section “FORCE training and other top-down approaches”.

The fading memory property can also be demonstrated numerically by implementing the closed loop system as a spiking network in Figure 5. The results in Figure 5, in addition to our previous results on the peri-stimulus time histograms of the Ode to Joy example reveals a critical difference with the rate networks. In the spiking networks, the spike train does not have to be stabilized for convergence (see

Figure 5, $Q = 5$, $Q = 7$) with larger Q increasing the stability of the spike train. The rates however do stabilize. This is also reminiscent of our second heuristic, as during FORCE training RLS essentially clamps the network output to the supervisor.

Referee 2: Minor Points

1. **I know that I complained about NEF being tucked away in the supplementary, but I'm undecided about whether it's new place in results helps the papers or detracts from the main message.**

We appreciate the referees comments on this, and concur that it does detract from the main result of the paper that FORCE training works with spiking networks. Thus, we have opted to remove the NEF Supplementary section S1, and Supplementary Figure S1 in addition to references to it in the manuscript.

2. **Figure S2 legend “for A networks. . .”**

This has been corrected for “for a network of”

3. **Line 184 “anyway” -> “any way”**

This has been corrected

4. **Line 186 “networks” -> “network’s”**

This has been corrected

5. **Line 192 the higher (lower) sentence is not very clear. Perhaps it's better to remove the “(lower)” part**

This has been rewritten to make it clearer.

“Oscillators with higher frequencies are learned over larger (G, Q) parameter regions in networks with faster synaptic decay time constants, τ_D (see Supplementary Figures S5-S7). Oscillators with lower frequencies required slower synaptic decay time constants”

6. **Line 110 “For biological plausibility (1000-5000)” – not clear what this means**

This has been reworded to

“All networks considered were intermediate in size, consisting of 1000-5000 neurons, depending on the particular application”

7. **Line 82 :“ The HDTS . . . discretize into assemblies.” Are these non-overlapping groups of neurons? Static/dynamic modes of the network?**

This is an excellent question. The neurons can belong to multiple assemblies and thus can overlap. This is demonstrated in Supplementary Figure S20 (the spike raster), or Supplementary Figure S17 (the voltage traces). In both networks, we can see that the spiking is modulated by a low frequency oscillation with a period of the width of an HDTS segment. This is due the fact for each component of the HDTS pulse, $\hat{x}_i(t)$, neuron j will receive the input $Q\eta_{ij}\hat{x}_i(t)$. If $\eta_{ij} < 0$, then neuron j is inhibited by the HDTS during that component of time while if $\eta_{ij} > 0$, neuron j is excited by the HDTS. This alone does not determine if a neuron will spike during the HDTS, as the static component of the weight matrix also contributes to the spiking activity of a neuron. As the η_j are randomly distributed over $[-1, 1]$ and static, a neuron has equal probabilities of being inhibited or excited by the HDTS component. Thus, a neuron can belong to multiple assemblies.

8. **Line 102-103: “ η defines tuning of neurons in the network” – tuning to the output/feedback. Not tuning in general.**

The referee is correct, and this has been modified.

9. **Line 170 - “In the rate based FORCE implementation, the static and learned rate equations have fluctuations of the same magnitude”. Perhaps change to “The contribution of the static and learned synaptic inputs. . .”**

This has been amended to *“The contributions of the static and learned synaptic inputs are of the same magnitude”*

10. **Line 174 “RLS had to be applied on a faster time scale in the spiking...”, but the numbers in parentheses are opposite.**

The referee is correct, and the statement has been reversed.

11. **Line 236: “error ... was -1.34, which is comparable to ... S8”. I couldn’t understand the relationship between the errors here. What is the scaling? Where on Figure S8 do we see $\log(L2) \approx -1.3$?**

In Supplementary Figures S5-S7, we have plotted the $\log(L_2)$ error in the range of $[-1, -5]$ and we have classified these results as convergent. Thus, -1.34 is within the range classified as convergent, albeit in the region of less accuracy. The colour legend is in the top right corner of these figures.

Referee 3: Major Points

Overall I find the presentation of the materials much improved - the structuring of the material is much more straightforward. And the additional results are helpful in clarifying the properties of force trained spiking networks.

Nonetheless, the presentation of the results remains very descriptive and I am still missing some intuition of how and why things work. While the amount of simulations is huge and should not increase in any way I wonder whether the text could be further improved in this respect.

1. **One aspect that I think should be further clarified is the nature of the ‘internal’ HDTS - if I haven’t missed anything this is a separate network explicitly trained in a supervised way to generate a target sequence-like output which is in its turn fed into a second network trained to do the actual task of interest. As a neural network technique there’s nothing wrong with doing things this way however I do wonder about the neurobiological implications -e.g where the supervision signal may originate either in the birdsong system or within the hippocampus.**

We appreciate the referees comment on this point. There were two implementations we considered for the internal HDTS. In the first case, a separate sub-network was considered that implemented the HDTS. The HDTS then fed into the primary network responsible for spontaneous recall as an input. This is exactly as the referee describes. In the second case, both the stimulus to be recalled and the HDTS were simultaneously encoded in a single, isolated, recurrent neural network. Thus, both implementations are possible. We have examined the experimental data to determine which implementation is more representative of biological circuits.

For the songbird circuit, evidence is mounting that the high-vocal center (HVC) is indeed the nucleus responsible for encoding a chain of activity which outputs onto the RA nucleus. Our HDTS serves as a proxy for this chain of activity. Evidence for this hypothesis comes from [Hahnloser et al., 2002] through single unit recordings in songbirds during singing. The authors find the precise temporal code in HVC_{RA} projection neurons. This was confirmed with intracellular recordings of neurons in [Long et al., 2010]. Furthermore, these intracellular recordings demonstrated that the time-keeping nature of this circuit was a network phenomenon, and not due to subthreshold (intrinsic) currents. Additional evidence comes from the cooling experiments from [Long and Fee, 2008] where a Peltier device is used to locally cool HVC. This functions to dilate the spike train generated by HVC neurons in time. This effect is mirrored in the singing behavior which is dilated in proportion to the temperature change. However, [Mooney, 2000] has recently argued that the clock-like behavior of the HVC might be due to a distributed recurrent neural loop. This is based on computing Q_{10} values which are slopes determining how much a sequence is dilated given every 10 degrees of cooling. The authors find that the Q_{10} values computed behaviorally (for the song) are significantly lower than the Q_{10} values computed for slice experiments with voltage traces. The authors argue this is due to the time keeping or synaptic chain is controlled through a distributed network involving four different nodes, including HVC and RA in a large recurrent loop. In this hypothesis, HVC is merely the output of the chain. In short, there is evidence for the HDTS being generated by a separate network, and some evidence that it may be generated by a distributed network involving four nodes. This matter is not settled. However, in either case what is not in dispute is the existence of the HVC chain of activity and its purpose as a clock.

For the hippocampus, one can also argue that the recently discovered time cells serve a similar function to the HDTS. First discovered in CA1 pyramidal neurons [Pastalkova et al., 2008], these neurons also fire in

a precise temporal sequence during an episodic memory task. They have also been discovered in area CA3 [Salz et al., 2016]. Entorhinal grid cells also having some tuning with respect to time [Robinson et al., 2017] and that the time cells of CA1 may inherit this property from the entorhinal cortex [Robinson et al., 2017]. Additionally, abolishing the theta oscillation in the hippocampus destroys time fields of time cells but not place fields [Wang et al., 2015] in place cells. This implicates the theta oscillation as a necessary component for time cell formation. The theta oscillation (and thus time cells) was abolished through the inactivation of the medial septum (MS). We can speculate that the MS might serve a similar purpose to the HVC and thus generate time cells. However, what might also occur is an interaction between MS inputs to hippocampal area CA3 and the local intrinsic dynamics of CA3 to create time cells. Again, to the best of our knowledge the exact circuit generating time cells remains unclear. However, the existence of time cells and their apparent partitioning of behaviourally relevant time scales is not in doubt.

We do agree with the referee that the neurobiology of how an HDTS might be implemented is an important point for the readers. Thus, we have included abbreviated versions of the above paragraphs in the discussion section "FORCE Trained Networks and Songbird Behaviors" and "Episodic Memories can be Encoded and Replayed with FORCE Training Using High Dimensional Temporal Signals".

2. Speeding up or reversing replay: is this simply because of learning element-in-HDTS to output associations such that the HDTS input actually drives the output? Or there something more complex going on?

We thank the referee for this line of inquiry. To investigate whether or not the HDTS to output associations are allowing for compression/dilation, we performed the time reversal while only plotting the temporal evolution of a few pixels for reference, $\hat{x}_i(t)$ (Figure 3). Additionally, we have plotted the time reversed supervisor, for comparison. Figure (3) illustrates that intra-segment reversal is not occurring and thus the reversal is primarily due to the HDTS element reversal. For the forward replay, the approximation is more accurate as the decoder accounts for the dynamics of the reservoir in addition to the HDTS output. Both sources are used optimally to decode the pixels. However, in reverse replay, the HDTS inputs are driving the decoding, and thus there is a corresponding loss of accuracy ($r = 0.98$ vs. $r = 0.9$). This is due to the intra-segment dynamics of the network not being time-reversed. A statement addressing this has been added to the section "High-Dimensional Temporal Signals Help FORCE Trained Networks to Encode and Replay Proxies of Episodic Memories". A similar effect likely occurs for compressed replay (the lack of intra-segment compression) as the accuracy also drops to this amount at roughly $8\times$ the compression factor.

Referee 3: Minor Points

1. Abstract: "illuminating information not easily obtainable in rate networks" not sure what this means exactly

This has been reworded to

"The networks are analyzed post-training to yield information not obtainable in rate networks. Examples include spike-timing statistics, peri-stimulus time histograms, and spike density distributions with respect to the mean population activity. "

2. line 51-52 I don't see how the interest in alternative models leads to the conclusion *Hence***, the necessity for a more widely applicable approach.**

The line "Hence the necessity for a more widely applicable approach has been dropped"

3. line 92 "confers an oscillation in a proxy of the local field potential (LFP)." needs rephrasing

This sentence has been rephrased to address both this comment, and the major point from referee 1. It now reads as:

"For the episodic memory circuit, an HDTS is critical for normal functioning. The HDTS also gives rise to an oscillation in the mean population activity of the network. "

4. line 248 remove space before .

This has been corrected.

5. **line 317 "on its own" -> autonomously?**

In accordance with the comments from Referee 1, we have dropped the designation of autonomous/non-autonomous. Instead, we have rewritten this as "without external inputs"

References

- [DePasquale et al., 2016] DePasquale, B., Churchland, M. M., and Abbott, L. (2016). Using firing-rate dynamics to train recurrent networks of spiking model neurons. *arXiv preprint arXiv:1601.07620*.
- [Enel et al., 2016] Enel, P., Procyk, E., Quilodran, R., and Dominey, P. F. (2016). Reservoir computing properties of neural dynamics in prefrontal cortex. *PLoS computational biology*, 12(6):e1004967.
- [Hahnloser et al., 2002] Hahnloser, R. H., Kozhevnikov, A. A., and Fee, M. S. (2002). An ultra-sparse code underlies the generation of neural sequences in a songbird. *Nature*, 419(6902):65–70.
- [Konstantinides and Yao, 1988] Konstantinides, K. and Yao, K. (1988). Statistical analysis of effective singular values in matrix rank determination. *IEEE Transactions on Acoustics, Speech, and Signal Processing*, 36(5):757–763.
- [Long and Fee, 2008] Long, M. A. and Fee, M. S. (2008). Using temperature to analyse temporal dynamics in the songbird motor pathway. *Nature*, 456(7219):189–194.
- [Long et al., 2010] Long, M. A., Jin, D. Z., and Fee, M. S. (2010). Support for a synaptic chain model of neuronal sequence generation. *Nature*, 468(7322):394–399.
- [Mooney, 2000] Mooney, R. (2000). Different subthreshold mechanisms underlie song selectivity in identified hvc neurons of the zebra finch. *The Journal of Neuroscience*, 20(14):5420–5436.
- [Pastalkova et al., 2008] Pastalkova, E., Itskov, V., Amarasingham, A., and Buzsáki, G. (2008). Internally generated cell assembly sequences in the rat hippocampus. *Science*, 321(5894):1322–1327.
- [Rivkind and Barak, 2017] Rivkind, A. and Barak, O. (2017). Local dynamics in trained recurrent neural networks. *Physical Review Letters*, 118(25):258101.
- [Robinson et al., 2017] Robinson, N. T., Priestley, J. B., Rueckemann, J. W., Garcia, A. D., Smeglin, V. A., Marino, F. A., and Eichenbaum, H. (2017). Medial entorhinal cortex selectively supports temporal coding by hippocampal neurons. *Neuron*, 94(3):677–688.
- [Salz et al., 2016] Salz, D. M., Tiganj, Z., Khasnabish, S., Kohley, A., Sheehan, D., Howard, M. W., and Eichenbaum, H. (2016). Time cells in hippocampal area ca3. *Journal of Neuroscience*, 36(28):7476–7484.
- [Sussillo and Abbott, 2009] Sussillo, D. and Abbott, L. F. (2009). Generating coherent patterns of activity from chaotic neural networks. *Neuron*, 63(4):544–557.
- [Wang et al., 2015] Wang, Y., Romani, S., Lustig, B., Leonardo, A., and Pastalkova, E. (2015). Theta sequences are essential for internally generated hippocampal firing fields. *Nature neuroscience*, 18(2):282–288.

Figures

Figure 1: (A) Shown above are the tuning curves $r_j(x)$ of 10 randomly selected LIF neurons out of a population of 200 neurons. Linear combinations of the tuning curves are fit to the functions $f(x) = x$ and $f(x) = \tanh(x)$. The tuning curves have uniformly distributed intercepts on $[-4, 4]$ for $f(x) = x$ and on $[-5, 5]$ for $f(x) = \tanh(x)$. The maximum firing rate is distributed uniformly on $[100, 200]$ Hz. (B) The approximation of $f(x) = \tanh(x)$ using $\hat{f}(x) = \sum_{j=1}^N \phi_j r_j(x)$ where the decoders ϕ_j are determined through L_2 optimization. The resulting L_2 error over x is 0.0137 and 0.0122 for $f(x) = x$ and $f(x) = \tanh(x)$, respectively. (C) A randomly generated 4 second signal is fed into the network that serves to integrate the dynamics $\tau_s \dot{x} = -x + h$ for a single node. The network output is plotted in blue, with the desired output in black. $\tau_s = 20ms$ was used for the rate nodes. (D) A randomly generated signal is fed into the network that approximates $\tanh(x)$. The network output is plotted in blue, with the desired output in black. (E) The nodes are coupled together to simulate the rate dynamics $\tau_s \dot{\mathbf{x}} = -\mathbf{x} + g\omega\mathbf{r}$ where \mathbf{r} and ω is a balanced weight matrix. The network consists of $2KN$ populations of spiking neurons for the K nodes in the rate network. We considered $K = 500$ with $N = 200$ spiking neurons per node. For $g < 0$, the spiking network has a stable equilibrium at $\mathbf{x} = 0$. (F) For $g > 1$, the spiking network behaves chaotically. (G) We FORCE trained a rate network to generate the weight matrix $\omega_{ij} = g\omega_{ij}^0 + \eta_i \phi_j^T$ on a 5 Hz sinusoidal oscillator. We used ω_{ij} to couple the nodes together in our large spiking network, in (E) and (F). Shown is the evolution of the firing rate for 5 randomly selected nodes. The rates remained chaotic. (G) The decoders ϕ^T determined from FORCE optimizing ϕ the rate equations are used to decode the firing rate of the K nodes. The supervisor (black) is not well approximated by the network (blue).

Figure 2: (A) The supervisor $f(t) = 1.3\sin(\pi t/60) + 0.65\sin(2\pi/50) + 0.22\sin(3\pi t/60) + 0.43\sin(4\pi t/60)$ (red dashed line) is FORCE trained in a network of $N = 1000$ $\tanh(x)$ neurons with an HDTS (blue) and without an HDTS (black). This is the default example used in the code from [Sussillo and Abbott, 2009]. (B) The target dynamics and the HDTS are plotted in time during the test phase. The dimension index is shown in the y axis. The HDTS consists of 16 sinusoidal components, evenly distributed over the period of the oscillation. As in the spiking neuron case, the HDTS increases replay accuracy. As shown by the drift in the network with an HDTS in (A), it decreases the frequency error. (C) The N singular values are plotted for the network with an HDTS (blue) and without an HDTS (black). Including the HDTS increases the magnitude of all the singular values, with the exception of the first few.

Figure 3: Shown above is the time evolution for 5 randomly selected pixels (black) in addition to the network approximant (blue). The HDTS is presented in either forward (top) or reversed (bottom) replay modes. The movie is replayed twice, in both cases.

Figure 4: A trajectory of the Lorenz system (right) is presented as a feedback term to a spiking LIF network consisting of 5000 neurons. The raster plot for 100 neurons is shown on the left. When the Lorenz system re-enters a region of phase space (right, marked in red twice), the spiking basis is largely identical (left, marked in red twice) and thus has stabilized during training.

Figure 5: A network of 2000 LIF neurons is simulated as a closed loop system to determine the stability of the spiking basis with regards to a 5 Hz sinusoidal oscillator. The voltage traces of 5 randomly selected neurons are shown for 2 seconds of simulation time. A second network with an identical initial weight matrix and conditions is FORCE trained (the “open loop” system) to reproduce the sinusoidal oscillator. The supervisor (black, right column) is a 5 Hz sinusoidal oscillator. The network output (blue) is convergent to the supervisor provided that the feedback term, Q is sufficiently large. The stability of the closed loop spiking basis is indicative of the success of FORCE training in the open loop system. Note that the rates have stabilized, while the spikes still have some variability.

REVIEWERS' COMMENTS:

Reviewer #1 (Remarks to the Author):

The authors have satisfied my remaining concerns.

Reviewer #2 (Remarks to the Author):

The rebuttal letter answers all of my concerns. There are a few points in the main text, however, that I think should be modified:

1. The reference to Rivkind & Barak 2017 in the Discussion is relevant, the one in the Results not so much.
2. The reference to Enel et al 2016 in the Discussion is relevant, but unrelated to the discussion in the rebuttal letter ("By imposing a dynamically richer representation..."). I leave it to the authors on whether to include this point in the text.
3. I think the result on higher dimension with HDTS (measured using SVD) is important enough to enter the main text. The figure itself is not necessary, but I think a sentence saying that you have some intuition on why HDTS improves performance can enter the discussion.
4. There's a typo in the time reversal part: "...due to the fact **THAT** temporal dynamics of the network are not reversed..."

Reviewer #3 (Remarks to the Author):

The authors have addressed my concerns to a reasonable degree. The final paper provides a detailed account of different types of dynamic behaviours that a spiking neural network can be taught to exhibit. While the biological relevance of the results remains limited, the paper does make a significant effort to link the results to known physiology.

Response to Referees

Wilten Nicola and Claudia Clopath

September 30, 2017

Reviewer 1

The authors have satisfied my remaining concerns.

We thank the referee for all their previous comments.

Reviewer 2

The rebuttal letter answers all of my concerns. There are a few points in the main text, however, that I think should be modified:

1. **The reference to Rivkind & Barak 2017 in the Discussion is relevant, the one in the Results not so much.**

We thank the referee for this comment. The reference in the results has been removed while the reference in the discussion has been retained.

2. **The reference to Enel et al 2016 in the Discussion is relevant, but unrelated to the discussion in the rebuttal letter (“By imposing a dynamically richer representation...”). I leave it to the authors on whether to include this point in the text.**

We thank the referee for this comment. The reference to Enel et. al., has been retained in the discussion. Due to word limit constraints, the reference has not been incorporated with regards to the previous discussion in the rebuttal letter.

3. **I think the result on higher dimension with HDTS (measured using SVD) is important enough to enter the main text. The figure itself is not necessary, but I think a sentence saying that you have some intuition on why HDTS improves performance can enter the discussion.**

We agree with the referee that the SVD result is important enough to enter the main text. Unfortunately, due to word limit constraints in the final version of this manuscript, we could not find suitable room to adequately describe this result in the main text.

4. **There’s a typo in the time reversal part: “...due to the fact ****THAT**** temporal dynamics of the network are not reversed...”**

We thank the referee for bringing this to our attention. This has been corrected.

Reviewer 3

The authors have addressed my concerns to a reasonable degree. The final paper provides a detailed account of different types of dynamic behaviours that a spiking neural network can be taught to exhibit. While the biological relevance of the results remains limited, the paper does make a significant effort to link the results to known physiology.

We thank the referee for all their previous comments.